# Recurrent RNA edits in human preimplantation potentially enhance maternal mRNA clearance

Yang Ding[1,2,6], Yang Zheng[1,3,6], Junting Wang[1,6], Hao Li [1], Chenghui Zhao[4], Huan Tao[1], Yaru Li[1], Kang Xu[1], Xin Huang[5], Ge Gao [2✉], Hebing Chen [1✉] & Xiaochen Bo [1✉]

Posttranscriptional modification plays an important role in key embryonic processes. Adenosine-to-inosine RNA editing, a common example of such modifications, is widespread in human adult tissues and has various functional impacts and clinical consequences. However, whether it persists in a consistent pattern in most human embryos, and whether it supports embryonic development, are poorly understood. To address this problem, we compiled the largest human embryonic editome from 2,071 transcriptomes and identified thousands of recurrent embryonic edits (>=50% chances of occurring in a given stage) for each early developmental stage. We found that these recurrent edits prefer exons consistently across stages, tend to target genes related to DNA replication, and undergo organized loss in abnormal embryos and embryos from elder mothers. In particular, these recurrent edits are likely to enhance maternal mRNA clearance, a possible mechanism of which could be introducing more microRNA binding sites to the 3'-untranslated regions of clearance targets. This study suggests a potentially important, if not indispensable, role of RNA editing in key human embryonic processes such as maternal mRNA clearance; the identified editome can aid further investigations.

[1] Institute of Health Service and Transfusion Medicine, Beijing 100850, China. [2] Biomedical Pioneering Innovation Center (BIOPIC), Beijing Advanced Innovation Center for Genomics (ICG), Center for Bioinformatics (CBI), and State Key Laboratory of Protein and Plant Gene Research at School of Life Sciences, Peking University, Beijing 100871, China. [3] State Key Laboratory of NBC Protection for Civilian, Beijing 102205, China. [4] Medical Innovation Research Division of Chinese PLA General Hospital, Beijing 100853, China. [5] Beijing Institute of Radiation Medicine, Beijing 100850, China. [6] These authors contributed equally: Yang Ding, Yang Zheng, Junting Wang. ✉email: gaog@mail.cbi.pku.edu.cn; chb-1012@163.com; boxc@bmi.ac.cn

The successful development of human embryos is based on the stringent gene regulation across the central dogma[1], among which several types of posttranscriptional modifications have been confirmed to contribute to maternal mRNA clearance. The dysregulation of such clearance could lead to severe developmental defects in non-human model organisms[2–4], and has been observed frequently in arrested embryos from patients[5]. Few of these discoveries, however, have examined the famous adenosine-to-inosine (A-to-I) RNA editing (referred to simply as RNA editing thereafter)[6].

As one of the well-known posttranscriptional modifications, RNA editing in humans converts the adenosines into inosines on double-stranded RNA sequences using the two adenosine deaminase acting on RNA (ADAR) family of enzymes, ADAR1 and ADAR2[7]. Because inosines are more like guanosines than the original adenosines, such editing can have various functional consequences, including the generation of non-synonymous substitutions during translation (recoding)[8] or novel protein isoforms due to altered splicing[9], the alteration of microRNA-target binding affinity[10,11], and the disruption of long stem loops in endogenous mRNA that might aid the self-tolerance of innate immunity[12]. In addition, previous studies have identified several disease-informative edits[13], suggesting their potential role in key developmental processes. Therefore, it is likely that RNA editing also plays an important role in human early embryonic development, possibly via a few key edits and/or a genome-wide tuning of editing activity.

The overall landscape of RNA editing in humans has been extensively studied before across various healthy adult tissues, with millions of edits identified[14–22]. These edits are mostly preferred on Alu elements on non-coding regions like introns and untranslated regions, rather than on coding sequences of mRNAs[7,15,20], and the editing levels of these edits in non-repetitive coding regions vary more between tissues than editing levels in repetitive regions[21]. In particular, an in silico estimated ~40% of human 3′-untranslated region (3′-UTR) edits may affect microRNA binding sites (MBSs), which possibly affects the targeting of many microRNAs[19]. These studies, however, have not examined human early embryos, and whether and how RNA editing could consistently contribute to human embryonic development remains largely unclear. Several recent studies have been conducted to investigate edits in human embryos using pilot embryo RNA-sequencing (RNA-seq) datasets[23–25], but the sample sizes have been limited and whether their conclusions drawn apply to most embryos remains unclear. In addition, the rapid primate-specific expansion of Alu elements in mRNAs[26,27], which are hotspots of RNA editing[28], hinders the determination of the functional role of RNA editing in human embryos by simple examination of their non-primate model organism counterparts[29].

In this study, we compiled, to the best of our knowledge, the first systematic A-to-I editome for human embryonic development based on 2071 embryonic RNA-seq samples. We then confirmed the existence of per-stage Recurrent Embryonic Edits (REEs; edits observed in ≥50% of samples) along with several lines of evidence suggesting their potential functions in human early embryonic development. In particular, we discovered a likely supportive role of REEs in enhancing maternal mRNA clearance, one of whose possible mechanisms is through the regulation of microRNA-based mRNA decay.

## Results

### Construction of an adapted identification pipeline for 2071 human embryonic RNA-seq datasets. Screening for systematically published datasets in the National Center for Biotechnology Information's Gene Expression Omnibus database[30] yielded a catalog of 2071 samples in 29 groups defined by developmental stages and cell types related to human embryonic development (Fig. 1a and Supplementary Data 1, 2). Because none of these samples have genotypes available, we chose a stringent approach with the use of RNA-seq-data alone[18] for the identification of edits. In particular, we removed PCR duplicates, and required the reads to have an average quality score ≥25 and a mapping quality score ≥20 (also see Supplementary Note 1 and Supplementary Fig. 1 for details of all steps and criteria). As an adaptation for RNA-seq datasets containing data on several-cell (e.g., 4-cell) and single-cell (e.g., oocytes) samples, we further minimized possible artifacts brought by genomic contamination by excluding all detected variant sites that overlapped with known genomic variants from worldwide genotyping studies (Fig. 1b and Methods)[31–34]. When tested on an independent dataset with paired DNA and RNA sequenced for each single cell[35] (Fig. 1c, Methods, and Supplementary Note 2), this pipeline generated a zero ratio of identified A-to-I RNA edits that overlapped with the DNA variants in the same cell across samples after filtering (Fig. 1d and Supplementary Fig. 2), supporting its application to the collected embryonic RNA-seq datasets.

### Identification of systematic A-to-I editome profile for human embryonic development. The application of the stringent pipeline to all 2071 curated samples resulted in the identification of a total of 989,191 editing sites in normal and other samples (Fig. 1e), with hundreds to tens of thousands of sites identified in each stage (Supplementary Fig. 3; see also Supplementary Data 3–5, and Supplementary Fig. 4 for the mapping rates, sequencing depth, and A-to-G proportions across all 12 nucleotide changes for these samples, Supplementary Figs. 5, 6 for the editing levels of these edits, and Supplementary Note 3, Supplementary Figs.7, 8 for the analysis of their Alu-editing index[36]). Consistent with previous large-scale identifications of RNA editing[18], we detected a high proportion of A-to-G mismatches (Fig. 1f and Supplementary Fig. 9), a high proportion of Alu edits among all edits similar to those in adult human tissues (as well as a previous pilot study on human early embryos[23]) (Fig. 1g), and a signature RNA-specific ADAR-binding motif across all of these sites (Fig. 1h). In addition, most such edits were located in 3′-UTR and introns (Supplementary Figs. 10–12), consistent with the observation in the previous pilot study on human early embryos[23]. These results supported the reliability of this human embryonic editome in revealing the dynamics of editing sites throughout embryonic development (see Fig. 1i for the example of the well-studied BLCAP Y2C recoding site[37]).

### Detection of thousands of organized REEs throughout early embryonic development. A per-stage search revealed that thousands of REEs were present in normal samples of all early embryonic stages (Fig. 2a, b). Compared with all observed edits, REEs were mostly located in 3′-UTR regions (Fig. 2c and Supplementary Figs. 13, 14) in addition to being mostly exonic (<50 vs. >75%; Fig. 2d and Supplementary Fig. 15). In addition, rather than being dispersed randomly like biological noises, >50% of REEs persisted through stage transitions until the 2-cell stage, and ~30% of REEs persisted through the 2-to-4-cell transition (Fig. 2e). It is also worth noting that most REEs did not disappear completely upon stage transition, although they were no longer REEs (as indicated by the scarcity of not detected edits in Fig. 2e). Furthermore, we observed that genes being targeted by REEs are likely to have their expression level drop as development progresses (Supplementary Figs. 16–21), and in most stage transitions we also observed a statistically significant (though weak as

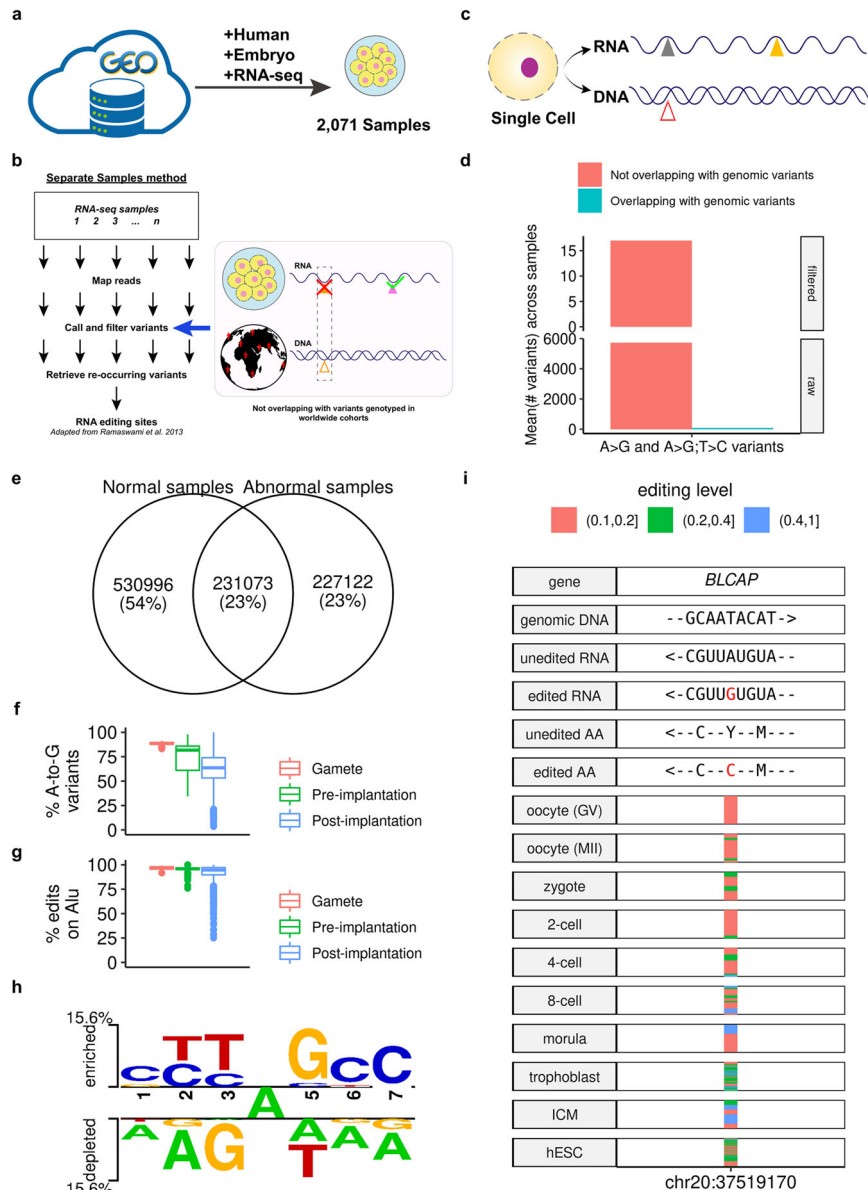

**Fig. 1 Identification and validation of the A-to-I editome for human embryos. a** Overview of RNA-seq curation from the public databases. **b** Overview of the adapted stringent pipeline. **c, d** The pipeline yielded a zero ratio of identified A-to-I RNA edits that overlapped with the DNA variants in the same cell across samples (**d**) in a paired DNA-RNA-sequencing dataset for single cells (**c**). The "#" denotes count. Also, see Supplementary Fig. 2 for the distribution of all possible types of nucleotide changes and the ADAR-binding motif derived from identified edits. **e** Total number of edits identified in all samples. The normal sites are those sites identified from 1797 normal, healthy samples, while the abnormal sites are those identified from 274 pathological samples (e.g., samples undergoing uniparental disomy (UPD) from Dataset GSE133854[38]), or samples with non-control treatment (e.g., treated with amanitin as in Dataset GSE101571[85]). Also, see Supplementary Figs. 10–12 for the comparison of the genomic distribution of normal and abnormal edits of the same stage. **f** A-to-G ratios for all variants detected across all samples. The proportion is defined as the union of strand-definite A-to-G variants and strand-ambiguous A-to-G/T-to-C variants (see Step (13) of Supplementary Note 1) to all variants. See also Supplementary Fig. 9 for this metric in Alu- and non-Alu-subsets. **g** Alu ratios for all edits across all samples. **h** The signature ADAR-binding motif computed from all edits. **i** The profile of the *BLCAP* Y2C recoding edit across stages; the horizontal stripes represent edited samples with the color denoting the editing level. Arrows indicate the direction from the 5'- to the 3'-end (for DNA and RNA) or from the N-terminal to the C-terminal (for protein). Symbols in boxplots follow the definition by "geom_boxplot" of the R package "ggplot2"[96]: the inner thick line indicates the median; the lower and upper boundaries (or hinges) of the box indicate the first and third quartiles (i.e., 25 and 75% quantiles), respectively; the upper whisker extends from the hinge to the largest value no further than 1.5 × inter-quartile range (the third quartile minus the first quartile), the lower whisker extends from the hinge to the smallest value at most 1.5 × inter-quartile range of the hinge, and data beyond the end of the whiskers are the outlier points, which are plotted individually.

being between −0.21 and −0.08) negative correlation between the editing level of each REE and the expression level of its targeted gene (Supplementary Fig. 22). These results suggest a consistent, stable pattern (and thus a possibly functional role) of (3′-UTR) REEs in early human embryonic development.

**REEs target similar genes enriched with DNA replication-related functions across early embryonic stages.** To gain insight into the functions that REEs might affect, we selected genes that are frequently targeted by REEs for each stage separately (Methods). We discovered hundreds of frequently targeted genes,

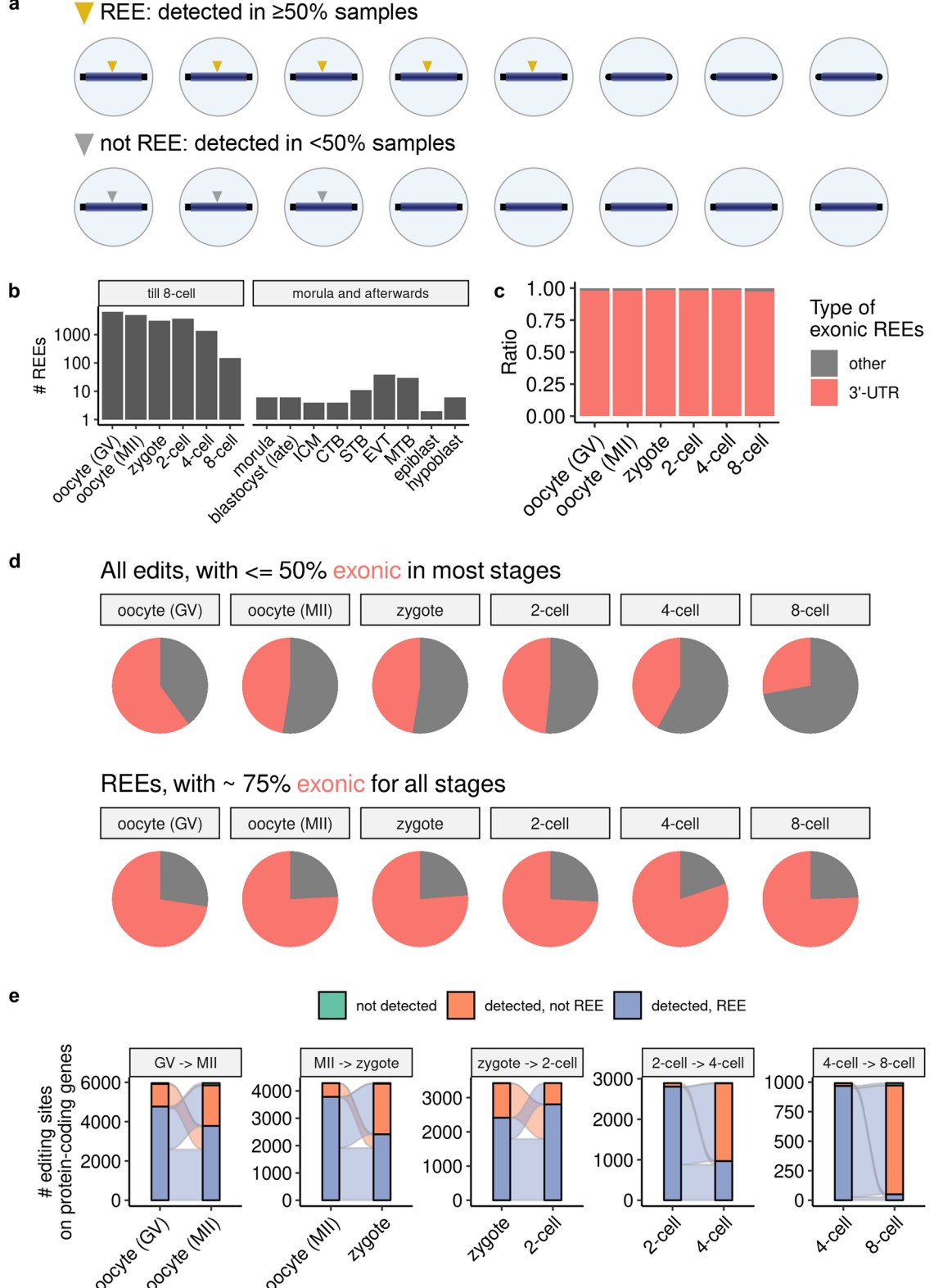

**Fig. 2 Thousands of organized REEs were detected in early human embryos. a** Definition of REE. **b** Count of REEs from normal samples per stage.
**c** Percentage of 3'-UTR REEs in early stages of embryogenesis. See also Supplementary Fig. 13 for percentages of general edits. **d** Percentage of exonic edits
and REEs (shown in red) in the early stages of embryogenesis. **e** Sankey plot describing the numbers of REEs passed to subsequent early stages. For clarity,
only REEs observed in at least one of the two stages appear in each subplot. For **c**–**e**, we considered only REEs on protein-coding genes. See also
Supplementary Figs. 14 and 15 for the percentage of 3'-UTR REEs, and the percentage of exonic edits and REEs in the late stages of embryogenesis.

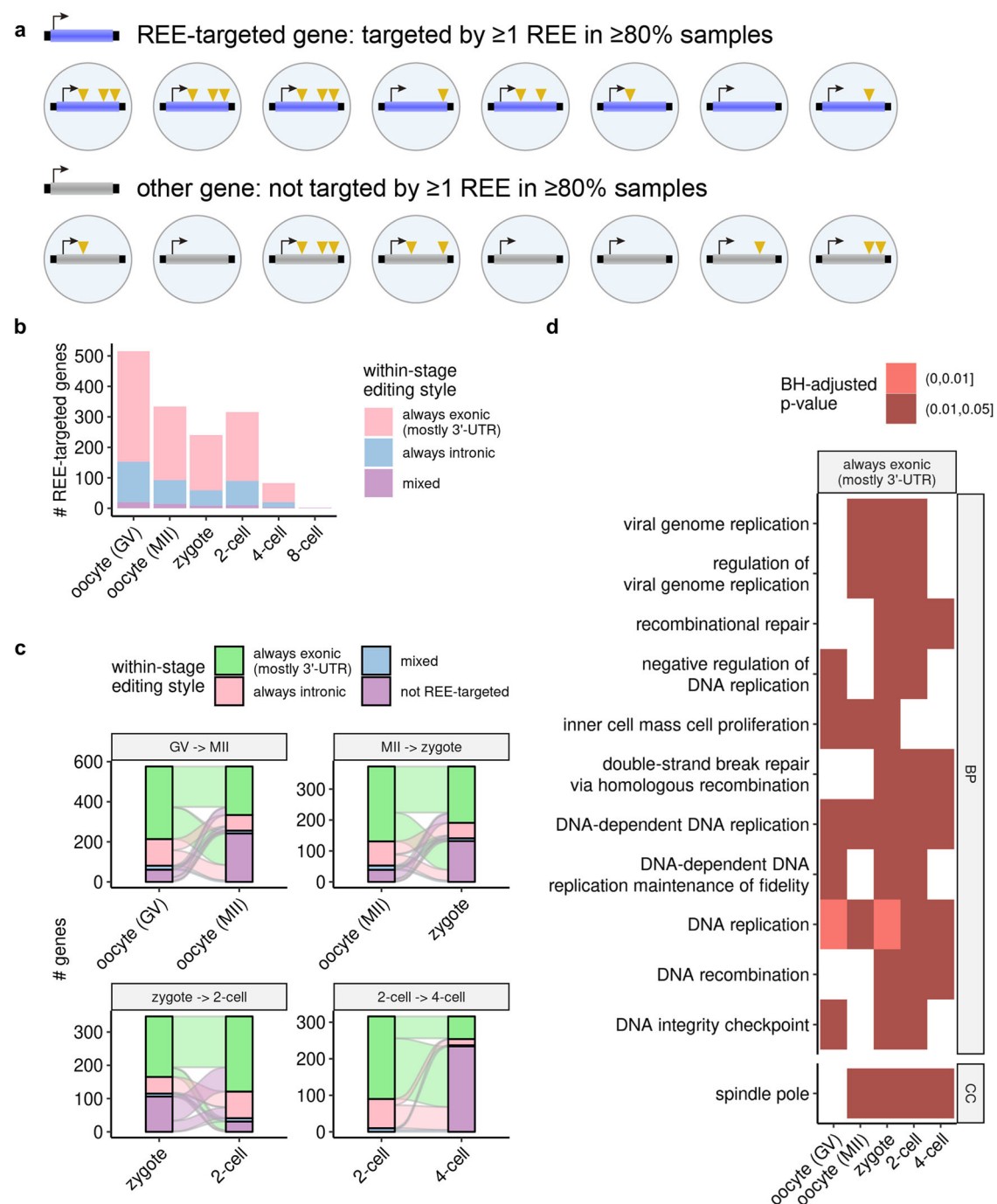

**Fig. 3 REEs target similar genes enriched with DNA replication-related functions across the early stages of embryogenesis. a** Definition of REE-targeted genes. Note that for further filtering for more reliable results, we used a more stringent criterion (edited by at least one REE in ≥80% of samples) than used for the definition of REEs. **b** Counts of REE-targeted genes per stage. **c** Sankey plot describing the number of REE-targeted genes passed to subsequent early stages. For clarity, only REE-targeted genes observed in at least one of the two stages appear in each subplot. **d** Cross-stage (≥3 stages) enriched functions of REE-targeted genes. Note that we performed enrichment analyses on 3'-UTR- and intronic-REE-targeted genes separately, and discovered cross-stage enriched functions only for the former (Methods). All *p* values were Benjamini–Hochberg-adjusted (BH-adjusted).

>50% of which were targeted primarily in 3′-UTR REEs in early embryonic stages (Fig. 3a, b). Similar to the REEs, these REE-targeted genes also displayed a large degree of overlap from the oocytes (GV) to the 2-cell stages, and most such genes observed in 4-cell embryos were also observed in the 2-cell stage (Fig. 3c). Given this consistent pattern, we investigated the specific functions that these genes share, and found that functions enriched across ≥3 stages were mostly related to DNA replication, a phenomenon observed only on genes targeted in exonic (primarily

3′-UTR) regions (Fig. 3d). These observations suggest a consistent functional impact of REEs in early human embryogenesis.

**Certain REE-matching edits could undergo organized loss in embryos with uniparental disomy and those from elder mothers.** To further investigate the functional importance of REEs, we examined for each early developmental stage whether REE-matching edits underwent an organized loss in embryos

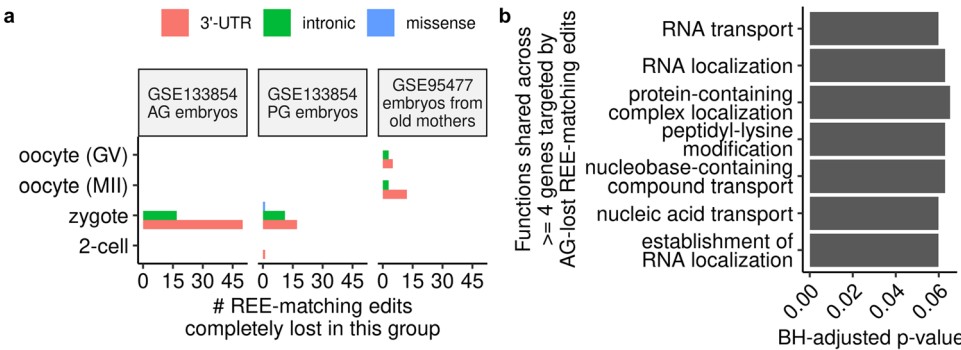

**Fig. 4 Certain REE-matching edits could undergo organized loss in embryos with uniparental disomy and those from elder mothers. a** Count of REE-matching edits that were completely lost in the abnormal uniparental disomy embryos (AG for androgenetic embryos and PG for parthenogenetic embryos) and embryos from elder mothers. See also Supplementary Figs. 23–25 for the sequence coverage of these edits in the pathological embryos/embryos from elder mothers, and Supplementary Figs. 26–28 for their editing levels in normal samples. **b** Biological processes enriched by four or more genes targeted by AG-lost REE-matching edits. Only those with Benjamini–Hochberg-adjusted (BH-adjusted) p values less than 0.1 were shown.

with particular phenotypes indicative of low embryo quality. An initial scan (see the section: Determination of the set of 107 REEs completely lost in a particular phenotypic group in Methods for more details) revealed 107 edits on 76 genes (Supplementary Data 6) that were REEs in normal embryos, but might be completely lost in the same stage in pathological embryos (GSE133854[38]) and embryos from elder mothers (GSE95477[39]) (Fig. 4a and Supplementary Figs. 23–28). These included an REE-matching edit, chr8:28,190,741, on the gene *ELP3*, the knockdown of whose mouse ortholog was shown to impair paternal DNA demethylation in mouse zygotes previously[40] (see the Supplementary IGV data (available from https://doi.org/10.5281/zenodo.7379397)[41] for its IGV plots of read alignments in all normal and PG zygotes). Gene ontology analysis of the genes with androgenetic (AG)-lost REEs revealed enrichment in various functions shared by four or more genes, and many of these functions were related to RNA metabolism (Fig. 4b and Supplementary Data 7), suggesting a potential link between these REEs and RNA metabolism in these pathological embryos.

**Targets of maternal clearance had more REE-induced micro-RNA binding sites than did nontargets**. Having gained a preliminary understanding of what genes and functions REE might affect, we then asked how REE would affect these genes. Because most exonic REEs are located in 3′-UTRs (Fig. 2d), the gene element containing most MBSs, many 3′-UTR REEs may affect genes by interfering with MBSs and thereby the microRNA-based regulatory program (see Fig. 5a for an example), a mechanism that has been studied extensively for RNA editing[10,11,19]. To confirm this, we annotated all MBSs on all editing-targeted transcripts before and after editing (with edited inosine treated as guanosine), and analyzed their associations with 3′-UTR edits. While the 3′-UTR REEs did not distinguish them from general 3′-UTR edits in the proportion of MBS-affecting edits (Supplementary Fig. 29), they were much more likely to induce MBSs if determined to overlap with MBSs (~50 vs.~33%; Fig. 5b). In particular, they were more likely to result in MBS gains than MBS losses (Fig. 5c), suggesting their potential role in the enhancement of the microRNA-mediated degradation of targeted transcripts.

Based on this observation, we speculated that REEs help to degrade mRNAs targeted by maternal mRNA clearance (referred to as clearance targets hereafter)[42] by introducing more MBSs (Fig. 5a). This hypothesis was supported by the observation that REEs result in bringing more MBSs on clearance targets than on other maternal genes (Fig. 5d; see also Supplementary Fig. 30 for the case where the net MBS change, i.e., accounting for the loss of

preexisting MBSs by REE, was considered, and also Supplementary Note 4, Supplementary Figs. 31, 32 for a preliminary case study of MBS-gaining REEs on a given gene). These results suggest a potential role of REEs (and possibly other RNA edits) in the enhancement of maternal mRNA clearance, a possible mechanism of which could be through the introduction of more MBSs.

## Discussion

By curating and analyzing the largest human embryonic editome to date, we showed that the early embryonic stages harbor thousands of REEs that are preferably exonic and highly shared between stages at the editing site and target gene levels. We also showed that these REEs could potentially enhance maternal mRNA clearance, a process that has been found to be associated with RNA editing in mouse embryos[6], one possible mechanism of which is by introducing more MBSs to clearance targets than to other maternal genes.

Although several studies have demonstrated the importance of certain editing events[43–45] and documented the adverse consequences of the disruption of one of the core editing enzymes ADAR1[46–49], the possible functional roles of RNA editing in key embryonic developmental processes remain largely unclear. Based on our observation of associations among REEs, MBSs, and maternal mRNA clearance, we propose a working model of how human embryos could take advantage of the RNA editing machine for better development: embryonic A-to-I RNA edits, including the REEs discovered in this study and possibly other accompanying edits, occur and result in the introduction of MBSs to (at least some) clearance targets more often than to other maternal genes; these targets are then more efficiently targeted and degraded by the microRNA machinery than they were in unedited form, thereby enhancing the maternal mRNA clearance (and thus the embryonic development[50–52]) (Fig. 6, left). Recent research has revealed the impairment of RNA editing in mouse oocytes upon knockout of *Cnot6l*, a deadenylase in the carbon catabolite repression 4-negative on TATA-less complex (CCR4-NOT complex) that is required for deadenylation-based maternal mRNA clearance[6]; although the roles of RNA editing in human and mice may not be directly comparable, this finding suggests that the microRNA-based effect of RNA editing on maternal mRNA clearance discovered in the present study might cooperate with other posttranscriptional modifications[2–4], possibly in an additive way[2], to advance maternal mRNA clearance. Consistent with this, previous studies have reported that the miRNA-based maternal mRNA decay pathway, if exists in embryos, might still

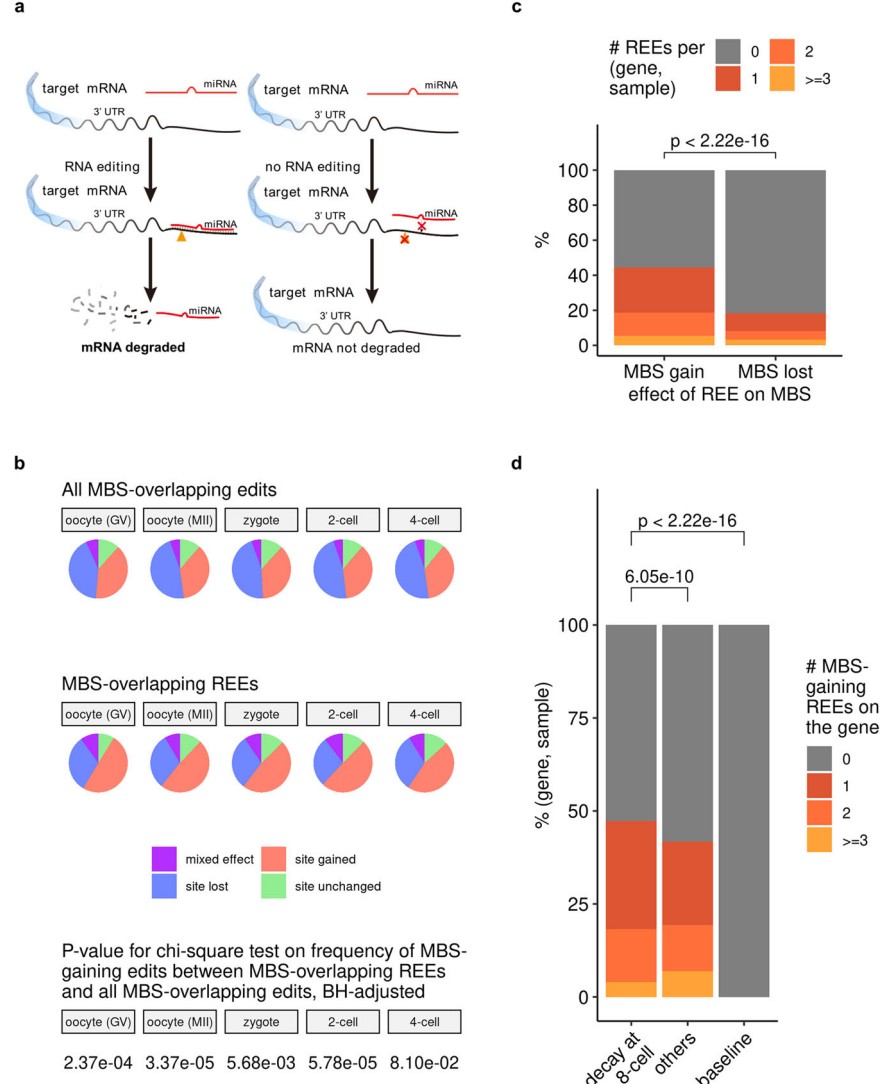

**Fig. 5 REEs induced more microRNA binding sites (MBSs) on maternal clearance targets than on nontargets. a** An example model of how RNA editing can result in the gaining of new MBSs. **b** 3'-UTR REEs are more likely to gain MBSs than general 3'-UTR edits. The BH-adjusted *p* values for the chi-square test on the frequency of MBS-gaining edits between MBS-overlapping REEs and all MBS-overlapping edits (where the null hypothesis is that whether an edit is MBS-gaining is independent of whether this edit is REE) were also shown. See Supplementary Data 25 (the source data behind Fig. 5b) for the number of site-gained (or non-site-gained) edits (or REEs) in each stage, and see also Supplementary Fig. 29, where no-overlap edits were also considered. **c** REEs are more likely to result in new MBS gains than in preexisting MBS losses. The *p* value was derived from the one-tailed, paired Wilcoxon's rank-sum test, with the alternative hypothesis being that the median value for the "MBS gain" group would be greater than that for the "MBS lost" group on each pair of (gene, sample). In total, this paired test involves 36,281 pairs of gene and sample (as described in Supplementary Data 26), and the estimated (pseudo) median and 95 percent confidence interval reported by R are 1.000017 and [1.000001, $+\infty$], respectively. **d** Genes targeted by maternal mRNA clearance have more MBS-gaining REEs than do other maternal genes (also see Supplementary Fig. 30). All *p* values were unadjusted and were derived from one-tailed, unpaired Wilcoxon's rank-sum tests, with the alternative hypothesis being that the median value for the left ("decay at 8-cell"; see the subsection: Annotation of maternal genes and targets of maternal mRNA clearance in Methods for more details about "decay at 8-cell" and "others") group would be greater than that for the right (others/baseline) group. The baseline group is just a value of 0; we plotted it here for the sake of visual clarity. The unpaired test between "decay at 8-cell" and "others" involves 17,060 and 17,755 pairs of (gene, sample) for "decay at 8-cell" and "others", respectively, and its estimated (pseudo)median and 95 percent confidence interval reported by R are $8.957086 \times 10^{-5}$ and [$8.636118 \times 10^{-5}$, $+\infty$], respectively. The test between "decay at 8-cell" and the 0 baseline involves 17,060 pairs of (gene, sample) for "decay at 8-cell", and has an estimated (pseudo)median of 1.499948 and a 95 percent confidence interval of [1.499974, $+\infty$] as reported by R.

be able to recruit PAN2-PAN3 and CCR4-NOT via the protein TRNC6A (also known as GW182) as discovered earlier[53–56]; as another well-known part of the posttranscriptional modifications, these complexes deadenylates RNA from the 3'-end to degrade them[57,58]. This might partially explain why the previously observed negative correlation between REE editing level and expression level of the targeted gene in stage transitions (Supplementary Fig. 22) was found to persist in a similar pattern on

target genes whose REE either can or can't gain additional MBSs, while began to lost on target genes that are free of predicted MBSs regardless of REE editing (Supplementary Fig. 33)—being pre-equipped with MBSs itself might be strong enough to degrade the target gene, and adding more MBSs on top of that might not accelerate the degradation much further. Therefore, the MBS-gaining edits on target genes of maternal clearance, while statistically more than on other maternal genes, might have additional

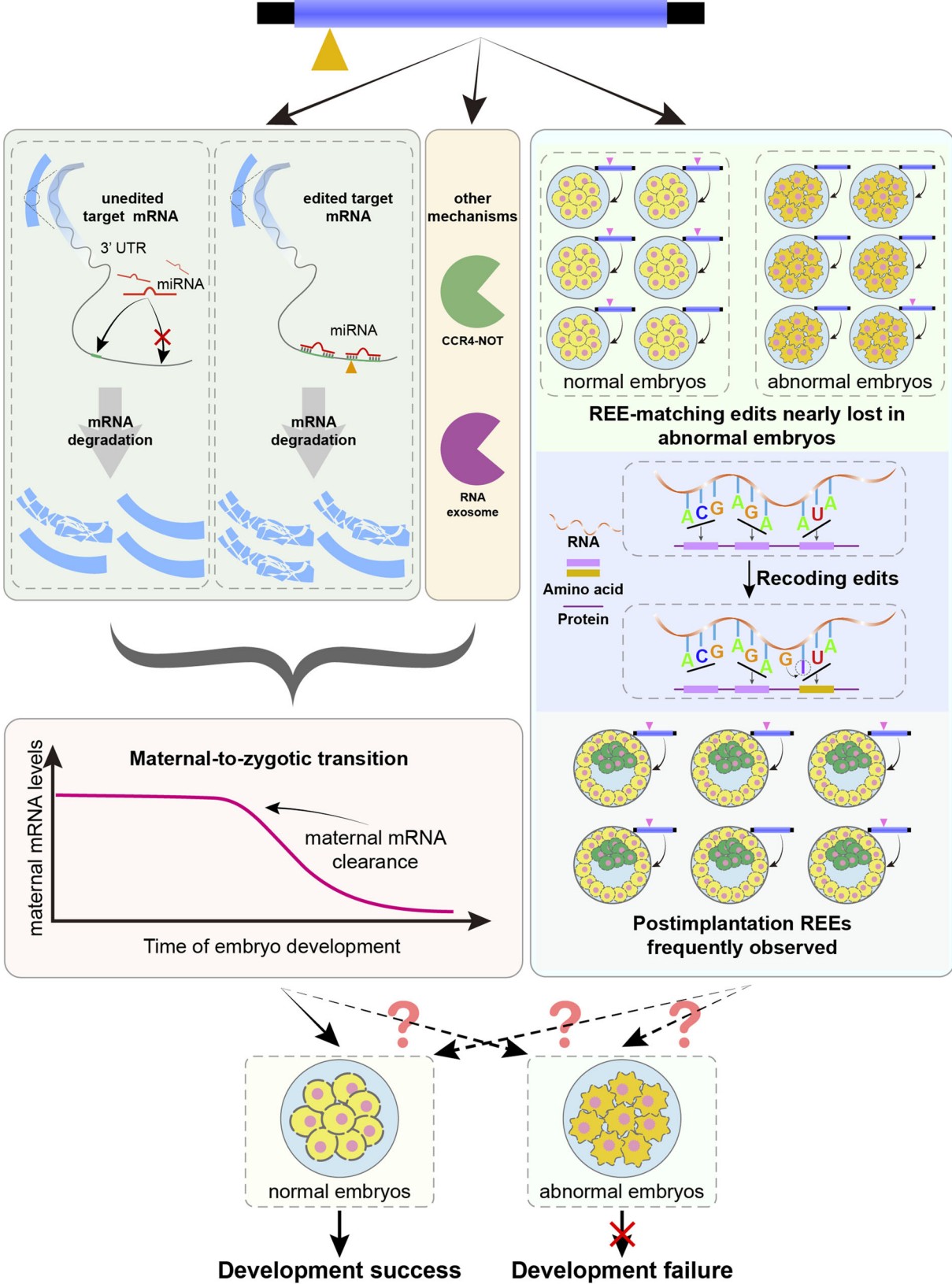

**Fig. 6 Proposed model of how human embryos take advantage of REEs (and other edits).**

functions other than recruiting the miRNA-based degradation machinery more efficiently. On the other hand, we did identify some REEs targeting some key components of RNA degradation, such as *CNOT6* for the CCR4-NOT-mediated degradation pathway[58] and *EXOSC6* for the RNA exosome-mediated degradation pathway[59] (Supplementary Note 5 and Supplementary Figs. 34, 35), suggesting other possible non-MBS roles of RNA editing in maternal mRNA clearance (Fig. 6, left).

Apart from altering the MBS count in clearance targets, REEs (and other edits) can, in theory, affect embryonic development in other ways (Fig. 6, right). In fact, in addition to the completely lost REE-matching edits identified (Supplementary Data 6), we discovered a subset of REE-matching edits that are nearly lost in cases of uniparental disomy[38] (Supplementary Data 8); these edits may be of additional critical value for scientific understanding and clinical applications. Likewise, one could also further examine the recoding edits in normal samples (Supplementary Data 9) in the editome to identify additional edits with critical functional impacts. Potentially useful insights could also be gained from the examination of REE-targeted genes (and their accompanying REEs) in postimplantation stages (Fig. 3b). Although scarce, several REE-targeted genes (Supplementary Data 10) are frequently edited by certain REEs; these REEs could be of special research interest, provided that they are validated to be non-somatic mutations by, for example, the examination of additional postimplantation embryos from independent individuals.

We'd note that, although we discovered the statistical association between REE and maternal mRNA clearance by analyzing large-scale omics datasets, currently we lack any new (experimental) data to test whether this association implies a causal relationship. In addition, we may have missed a certain number of edits (or even REEs) in the current editome due to the relatively low sequencing depth of early single-cell RNA-seq techniques and potential uncertainty brought by e.g., the random assignment of multi-mapped reads to each site, although we sought to cover as many reliable edits as possible by screening a set of thousands of samples with the application of a stringent pipeline for candidate RNA edits. More informative REEs (and their additional functions) may be discovered with deeper sequencing. This is also important for preventing the failure of identifying certain edits from wrongly identifying condition-specific loss of REEs, which might be of potential clinical values. For example, the edit at chr9:132375956 induces missense recoding during the translation of transcription termination factor 1 for ribosomal gene transcription (*TTF1*; ENSG00000125482[60]). This edit was found to be REE in normal zygotes, and was determined to be completely lost from four parthenogenetic zygotes, seemingly suggesting the potential clinical value of the detection of this edit in cases of uniparental disomy. However, a closer examination of read alignments at this site in these samples showed that such absence might arise from insufficient read coverage that failed the identification pipeline (Supplementary Fig. 36). In addition, the functional relevance of this editome to embryonic development is far from being extensively studied; specifically, REEs might be functionally important in key embryonic processes other than maternal mRNA clearance, such as those involving DNA replication and repair (as suggested by the results illustrated in Fig. 3d). It is also worth noting that the total number of REE-matching edits might be associated with certain abnormal embryos, further suggesting potential roles of REEs in the associated phenotypes (Supplementary Notes 6, 7 and Supplementary Figs. 37–41).

Another potential limitation of our work is the lack of cross-species examination that would help to locate conserved REEs (or other possible editing patterns) of important functions. Nevertheless, due to the expansion of editing-prone Alu elements in human, RNA editing in human is so strong compared to that in mouse[61] that the editing profile in adult tissue samples from human and mouse, even when only conserved editing sites were considered, would group samples by species (human vs. mouse) rather than by tissue type[21], suggesting a mouse-specific editing pattern (and thus a possibly mouse-specific working model of A-to-I edits) in mouse early embryos. Therefore, while mouse embryonic A-to-I edits might contribute to embryonic development, it is not very likely for them to work mostly by strictly following our proposed model. A possible alternative would be to investigate individual conserved edits between human and mouse (as done in adult tissues by ref. [21]) on maternal transcripts. Such conserved maternal transcripts, if exist, would suggest a heavily conserved mechanism of A-to-I editing on regulating clearance of maternal transcripts, possibly by inducing MBS.

In this study, we have introduced, to the best of our knowledge, the first large-scale A-to-I RNA editome for early human embryos, the analysis of which revealed a consistent early-stage editing pattern (of REEs) with probable functional importance in microRNA-based maternal mRNA clearance. These discoveries, along with the editome itself, are valuable resources for further examination of the interplay between RNA editing and other mechanisms involved in maternal mRNA clearance, as well as the identification of additional roles of A-to-I RNA editing in early human embryonic development.

## Methods

**Compilation of human embryonic RNA-seq datasets.** In addition to including human embryonic RNA-seq datasets whose A-to-I editomes have been studied previously[23,24], we used GEOmetadb[62] to search GEO[30] for all RNA-seq samples submitted before October 1st, 2020, using the keyword "embryo" and the species restriction of *Homo sapiens*. We filtered the datasets identified by this search to identify paired-end RNA-seq data with read length ≥75 × 2 bp, to increase the accuracy of A-to-I RNA editome identification[63]. For single-cell RNA-seq datasets, we required that the sequencing technology not be based on cell barcoding, because they are essentially single-ended RNA-seq sequencing for transcripts; the other end is used for barcoding cells and contains no information on transcript sequences. This process yielded a total of 2071 samples (1797 normal and 274 abnormal) from 18 datasets (see Supplementary Data 1, 2, 11 and Supplementary Fig. 42 for the details of these samples), which were sent to the A-to-I RNA editome identification pipeline.

**Identification of the A-to-I editome and REEs (and REE-targeted genes) within it.** We adapted a published pipeline[18] used in the Genotype-Tissue Expression A-to-I editome study[21] (see Supplementary Note 1 and Supplementary Fig. 1 for the details of the entire pipeline, including all the key steps and adaptations). Briefly, we: (1) generated a new reference genome by concatenating the hg38 assembly and all sequence fragments spanning known junction sites from the version 32 annotation of GENCODE[64]; (2) aligned quality-controlled reads to this new reference; (3) mapped these alignments back onto hg38 coordinates; (4) called variants with GATK[65]; and (5) filtered for A-to-G variants that did not overlap with common genomic variants or regions prone to algorithmic errors, and with enough read and sample support. In particular, we removed PCR duplicates, and required the reads to have an average quality score of ≥25 and a mapping quality score of ≥20. In addition, edits located in Alu elements will be included in the summary in Fig. 1e as long as detected in at least one sample, while edits located not in Alu elements will be included only if detected in at least two normal samples (or two abnormal samples) of the same stage (see Supplementary Data 2 for the details of each stage). REEs were then identified for each stage by filtering for those edits observed in ≥50% of samples in that stage. Similarly, a gene in a given stage was considered as an REE-targeted gene in that stage, if it was edited by at least one REE in ≥80% of samples in that stage.

To exclude possible artifacts in this pipeline as much as possible, we expanded the set of genomic variants used in step 5 above. Specifically, in addition to data from dbSNP version 151[66], the University of Washington Exome Sequencing Project[31] (https://evs.gs.washington.edu/EVS/), and the 1000Genomes Project[33], we used data from the Genome Aggregation Database[34] and the NCBI's Allele Frequency Aggregator project[32] which span more than hundreds of thousands of individuals to exclude variants that overlapped with population genomic variants found in these studies or projects. Variants passing through this filter are very unlikely to come from genomic variation.

**Annotation of the A-to-I editome.** We obtained from the GATK variant call format output the chromosome and position for each A-to-I edit in each sample, as

well as its read coverage (AN), the number of reads supporting the editing (AC), and the editing frequency AF which is obtained by dividing AC with AN. We then annotated these edits using SnpEff[67] with GENCODE version 32 annotation, and classified them according to their SnpEff 'Annotation' Field: coding sequence regions, 5′-UTRs, 3′-UTRs, exonic regions of non-coding transcripts, introns, and intergenic regions (Supplementary Data 12). When a given edit was of different types on different transcripts of a given gene locus (e.g., in the coding sequence region of one transcript and the 3′-UTR of another), we assigned the edit type in the following order: coding sequence >5′-UTR >3′-UTR > non-coding exonic > intronic > intergenic.

**Validation of the reliability of the adapted pipeline for cells using paired DNA- and RNA-sequencing datasets.** We validated our adapted pipeline using paired single-cell DNA-/RNA-seq datasets for the A375 cell line[68] (we did not use the dataset[69] used by ref. [35] because it is not publicly available). For each A375 cell with both DNA and RNA sequenced, we downloaded the raw reads and applied our pipeline with the following modifications: (1) we used Zachariadis et al.'s read preprocessing strategy[68] (https://github.com/EngeLab/DNTRseq); (2) whereas we applied all filters for RNA-seq data to obtain identified editing events, for DNA-Seq we stopped at the raw variant calling results generated by GATK and treated them as the ground truth for genomic variants; and (3) due to the low sequencing depth of these samples, we adjusted the read coverage filter. Specifically, we filtered for Alu edits with at least two reads covered and an editing level of at least 0.1, and for non-Alu edits additionally with at least two reads with mismatches. See Supplementary Note 2 for the full description of this validation, including its background and results.

**Motif visualization for editing sites.** We used Two Sample Logo (version 1.23)[70] to plot the ADAR-binding sequence motif. For the background sequence file (file for the -N option), we chose all 7-bp subsequences of GENCODE version 32 transcript sequences (ftp://ftp.ebi.ac.uk/pub/databases/gencode/Gencode_human/release_32/gencode.v32.transcripts.fa.gz) whose fourth nucleotide was adenine.

**Determination of the set of 107 REEs completely lost in a particular phenotypic group.** We started with all REEs identified in oocytes (GV), oocytes (MII), zygotes, 2-cells, 4-cells, 8-cells, and morula. Then, we selected the union of the following 12 groups of REEs as the set of 107 edits: (1) for REEs identified in oocytes (GV), we selected those that could not be detected in any oocytes (GV) from elder mothers from GSE95477[39]; (2) for REEs identified in oocytes (MII), we selected those that could not be detected in any oocytes (MII) from elder mothers from GSE95477[39]; (3) for REEs identified in zygotes, we selected those that could not be detected in any androgenetic (AG) zygotes from GSE133854[38]; (4) for REEs identified in zygotes, we selected those that could not be detected in any parthenogenetic (PG) zygotes from GSE133854[38]; (5) for REEs identified in 2-cells, we selected those that could not be detected in any AG 2-cells from GSE133854[38]; (6) for REEs identified in 2-cells, we selected those that could not be detected in any PG 2-cells from GSE133854[38]; (7) for REEs identified in 4-cells, we selected those that could not be detected in any AG 4-cells from GSE133854[38]; (8) for REEs identified in 4-cells, we selected those that could not be detected in any PG 4-cells from GSE133854[38]; (9) for REEs identified in 8-cells, we selected those that could not be detected in any AG 8-cells from GSE133854[38]; (10) for REEs identified in 8-cells, we selected those that could not be detected in any PG 8-cells from GSE133854[38]; (11) for REEs identified in morulae, we selected those that could not be detected in any AG morulae from GSE133854[38]; and (12) for REEs identified in morulae, we selected those that could not be detected in any PG morulae from GSE133854[38].

**Gene-level enrichment analysis.** For gene ontology term enrichment analysis, we used the "enrichGO" function in clusterProfiler[71] with the org.Hs.eg.db database[72] to analyze enriched terms for each type of genes in each stage. To correct for multiple hypothesis testing, we pooled all enrichment results and adjusted them using the Benjamini–Hochberg method.

**Annotation of MBSs and effects of REEs on them.** We intersected the predictions of TargetScan[73] (version 7.0) and miRanda[74] (version 1.9) to annotate MBSs in 3′-UTRs. For the multi-species-alignment-and-seed-region-based predictor TargetScan, we used its own miRNA family info (http://www.targetscan.org/vert_80/vert_80_data_download/miR_Family_Info.txt.zip) and picked only those human ones that are highly conserved (i.e., with "Family Conservation" being 2; see https://www.targetscan.org/faqs.Release_7.html), and the multi-species 3′-UTR input for each chromosome was generated by subsetting the UCSC 30-way alignment in MAF format (http://hgdownload.soe.ucsc.edu/goldenPath/hg38/multiz30way/) with the "interval_maf_to_merged_fasta.py" script from Galaxy tools[75] [https://github.com/galaxyproject/tools-iucand https://github.com/galaxyproject/galaxy(release 21.01)] and the BED file describing the 3′-UTRs for that chromosome. For the full-mature-sequence-based predictor miRanda, we used the mature miRNA sequence accompanied in the miRNA family info downloaded above, and the human 3′-UTR transcript sequences from the 3′-UTR alignment used by TargetScan.

Both TargetScan and miRanda were used with default parameters. During the intersection, we noted that the predicted MBS's were defined at different levels for these two tools (Supplementary Fig. 43). For TargetScan, its predicted MBS is an alignment of a given miRNA family, denoted by its seed region sequence (i.e., the 2–8 nucleotides on the mature miRNA sequence), onto the given 3′-UTR sequence. For miRanda, its predicted MBS is an alignment of a given mature miRNA sequence onto the given 3′-UTR sequence. Because multiple mature miRNA sequences can share the same seed region sequence (and thus belong to the same miRNA family), we need to take the intersection at the miRNA family level. Therefore, we collapsed the miRanda predictions to the miRNA family level before taking the intersection. Specifically, we collapsed into a single prediction all those miRanda predictions that share (1) the same miRNA family, (2) the same target 3′-UTR sequence, (3) the same seed region site type (as specified by TargetScan), and (4) the same start and end positions on the 3′-UTR sequence the seed region aligns to (Supplementary Fig. 44). miRanda predictions that do not share all of these four properties were considered different predictions. As required by TargetScan, during the computation of the site type, we only considered exact matches (i.e., A-U/T and C-G), and wobble pairs (e.g., G-U/T) were excluded. We then took the intersection of TargetScan predictions and the collapsed miRanda predictions. Similar to the collapsing pipeline above, an MBS was considered in this intersection (i.e., shared by both tools), if its TargetScan prediction and miRanda prediction share all the four properties above.

To annotate the effect of each REE on MBSs, we first predicted the MBS's on the edited transcript sequences. Specifically, we modified the multi-species 3′-UTR input for TargetScan (or the edited 3′-UTR sequences for miRanda) by replacing the adenine at the REE site in the human 3′-UTR sequence with guanine, and fed this modified multi-species 3′-UTR input to TargetScan (or the modified 3′-UTR sequence to miRanda) again; in this way, one modified multi-species 3′-UTR input for TargetScan (and one modified 3′-UTR sequence input for miRanda) was generated for each pair of (REE, transcript).

We then annotated an REE on a given combination of gene and microRNA family as follows: (1) the REE was annotated as "no overlaps" if, for each of all transcripts of the gene locus, the REE did not fall into any preexisting MBS, nor would it introduce any new MBS; (2) otherwise, the REE was annotated as "site unchanged" if, for each of all transcripts of the gene locus, the number of new MBSs of the microRNA family that it introduced was equal to the number of preexisting MBSs of the microRNA family that it removed; (3) otherwise, the REE was annotated as "MBS-gaining" / "site gained" / "MBS gain" if both of the following two conditions were satisfied: for each of all transcripts of the gene locus, the number of new MBSs of the microRNA family that it introduced was no less than the number of preexisting MBSs of the microRNA family that it removed, and for at least one transcript of the gene locus, the number of new MBSs of the microRNA family that it introduced was strictly greater than the number of preexisting MBSs of the microRNA family that it removed; (4) otherwise, the REE was annotated as "MBS losing" / "site lost" / "MBS lost" if both the following two conditions were satisfied: for each of all transcripts of the gene locus, the number of new MBSs of the microRNA family that it introduced was no greater than the number of preexisting MBSs of the microRNA family that it removed, and for at least one transcript of the gene locus, the number of new MBSs of the microRNA family that it introduced was strictly smaller than the number of preexisting MBSs of the microRNA family that it removed; (5) otherwise, the REE was annotated as "mixed", where it is deemed to satisfy both of the following two conditions: for at least one transcript of the gene locus, the number of new MBSs of the microRNA family that it introduced was strictly greater than the number of preexisting MBSs of the microRNA family that it removed, and for at least one transcript of the gene locus, the number of new MBSs of the microRNA family that it introduced was strictly smaller than the number of preexisting MBSs of the microRNA family that it removed.

**Annotation of maternal genes and targets of maternal mRNA clearance.** We used STAR[76] to align the trimmed reads from the adapted RNA identification pipeline onto hg38 and then StringTie[77] to estimate the expression level of each gene. We then defined maternal genes as those with median FPKM >2 in at least one of the oocyte (GV) and oocyte [metaphase of second meiosis (MII)] stages. Finally, we annotated a maternal gene as a target of maternal mRNA clearance ("decay at 8-cell" in Supplementary Figs. 30, 38b and Fig. 5d) if the smaller median FPKM value between the oocyte (GV) and oocyte (MII) values was more than twice the median FPKM in the 8-cell stage. All other maternal genes that did not meet this criteria were considered as "others" in Supplementary Figs. 30, 38b and Fig. 5d. All normal samples from oocyte (GV), oocyte (MII), and 8-cell were considered.

**Statistics and reproducibility.** All statistical tests are Wilcoxon's rank-sum test and all adjusted $p$ values have been adjusted by the Benjamini–Hochberg method, unless otherwise specified. The ranges of sample sizes in each figure (where relevant) are available in their figure legends.

**Ethics information of datasets used in this study.** Here we reiterate the ethics information of datasets used in this study (Supplementary Data 1). The study by Yan et al. (data available in NCBI GEO with the identifier GSE36552)[78] was

approved by the Reproductive Study Ethics Committee of Peking University Third Hospital (Research License 2011S2003 and 2011S2018) and was informed consent acquired. The study by Xue et al. (data available in NCBI GEO with the identifier GSE44183)[79] was approved by the Institutional Review Board (IRB) on Human Subject Research and Ethics Committee in the First Affiliated Hospital to Nanjing Medical University, China, and was informed consent acquired. The study by Guo et al. (data available in NCBI GEO with the identifier GSE49828)[80] was approved by the Reproductive Study Ethics Committee of Peking University Third Hospital (Research license 2012SZ015), and was informed consent acquired. The study by Yanez et al. (data available in NCBI GEO with the identifier GSE65481)[81] was approved by the Stanford University Institutional Review Board, and was informed consent acquired. The study by Dang et al. (data available in NCBI GEO with the identifier GSE71318)[82] was approved by the Reproductive Study Ethics Committee of Peking University Third Hospital (Research license 2012SZ015) and was informed consent acquired. The study by Hendrickson et al. (data available in NCBI GEO with the identifier GSE72379)[83] was approved by Institutional Review Board and was informed consent acquired. The study by Reyes et al. (data available in NCBI GEO with the identifier GSE95477)[39] was approved by the Western Institutional Review Board (IRB#1151520) and was informed consent acquired. The study by Fogarty et al. (data available in NCBI GEO with the identifier GSE100118)[84] was approved by the UK Human Fertilisation and Embryology Authority (HFEA) and was informed consent acquired. The study by Wu et al. (data available in NCBI GEO with the identifier GSE101571)[85] was approved by the Institutional Review Board (IRB) of The First Affiliated Hospital of Zhengzhou University (2015KY-NO.31) and Tsinghua University (20170009), China, and was informed consent acquired. The study by Lv et al. (data available in NCBI GEO with the identifier GSE125616)[86] was approved by the Institutional Review Board (IRB) of Tongji Hospital in Tongji University (KYSB-2017-072) and was informed consent acquired. The study by Wamaitha et al. (data available in NCBI GEO with the identifier GSE126488)[87] was approved by UK Human Fertilization and Embryo Authority (HFEA) (with License number R0162) and the Health Research Authority's Cambridge Central Research Ethics Committee, IRAS project ID 200284 (Cambridge Central reference number 16/EE/0067), and was informed consent acquired. The study by West et al. (data available in NCBI GEO with the identifier GSE130289)[88] was approved by the Western Institutional Review Board (study no. 1179872) and was informed consent acquired. The study by Leng et al. (data available in NCBI GEO with the identifier GSE133854)[38] was approved by the ethical committee of the Reproductive & Genetic Hospital of CITIC-XIANGYA (Research license LL-SC-SG-2013-012) and was informed consent acquired. The study by Xiang et al. (data available in NCBI GEO with the identifier GSE136447)[89] was approved by the Medicine Ethics Committee of The First People's Hospital of Yunnan Province (2017LS[K]NO.035) and was informed consent acquired.

For Cacchiarelli et al. (data available in NCBI GEO with the identifier GSE62772)[90], Szabo et al. (data available in NCBI GEO with the identifier GSE64417)[91], Choi et al. (data available in NCBI GEO with the identifier GSE73211)[92], and Lau et al. (data available in NCBI GEO with the identifier GSE119324)[93], we only used RNA-Seq data of human embryonic stem cells from them, where ethics information is not available for such cells.

## Data availability

NCBI GEO accessions of all raw sequencing datasets used in this manuscript are available from their original publications (see Supplementary Data 1 for the full list of accession codes), and the compiled editome (and some intermediate results) is available from Zenodo, with link: https://zenodo.org/record/665852[94]. The Supplementary IGV data is available from Zenodo, with link: https://doi.org/10.5281/zenodo.7379397[41]. Source dataset related to the main figures are available as Supplementary Data 13–33.

## Code availability

Codes for reproducing the results reported in this article are available from the GitHub repository, link: https://github.com/gao-lab/HERE. These codes are also available from the Zenodo repository, link: https://zenodo.org/record/7386496[95].

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

## Acknowledgements
The authors wish to thank Qiya Huang, Longteng Wang, Dr. Cheng Li, and Zi-Yu Chen for their helpful feedback and discussions. The analysis was supported by the High-performance Computing Platform of Peking University. The authors would like to thank the developers of Snakemake for this excellent workflow management tool. This work was supported by the Beijing Natural Science Foundation [http://kw.beijing.gov.cn/; 5204040 to H.L.], the Beijing Nova Program of Science and Technology [https://mis.kw.beijing.gov.cn; Z191100001119064 to H.C.], the National Key Research and Development Program [2016YFC0901603 to G.G.], the China 863 Program [2015AA020108 to G.G.], the National Natural Science Foundation of China [http://www.nsfc.gov.cn; 81973244 to H.L., 31801112 to H.C., and 61873276 to X.B.], and the State Key Laboratory of Protein and Plant Gene Research and the Beijing Advanced Innovation Center for Genomics (ICG) at Peking University [to G.G.]. The research of G.G. was supported in part by the National Program for Support of Top-notch Young Professionals. Funding for open access charge: the Beijing Natural Science Foundation [http://kw.beijing.gov.cn/; 5204040 to H.L.].

## Author contributions

Conceptualization: H.C. and X.B. Study coordination and supervision: G.G., H.C., and X.B. Data curation: Y.D., J.W., H.T., Y.L., and K.X. Computational analyses: Y.D., H.L., C.Z., and X.H. Project management: G.G., H.C., and X.B. Writing: Y.D., Y.Z., J.W., H.L., C.Z., G.G., H.C., and X.B. All authors reviewed and edited the manuscript.

## Competing interests

The authors declare no competing interests.
