## [Peer Review File · Communications Biology]

Recurrent RNA edits in human preimplantation potentially enhance maternal mRNA clearanceREVIEWER COMMENTS

Reviewer #1 (Remarks to the Author: Overall significance):

In this study the authors performed integrated analysis for large number of published human embryo RNA-seq datasets, including both normal and abnormal embryos, to systematically analyze the A-to-I editome for human early embryonic development. They proposed a regulation model which REs enhance maternal mRNA clearance by introducing more microRNA binding sites to the 3'-untranslated regions of clearance targets. In general the biological question is of potential important and interesting. However, there are some questions need to be addressed.

Reviewer #1 (Remarks to the Author: Strength of the claims):

1.As the author claimed they identified systematic A-to-I editome profile by analyzing the variant sites in RNA-seq. Theoretically, it is necessary to show the distribution of editing level of all variant sites. A global view is important.Also it is better to clearly describe the cutoff they used to distinguish the identified A-to-I editing sites and background noise.

2.In the results of Fig2A, the author only showed the overlap of editing sites in normal and other samples. However the identification of “normal” and “other” is unclear, which should be explained, at least mentioned in legends. More detailed analysis in this part would be preferred. For instance, what is the genome enrichment, gene and location (3'UTR, 5'UTR) preference for the overlap sites and the unique sites.

3. It is better to show more detail on the indicated important REs. For instance, the REs loss and the RNA-seq IGV trace on TTF1, which they mention in embryos with uniparental disomy and those from elder mothers. More IGVs of REs on important genes should be present.

4.Here they used Suv39h2 as the example of RE-induced MBS-gain genes, however they did not show the editing level of this REs and the signal change of RNA-seq is also needed, as it will be a supporting evidence of the RE-induced maternal mRNA-clearance model

5.In this manuscript the causal relation between A-I editing induced MBS and maternal RNA clearance was not logically convincing, as they only showed a week difference of MBS-gaining REs amount between 8-cell decay and undecay genes. Normally a causal relation need to be proved by the rescue experiments. More importantly the analysis of RNA level and expression pattern was barely appeared in this manuscript, which make it difficult to believe the RE-induced maternal decay model. I can understand that the batch effect and individual difference could impact the calculation of FPKM of individual genes. Here they analyzed GSE95477 which REs appeared different in young and old mother, and therefore they can further compare the impact of REs on RNA decay in this two group of embryos. For example for the genes loss REs in a particular stage, will the RNA abundance increased in this stage or appeared deficient decay in a later stage?

6.If the RE induced MBS enhanced the decay of maternal RNA, could the author show the time course of RE gain or loss and RNA abundance change? Also if this model worked, will the editing level of REs on maternal decay genes decreased upon the clearance of the edited transcripts. Could the RE induced MBSs related with the regulation of translation by miRNA?

7.As they show REs existed from MII oocyte to 2-cell stage, however if the maternal decay occurred

in 8-cell stage, why the transcripts with RE induced MBS keep stable before 8-cell stage? And how they would explain the role of RE induced MBSs in un decayed genes(Fig 6d). Here the author should explain how they identified the maternal decay genes and what is the identification of “others”

8. Could the proposed model be conservative between species? Is it possible to find conserved maternal transcripts with RE induced MBS in mice?

Reviewer #1 (Remarks to the Author: Reproducibility):

1. Comparison of the number of editing sites between this study and previous reports should be considered as quality control.
2. In Fig2a, only 23% sites were shared between normal and abnormal samples, what is the potential biological explanation?
3. Does the number in Fig2b represent intersect or union number of sites from different replicates of the same sample? Also, number of reproducible sites from different replicates was expected as quality control.
4. P7.I149 seems should be “median” instead of “mean”.

Reviewer #2 (Remarks to the Author: Overall significance):

By curating and analyzing the largest human embryonic editome to date, Ding and coauthors showed that the human preimplantation embryos at various embryonic stages harbour thousands of REs that are preferably exonic and highly shared between stages at the editing site. They also proposed that these REs could potentially enhance maternal mRNA clearance, a process that is crucial for successful maternal-to-zygotic transition, by introducing more MBSs to clearance targets than to other maternal genes.

Reviewer #2 (Remarks to the Author: Impact):

This study introduced the first large-scale A-to-I RNA editome for early human embryos, the analysis of which revealed a consistent early-stage editing pattern (of REs) with probable functional importance in microRNA-based maternal mRNA clearance. The editome itself is a valuable resource for further examination of the interplay between maternal RNAs and early embryo development. However, many of the conclusions need to be experimentally confirmed. And some key hypotheses also need to be examined by bench works.

Reviewer #2 (Remarks to the Author: Strength of the claims):

Specific comments:

The title of “Large-scale identification of recurrent RNA edits in human embryos” is unclear. While the authors only investigated A-to-I RNA editing in human preimplantation embryos, it is better for them to be more specific in defining the scope of research in the title.

The reader would benefit from a more detailed description of A-to-I RNA editing (i.e., frequency, sequence preference, and distribution in the 5' and 3' untranslated region, coding sequences of mRNAs) and the biochemical pathways underlying this edit (i.e., enzymes and RNA-binding proteins that mediate this posttranscriptional modification) in the Introduction section.

The authors should be sure to define abbreviations in figure legends and in the main text and minimize the number of abbreviations. As the manuscript is now, it is difficult to follow the flow on the story due to so many abbreviations.

The authors showed that A-to-I RNA editing could potentially enhance maternal mRNA clearance, by introducing more MBSs to clearance targets than to other maternal genes. However, the role of an microRNA-based regulatory mechanism in maternal mRNA clearance has not been confirmed in mammalian early embryos. Therefore it is reluctant to explain the association of A-to-I RNA editing with maternal mRNA decay using this mechanism. Potential involvement of A-to-I RNA editing in PAN2-PAN3, CCR4-NOT, and exosome mediated RNA clearance pathways should be considered.

The occurrence and functions of A-to-I RNA editing in human preimplantation embryos are completely based on high throughput data analyses. The results of analyses are no doubt valuable, but it is equally important to experimentally conform the location of A-to-I RNA editing on some representative maternal transcripts, and provide experimental results that support the hypotheses that A-to-I RNA editing indeed affect the dynamics of maternal mRNA clearance.

In Figure 7, there is an error about the dynamics of maternal mRNA clearance in human embryos: the major wave of maternal mRNA clearance occur at the 8 cell stage, not after blastocyst formation as is shown in the left conner of the proposed model.

Authors need to follow the guidelines of gene symbols, protein symbols, RNA symbols. Based on our understanding gene and RNA symbols should be italicized.

Reviewer #3 (Remarks to the Author: Overall significance):

In the submitted manuscript, the authors reported their analysis of A-to-I RNA editing in human embryos. They found that editing is diminished in abnormal embryos and that it can introduce microRNA binding sites in 3'UTR of maternal transcripts to facilitate their clearance. The topic of the study is interesting and might be clinically relevant to reproduction and fertility. Nevertheless, I think the work can be further improved.

Reviewer #3 (Remarks to the Author: Impact):

Communications Biology would be most appropriate for the manuscript.

Reviewer #3 (Remarks to the Author: Strength of the claims):

Major comments:

- 1) The authors should elaborate more about their analysis steps in the main text. Details of any additional filters applied (besides SNP removal) are missing from Figure 1b. As written, there seems to be nothing special about the authors' computational pipeline.
- 2) Figure 1 should include the distribution of all 12 possible types of nucleotide changes (A-to-C, A-to-G, A-to-T etc) after variant calling with or without application of additional filters.
- 3) Figure 1D: How was the false positive (FP) rate calculated? Please describe in main text.
- 4) The A-to-G percentage should be given for non-Alu and Alu sites separately.
- 5) The authors should calculate the Alu editing index (AEI) for each sample. Does the extent of editing correlate with ADAR expression levels?
- 6) Figure 1 & 2 can be combined.
- 7) What is the number of samples for each developmental stage (Figure 3b)?
- 8) Figure 3c,d: It would be useful to show similar graphs for later developmental stages (morula and afterwards) as well. Is the preponderance of sites in 3'UTR observed only in the earlier developmental stages?
- 9) Genes with recurrent edits (REs) in early stages - do their expression levels go down in later developmental stages (morula and afterwards), indicative of transcript degradation?
- 10) The authors wrote that "An initial scan revealed 107 edits on 76 genes that were REs in normal embryos, but completely lost in the same stage in pathological embryos and embryos from elder mothers." What is the initial scan? Please provide more details on how the 107 edits on 76 genes were shortlisted. For each site, was there sufficient sequencing coverage in the abnormal embryos? What were the editing levels of these 107 sites in the normal embryos?
- 11) Figure 5a,c: What are AG embryos and PG embryos? Also, what is "BI"? Please explain in the figure legend.
- 12) I see "amanitin" in Figure 5c. Amanitins are compounds that block RNA polymerase II, thereby inhibiting transcription. How is this drug treatment relevant to the manuscript?
- 13) Figure 5c: Data from GSE133854 don't look convincing. This seems to invalidate the authors' claim that there are "fewer RE-matching edits in protein-coding genes of abnormal embryos and embryos from elder mothers".
- 14) MicroRNA prediction programs are often inaccurate. Instead of relying solely on TargetScan, can the authors use at least two different software and take the intersection?

15) Figure 6d: Where did the authors obtain the genes that "decay at 8-cell" from? I don't see any gene expression analysis. (Similar to comment #9 above.)

Minor comments:

16) In the introduction, the authors wrote "The successful development of human embryos is based on a well-regulated network that spans multiple omic layers." It is unclear what "multiple omic layers" mean.

17) Figure 3a is redundant.

18) At the end of the results section, the authors wrote that "This hypothesis was immediately validated by the observation that ..." I suggest replacing "immediately validated" with "supported". The authors did not perform any independent experimental validations.

19) In the discussion, the authors wrote that "The role of A-to-I RNA editing in human has been ambiguous for a long time." This is not true. Multiple papers have clarified the diverse roles of editing in human.

20) In the methods section, the authors wrote "For single-cell RNA-Seq datasets, we required that the sequencing technology not be based on cell barcoding." Can the authors explain why?

21) One of the supplementary datasets (the largest one) appears to be filled with nonsensical characters.

22) There are scattered English language errors. Please proofread.

Reviewer #3 (Remarks to the Author: Reproducibility):

Please see comments under "Strength of the claims", for example, comments #13 and #14.

Response to reviewers' comments

We thank the reviewers for highlighting the quality of our work and for their valuable comments, and have revised the manuscript accordingly. Specifically,

1. We supported the association between REE ("Recurrent Embryonic Edit"; formerly the RE ("Recurrent Edit")) and maternal mRNA clearance with evidence at the level of gene expression (Supplementary Fig. 8, 16, 22-23, and Supplementary Notes 4).
2. We validated most of our conclusions related to REE-related microRNA binding sites (MBS) using the intersection of predictions by both TargetScan and miRanda (Fig. 5b-d, Supplementary Fig. 14-15) with detailed methods (Methods; Supplementary Fig. 18-19).
3. We detailed the background of RNA editing, the methods necessary for identifying the edits (Supplementary Notes 6, Supplementary Tables 1, 7-8, and Supplementary Fig. 5), additional description of these edits (Supplementary Notes 2, Supplementary Fig. 1, 2, 4, 6, 9-11, 17, 20-21, and Supplementary Data 1), and the discussion of the potential relevance of these edits to maternal mRNA clearance (Supplementary Fig. 24-25) as suggested by reviewers.

All the changes (except for the replacement of "RE" with "REE") in the manuscript and supplementary material are highlighted in blue font. Please see the replies to the specific reviewer comments below for further details. We thank again for the editors and reviewers' efforts.

Reviewer #1 comments

Remarks of the Author: Overall Significance

In this study the authors performed integrated analysis for a large number of published human embryo RNA-seq datasets, including both normal and abnormal embryos, to systematically analyze the A-to-I editome for human early embryonic development. They proposed a regulation model in which REs enhance maternal mRNA clearance by introducing more microRNA binding sites to the 3'- untranslated regions of clearance targets. In general the biological question is of potential important and interesting. However, there are some questions that need to be addressed.

We thank for the reviewer's evaluation and the comments below that help to strengthen our study.

Remarks to the Author: Strength of the claims

1. As the author claimed they identified systematic A-to-I editome profile by analyzing the variant sites in RNA-seq. Theoretically, it is necessary to show the distribution of editing level of all variant sites. A global view is important.

We thank the reviewer for pointing out the lack of an analysis at the global level, and we agree that such distribution is necessary.

We have added the distribution of editing level of all variant sites (across all samples) as Supplementary Fig. 4 (also the Additional Fig. 1 below) (Page 4, Lines 21-22 in main text; Pages 18- 20 in supplementary material).

Additional Figure 1. Distribution of editing level in each sample group. Note that although we have used a very stringent pipeline to remove as many false positives as possible (see Methods), we still found a number of editing sites with an editing level of 1. These editing sites should be examined with caution.

Also it is better to clearly describe the cutoff they used to distinguish the identified A-to-I editing sites and background noise.

We thank the reviewer for pointing out the lack of clarity of methodological details in our methods.

The "cutoff" we used to distinguish the identified A-to-I editing sites and background noise is a combination of multiple filters detailed below, as described earlier in the previous Supplementary Notes 1 (Supplementary Notes 6 in the revision). We found that we did not explicitly state the order by which we applied these filters, so we reorganized this part as below and updated the Supplementary Notes 6 as shown below (Pages 6-12 in supplementary material) and referred the readers to this in the main text (Page 12, Lines 16-17):

Steps (1-9) for first-round per-sample filtering (with relevant Perl script in the GitHub repository marked when available):

- (https://github.com/gao-lab/HERE/blob/master/scripts/S15_1____get_sample_RNA_editing_sites_v3/step07____apply_complex_filter/complex_filter_1/Convert_VCF.pl) Starting from the GATK-called VCF file, we filtered variant sites ('sites' for short) for those on autosomes, chromosome X, or chromosome Y with the FILTER being "PASS" and at least one read covered.
- (https://github.com/gao-lab/HERE/blob/master/scripts/S15_1____get_sample_RNA_editing_sites_v3/step07____apply_complex_filter/complex_filter_1/ref_filter.pl) We filtered for sites that had not been labeled as a dbSNP variant during GATK variant calling, or had been labeled as the cDNA molecular type as annotated by UCSC Genome Browser dbSNP version 151 (<http://genome.ucsc.edu/cgi-bin/hgTrackUi?db=hg38&g=snp151Flagged>).
- (https://github.com/gao-lab/HERE/blob/master/scripts/S15_1____get_sample_RNA_editing_sites_v3/step07____apply_complex_filter/complex_filter_1/threads_rmMismatch.pl and https://github.com/gao-lab/HERE/blob/master/scripts/S15_1____get_sample_RNA_editing_sites_v3/step07____apply_complex_filter/complex_filter_1/Remove_mismatch_first6bp.pl) For each site, we re-counted the number of mismatch reads on this sites based on all realigned reads (as produced by GATK IndelRealigner) whose mismatched base at this site was located outside the first 5- prime 6 bp of the read and had a base quality (as recorded in the realigned bam file) no less than 58. If no reads with mismatched bases were available for a given site after re-counting, this site was discarded. In addition, we discarded sites with two or more different types of mismatch bases (i.e., sites that are potential polymorphisms).

- (https://github.com/gao-lab/HERE/blob/master/scripts/S15_1____get_sample_RNA_editing_sites_v3/step07____apply_complex_filter/complex_filter_1/Alu_filter.pl) We then separated the sites into those on and not on Alu elements, as annotated by the UCSC Genome Browser RepeatMasker track (<http://genome.ucsc.edu/cgi-bin/hgTrackUi?db=hg38&g=rmsk>); the resulting subsets were named "Alu subset" and "other subset", respectively. No sites were discarded in this step.
- (https://github.com/gao-lab/HERE/blob/master/scripts/S15_1____get_sample_RNA_editing_sites_v3/step07____apply_complex_filter/complex_filter_1/FreqSimple.pl) We picked the "other subset" and filtered for sites that both (a) had a mismatch frequency > 0.1, and (b) did not overlap with any simple repeats described by the simple repeats annotated by the UCSC Genome Browser RepeatMasker track.
- (https://github.com/gao-lab/HERE/blob/master/scripts/S15_1____get_sample_RNA_editing_sites_v3/step07____apply_complex_filter/complex_filter_1/threads_rmSJandHomo.pl, https://github.com/gao-lab/HERE/blob/master/scripts/S15_1____get_sample_RNA_editing_sites_v3/step07____apply_complex_filter/complex_filter_1/RemoveHomoNucleotides.pl, and https://github.com/gao-lab/HERE/blob/master/scripts/S15_1____get_sample_RNA_editing_sites_v3/step07____apply_complex_filter/complex_filter_1/Filter_intron_near_splicejunctions.pl) We picked the output of Step (5), and filtered for sites that both (a) was not located within 4bp intronic regions of splicing sites defined by the UCSC Genome Browser knownGenes track (<http://genome.ucsc.edu/cgi-bin/hgTrackUi?db=hg38&g=wgEncodeGencodeV32>), and (b) was not located within a homopolymer >= 5bp (i.e., a >= 5bp sequence segment consisting of identical nucleotides).
- (https://github.com/gao-lab/HERE/blob/master/scripts/S15_1____get_sample_RNA_editing_sites_v3/step07____apply_complex_filter/complex_filter_1/threads_BlatCandidates.pl and https://github.com/gao-lab/HERE/blob/master/scripts/S15_1____get_sample_RNA_editing_sites_v3/step07____apply_complex_filter/complex_filter_1/BLAT_candidates.pl) We picked the output of Step (6), and for each its site we (a) extracted all those realigned reads with the mismatched base for this site (as produced by GATK IndelRealigner), (b) aligned them to the entire genome with blat (with the following parameters: -stepSize=20 -repMatch=2253 -minScore=20 -minIdentity=0 -noHead) as suggested by Lo Giudice et al. [1], (c) identified for each read whether it can uniquely map to this site again (with its Blat score * 0.95 higher than the Blat score of the second best hit), and (d) kept this site only if the number of reads that can uniquely map to this site is no less than the number of reads that can't. We then update the number of mismatches on this site with the number of reads that can uniquely map to this site.
- (https://github.com/gao-lab/HERE/blob/master/scripts/S15_1____get_sample_RNA_editing_sites_v3/step07____apply_complex_filter/complex_filter_1/nonAlu_filter_new.pl) We picked the output of Step (7), and used UCSC Genome Browser RepeatMasker track to separate these sites into those that were on non-Alu repeat elements, and those that were not on repeat elements. No sites were discarded in this step.

- We used `bcftools concat --allow-overlaps` to combine the sites called on Alu elements from Step (4), sites on non-Alu repeat elements from Step (8), and sites not on repeat elements from Step (8). No sites were discarded in this step.

Step (10)

We used `bcftools isec --collapse all` to identify all those genomic variants from the following worldwide population studies that overlapped with any of the sites discovered in any samples in Steps (1-9).

- the University of Washington Exome Sequencing Project (<http://evs.gs.washington.edu/EVS/>),
- the 1000Genomes Project [2] (http://ftp.1000genomes.ebi.ac.uk/vol1/ftp/data_collections/1000_genomes_project/release/20190312_biallelic_SNV_and_INDEL/),
- all version 2.1.1 exome and genome variants (using the `liftover_hg38` versions) and all version 3.0 genome variants of the gnomAD project [3], and
- the National Center for Biotechnology Information's Allele Frequency Aggregator project (which involves the computation of allele frequencies for variants in the database of Genotypes and Phenotypes across approved un-restricted studies) [4] (https://ftp.ncbi.nih.gov/snp/population_frequency/latest_release/freq.vcf.gz ; downloaded on November 11th, 2020).

We then used `bcftools isec --collapse all` to identify all those variant sites discovered in any samples in Steps (1-9) that overlapped with any of these genomic variants, and discarded these variant sites.

Step (11)

Starting with the variant sites kept in Step (10), we further kept those sites that met either of the following: (a) this site was located in an Alu element, and had ≥ 10 reads covered, and had a $\geq 10\%$ mismatch frequency; (b) this site was not located in an Alu element, and had ≥ 10 reads covered, and had a $\geq 10\%$ mismatch frequency, and had ≥ 3 reads supporting the mismatch.

Step (12)

Starting with the variant sites kept in Step (11), we further kept those sites that met either of the following: (a) this site was located in an Alu element; (b) this site was not located in an Alu element, and was observed in at least two normal samples (or two abnormal samples) of the same stage (see Supplementary Table 8 for the details of each stage).

Step (13)

Starting with the variant sites kept in Step (11) and their SnpEff annotation (run with parameters `snpEff ann -lof` and the GENCODE v32 annotation), we kept those variants annotated with either the A>G (i.e., A-to-G) SnpEff event only, or both the A>G and the T>C (i.e., T-to-C) SnpEff event.

We deliberately kept the latter type of variant, which appeared in at least two transcripts of different orientations (i.e., a variant reported by SnpEff to induce an A-to-G shift in one transcript and a thymine-to-cytosine shift in another), because a previously well-studied A-to-I RNA editing event found to recode tyrosine to cysteine in the *BLCAP* (bladder cancer-associated protein) gene [5], persist in human embryonic stem cells and multiple human tissues [6], and promote cell proliferation [7] is this type of variants.

For the sake of reproducibility, please elaborate on all Methods. Please note that Nature Portfolio journals do not enforce a word limit on this section, and refer to the Open Research Evaluation at the end of the document for additional guidelines.

We thank for the editor's suggestions and have added the corresponding details necessary for the reproducibility (see reply to the above).

2. In the results of Fig2A, the author only showed the overlap of editing sites in normal and other samples. However the identification of "normal" and "other" is unclear, which should be explained, at least mentioned in legends. More detailed analysis in this part would be preferred. For instance, what is the genome enrichment, gene and location (3'UTR, 5'UTR) preference for the overlap sites and the unique sites.

This point would be necessary for further consideration at *Communications Biology*.

We thank the reviewer and the editor for pointing out the lack of clarity in usage of these terms.

We have added to Fig. 2a (now renumbered as Fig. 1e)'s legend an explanation of what the "normal" and "other" (which we believed to be "abnormal") sites stand for (Page 21, Lines 6-9 in main text). Briefly, the "normal" sites are those sites identified from normal, healthy samples, while "abnormal" sites are those identified from pathological samples (e.g., samples undergoing Uniparental Disomy (UPD) from Dataset GSE133854 [8]), or samples with non-control treatment (e.g., treated with amanitin as in Dataset GSE101571 [9]). Details of whether each sample is "normal" or "abnormal" are documented in the column "is.normal" in the phenotype table accompanied with the manuscript ("TRUE" for "normal" samples and "FALSE" for "abnormal"; see Supplementary Table 7; also available from <https://github.com/gao-lab/HERE/blob/master/report.ver2/210215-sixth-dataset/201221-fifth-phenotype-collection/sample.stat/all.2071.samples.csv>).

We have also examined the genome enrichment between these two sets of editing sites, gene and location (3'UTR, 5'UTR) preference for the overlap sites and the unique sites (which we assume to be normal- or abnormal-exclusive sites) (Page 21, Lines 10-11 in main text; Pages 13-15 in

supplementary material) (Additional Fig. 2a and Supplementary Fig. 2a). It appears that the overlap sites ("both" in the figure) are more 3'-UTR/intron-enriched than the unique sites ("normal only" and "abnormal only") (Additional Fig. 2b and Supplementary Fig. 2b). In particular, this enrichment has a clear statistical significance in most stages (with the chi-square test p -values being less than $2.2e-16$ for most combinations listed in Additional Fig. 2b). Whether this is of biological significance or mostly due to technical artifacts and individual variability remains to be investigated.

a

b

Additional Figure 2. Genomic distribution of edits from normal and/or abnormal samples per stage (a) and the deviation of 3'-UTR-or-intron edit ratio of the overlap edits from that of the unique edits (b). Only stages with both types of samples were shown. All p -values were from chi-square test and BH-adjusted.

3. It is better to show more detail on the indicated important ERs. For instance, the REs loss and the RNA-seq IGV trace on TTF1, which they mention in embryos with uniparental disomy and those from elder mothers. More IGVs of ERs on important genes should be present.

We are grateful for this helpful comments, as examining case loci with IGV has greatly helped the test of validity of our codes and potential pitfalls of our claims, which we described below.

We made visual inspection of the *TTF1* recoding REE-matching edit (chr9:132,375,956) on all PG zygotes and identified for sample GSM3928566 that this region was missed by our pipeline due to insufficient read coverage (8 compared with the threshold 10 in our pipeline) despite a large count of mismatch-supporting reads, and led to a putative false negative call (Additional Fig. 3 and also Supplementary Fig. 17). This case was thus not solidly supported, and we moved it to the Discussion where we discussed one of the limitations of this study: the false negatives due to insufficient sequencing depth (Page 10, Lines 17-25 in main text).

Additional Figure 3. IGV tracks showing read alignment at the *TTF1* recoding edit (chr9:132,375,956) in the PG zygote GSM3928566 from the dataset GSE133854.

As an alternative, we added the IGV plots of the REE-matching edit chr8:28,190,741 on *ELP3* on all normal and PG zygotes; 3 out of all 4 PG zygotes have many reads covered at this site with so little editing that they failed to pass our identification pipeline (Supplementary Data 1).

We thank again for this helpful comment and added this to Results (Page 6, Lines 13-16 in main text; Page 47, Lines 20-21 in supplementary material).

4. Here they used *Suv39h2* as the example of RE-induced MBS-gain genes, however they did not show the editing level of this REs and the signal change of RNA-seq is also needed, as it will be a supporting evidence of the RE-induced maternal mRNA-clearance model.

We thank the reviewer for this critical suggestion (and also the suggestions in Comments #5 and #6 below) on supporting the relationship among REE, REE-induced MBS, and maternal mRNA clearance with the key evidence at the level of expression change.

Upon double-checking the distribution of REEs across stages, we found that most of them were from those stages (i.e., oocyte (MI), zygote (2PN), 2-cell (3PN), and 4-cell (3PN)) that have not been included in REE analysis (see the renumbered Fig. 2a); in addition, some of these stages have too few samples (<10) to make the REE reliable (see Supplementary Table 8 for their sample sizes). We therefore discarded those REEs and kept only three REEs, all of which come from oocyte (GV).

We then examined the correlation between their editing level and the FPKM of *SUV39H2* in the same sample, and found that the only MBS-gaining REE, along with one MBS-neutral REE, has a clear negative Spearman's correlation coefficient (Additional Fig. 4), thus supporting the model of REE inducing clearance of maternal mRNA.

Additional Figure 4. Scatterplot for the editing level of MBS-related REE in oocyte (GV) on *SUV39H2* in each sample and the FPKM of *SUV39H2* in the same sample. Note that we only examined for each REE those oocyte (GV) samples where this REE was detected; therefore, points for different REEs correspond to different sample subsets, and their FPKM might not match completely. The trend of association is visualized by linear regression (with line denoting the fitted line and grey shadow denoting the 95% confidence interval).

However, when addressing Comment #14 by Reviewer 3, we found that the REEs from oocytes (GV) on this gene were not determined to be MBS-gaining any more (as excluded by miRanda). Therefore, we decided to move the old Fig. 6e, along with the Additional Fig. 4, into Supplementary Notes (see Supplementary Notes 4 for more details) (Page 8, Lines 5-6 in main text; Pages 4, 43 in supplementary material).

5. In this manuscript the causal relation between A-I editing induced MBS and maternal RNA clearance was not logically convincing, as they only showed a week difference of MBS-gaining REs amount between 8-cell decay and undecay genes. Normally a causal relation need to be proved by the rescue experiments. More importantly the analysis of RNA level and expression pattern was barely appeared in this manuscript, which make it difficult to believe the RE-induced maternal decay model. I can understand that the batch effect and individual difference could impact the calculation of FPKM of individual genes. Here they analyzed GSE95477 which REs appeared different in yang and old mother, and therefore they can further compared the impaction of REs on RNA decay in this two group of embryos. For example for the genes loss REs in a particular stage, will the RNA abundance increased in this stage or appeared deficient decay in a later stage?

Concerns about the biological relevance of RNA editing and maternal RNA clearance prohibit further consideration by *Nature Genetics* and *Nature Communications*. Addressing this point or qualifying these conclusions would be necessary for further consideration at *Communications Biology*.

We thank the reviewer and the editor for pointing out the missing of this key evidence supporting our claim in the old Figure 5e (now renumbered as Figure 4e).

The limited sample size in GSE95477 (5 young and 5 old mothers, each of whom has one oocyte (GV) and one oocyte (MII)) might not be sufficient to rule out potential batch effects in the single cell-based expression profile when analyzing the relationship between FPKM and REE-matching edits in this dataset. Therefore, we recommend addressing this relationship within normal samples, where the sample size is larger. See the response to Comment #6 of the same reviewer (i.e., the next comment) for more details.

For the sake of completeness, below we examined the trend of RNA abundance upon loss of REE(- matching edit)s on GSE95477 samples, consisting of paired oocytes (GV) and oocytes (MII) from these 10 patients. We, however, did not regard any results below as being free from major non- biological artifacts like the batch effects mentioned by the reviewer. After discarding outlier samples of GSE95477 (as described in Supplementary Notes 3), 8 patients (4 young and 4 old) were left with a pair of oocyte (GV) and oocyte (MII) samples, further limiting the solidity of the conclusions derived therein.

Examine the within-sample correlation

We first examined for each maternal gene whether the number of its REE-matching edits in a given sample is inversely correlated with its expression (quantified by FPKM) in the same sample. In accordance with the old Figure 5e (now renumbered as Figure 4e), we computed the correlation for each gene in each sample group of (oocyte (GV)/oocyte (MII), oocytes from young / old mothers). To better estimate the Spearman's correlation coefficient of a given gene in a given sample group, we only examined those genes that satisfy all the followings: (a) the gene has ≥ 1 REE-matching edit identified in ≥ 3 samples of the sample group; (b) the FPKM of this gene is always ≥ 5 across all samples of the stage of the sample group; and (c) the FPKM of this gene must change across samples

in the sample group (otherwise its variance is zero, and the Spearman's correlation cannot be computed).

As shown in Additional Fig. 5, while the correlation is not uniformly negative, we discovered a statistically significant negative shift in correlation coefficients in oocytes (MII) from young mothers compared to oocytes (MII) from old mothers, suggesting that more REEs on a given maternal gene is more likely to lead to lower FPKM in oocytes (MII) from young mothers than in oocytes (MII) from old mothers.

Additional Figure 5. Distribution of Spearman's correlation coefficient between the number of REE-matching edits identified on a gene, and its FPKM in the same sample. The sample groups are consistent with those in Figure 4e. All p -values were unadjusted and were derived from one-tailed Wilcoxon's Rank Sum tests, with the alternative hypothesis being that the median of the correlation from young mothers is smaller than that from old mothers.

Examine the cross-stage correlation

We then examined for each maternal gene whether the number of its REE-matching edits in the oocyte (GV) stage is inversely correlated with its expression (quantified by FPKM) in the oocyte (MII) stage of the paired patient. To better estimate the Spearman's correlation coefficient of a given gene in a given sample group, we only examined those genes that satisfy both the followings: (a) the gene has ≥ 1 REE-matching edit identified in ≥ 6 oocytes (GV); and (b) the FPKM of this gene must change across all oocytes (GV) (otherwise its variance is zero, and the Spearman's correlation cannot be computed).

As shown in Additional Fig. 6, only about half of such genes displayed a negative correlation coefficient, most of which are weak (between -0.5 and 0.0), suggesting that the effect of REE- matching edits on expression level in subsequent stages, if negative, would be mostly weak. We also note that, for targets of maternal clearance (the "decay at 8-cell" genes), the correlation dropped statistically significantly in oocytes of elder mothers. We currently do not have an explanation for this.

Additional Figure 6. Distribution of Spearman's correlation coefficient between the number of REE-matching edits identified on a gene in an oocyte (GV), and its FPKM in the oocyte (MII) of the paired patient. All p -values were unadjusted and were derived from one-tailed Wilcoxon's Rank Sum tests, with the alternative hypothesis being that the median of the correlation from young mothers is larger than that from old mothers.

6. If the RE induced MBS enhanced the decay of maternal RNA, could the author show the time course of RE gain or loss and RNA abundance change? Also if this model worked, will the editing level of REs on maternal decay genes decreased upon the clearance of the edited transcripts. Could the RE induced MBSs related with the regulation of translation by miRNA?

This point would be necessary for further consideration at *Communications Biology*.

Again, we are grateful for the reviewer and the editor for pointing out this key missing evidence at the expression level, and we investigated this extensively, as described below.

We note that a clearance of targeted genes might make it harder to accurately detect editing levels of edits on the remaining transcripts of these targeted genes; therefore, current RNA-seq datasets might not be a decent data for answering such questions. Nevertheless, to answer all these questions quantitatively, we decided to describe REE gains and losses in a more quantitative way by their

editing levels, i.e., an increment in a REE editing level describes its gain, and a decrement describes its loss. We then examined the time course trend as suggested by the reviewer for each [former stage, latter stage] pair of consecutive early stages.

Addressing the first question

We examined whether a higher editing level of a given REE in a given former stage is associated with a lower expression level of the gene hosting the edit (quantified by FPKM here) in the immediate latter stage. However, because samples between consecutive stages could come from different individuals of different datasets, we could not examine this association using paired samples for each gene separately. Therefore, we tried to examine this association across all pairs of [REE, gene hosting this REE] at once, by computing the Spearman's correlation coefficient between the median editing level of a given REE in the former stage, and the median FPKM of the gene hosting this edit in the latter stage.

Additional Fig. 7 (also Supplementary Fig. 8) shows that the associations for all consecutive stage pairs examined were statistically significantly negative, thus supporting the hypothesis that more REEs on a given gene (in the sense of editing level) in a stage could lead to a lower expression level of the same gene in the latter stage. We hoped that this answered the first question in this comment.

We added this part to Results (Page 5, Lines 13-15 in main text; Page 22 in supplementary material).

Additional Figure 7. Spearman's correlation coefficient between median editing level of a REE in a given stage and median FPKM of the gene hosting that REE in the latter stage. Only normal samples from early stages were considered. For each stage pair, only genes with FPKM > 0.001 were plotted. The trend of association is visualized by linear regression (with orange line denoting the fitted line and grey shadow denoting the 95% confidence interval). The p -value shown in each scatterplot is the Benjamini-Hochberg-adjusted p -value for two-sided correlation test (computed by R's `cor.test(method="spearman")`).

Addressing the second question

For the second question, it is currently impossible to determine from RNA-seq data alone the clearance of 'edited' transcripts; we can only tell whether the clearance of all (both edited and unedited) transcripts of a target gene of maternal clearance is accompanied by a drop in editing level of REEs on this gene (in, e.g., the next stage). Specifically, we examined whether the following is true in general for targets of maternal clearance: for a given maternal gene, a drop in its median FPKM upon the transition from the former stage to the latter stage is accompanied by a drop in the editing level of a REE on this gene. As is concerned by the second question, we focused on those edits that are REEs in the former stage; these REEs could be detected as either REE or non-REE in the latter stage (Additional Fig. 8), and we only focused on those that were

detected as REE in the latter stage.

Additional Figure 8. Count of (gene, REE) pairs for REE-to-REE and REE-to-non-REE groups in each early stage transition. Only those (gene, REE) pairs whose gene undergoes a drop in median FPKM upon stage transitions, and whose REE is identified as REE in the former stage, were considered.

Additional Fig. 9 shows that in all stage transitions except for zygote -> 2-cell, a drop in median FPKM will in general lead to an increment (rather than a decrement as suggested by the reviewer) in the editing level of REEs shared between the two stages. A possible explanation is that the addition of these REEs to transcripts is faster than the baseline decay speed in these stage transitions (and relatively slower in zygote -> 2-cell transition). While we currently cannot test this immediately using the given data modalities, an in vitro quantification of the edited and unedited copies of such transcripts involving the *ADAR* enzymes and the RNA degradation mechanisms that have been found to be active in embryonic development (e.g., the CCR4-NOT complex, as suggested by Comment #4 of Reviewer #2) might help validate this. We also note that, when comparing editing level changes between targets of maternal clearance and other maternal genes, we observed a statistically significant smaller increment in median editing level on targets of maternal clearance than on other maternal genes (in oocyte (GV) -> oocyte (MII) and zygote -> 2-cell only), possibly suggesting that targets of maternal clearance undergo a faster decay than do other maternal genes in these stage transitions.

Because our main focus is not on how these REEs were regulated, we did not include this part in our manuscript.

Additional Figure 9. Boxplot of increment in median editing level for (gene, REE) in each early stage transition. Only those (gene, REE) pairs whose gene undergoes a drop in median FPKM upon stage transitions, and is a maternal gene (could be either targets of maternal clearance or other maternal genes), and whose REE is identified as REE in both the former and the latter stages, were considered.

Addressing the third question

For the third question, we returned to the results for the first question, and recomputed the Spearman's correlation coefficients for each of the three groups of REEs separately: REEs whose hosting transcripts are irrelevant to MBS (i.e., no MBS targets were identified in the unedited and edited versions of the transcript at all), REEs that gain MBS, and REEs that do not gain MBS but still allow MBS to be identified. Additional Fig. 10 (also Supplementary Fig. 16) shows that this correlation is

observed for the latter two cases (though without a huge difference between them) but not the first, suggesting that the REE-related decay, among which REE-induced MBS might contribute much, is mostly associated with miRNA-based regulation.

We added this part to Discussion, along with addressing other RNA degradation pathways (e.g., CCR4-NOT; see the reply to Comment #4 by Reviewer #2) (Page 9, Lines 7-23 in main text). Briefly, the possibility of REE affecting other RNA degradation pathways might partially explain why the negative correlations are similar between those target genes whose REE can gain additional MBSs and those whose REEs can't.

Additional Figure 10. Spearman's correlation coefficient between median editing level of a REE in a given stage and median FPKM of the gene hosting that REE in the latter stage, with (gene, REE) pairs grouped by how the REE is related to MBS. Only normal samples from early stages were considered. For each stage pair, only genes with FPKM > 0.001 were plotted. The trend of association is visualized by linear regression (with orange line denoting the fitted line and grey shadow denoting the 95% confidence interval).

7. As they show REs existed from MII oocyte to 2-cell stage, however if the maternal decay occurred in 8-cell stage, why the transcripts with RE induced MBS keep stable before 8-cell stage?

And how they would explain the role of RE induced MBSs in undecayed genes (Fig 6d).

Here the author should explain how they identified the maternal decay genes and what is the identification of "others"

We thank the reviewer for pointing out the confusion arising from our lack of writing clarity.

For the first question, we did not claim that the maternal decay occurred in 8-cell stage; rather, we identified mRNAs undergoing maternal decay by a considerably lower median of FPKM (<50%) at 8-cell stage compared to that in oocytes. These mRNAs might have their expression level dropped in early stages, e.g., 2-cells or 4-cells. We had not focused on when the maternal decay occurred for those transcripts with REE-induced MBS.

For the second question (i.e., the role of REE-induced MBSs in the undecayed genes, which we assume to be the "others" maternal genes in the old Fig. 6d (now renumbered as Fig. 5d)), we speculated that these MBSs are not strong enough (e.g., in the sense that the overall number of MBSs induced on these genes is small compared to that on the targets of maternal clearance, as suggested in the old Fig. 6d) to make these genes decay.

For the third question, we have clarified this by updating how we identified maternal decay genes (and also the "others" genes) in the subsection "Annotation of maternal genes and targets of maternal mRNA clearance" in Methods (Page 17, Lines 20-24 in main text) and all other places where these two terms were used (Page 27, Lines 13-15, Page 29, Lines 8-10 in main text; Page 36, Lines 6-8 in supplementary material).

8. Could the proposed model be conservative between species? Is it possible to find conserved maternal transcripts with RE induced MBS in mice?

If feasible, please comment on this analysis, though it could also be addressed as a future direction in the Discussion, for further consideration at *Communications Biology*.

We thank the reviewer and the editor for pointing out this important and interesting direction for the study of the role of RNA editing across species.

Due to the expansion of editing-prone Alu elements in human, RNA editing in human is so strong compared to that in mouse [10] that the editing profile in adult tissue samples from human and mouse,

even when only conserved editing sites were considered, would group samples by species (human vs. mouse) rather than by tissue type [11], suggesting a mouse-specific editing pattern (and thus a possibly mouse-specific working model of A-to-I edits) in mouse early embryos. Therefore, while mouse embryonic A-to-I edits might contribute to embryonic development, it is not very likely for them to work mostly by strictly following our proposed model. We have updated the discussion above to Discussion (Page 11, Lines 3-11 in main text).

For the second question, it is technically feasible to locate conserved edits between human and mouse (as done in adult tissues by Tan et al. [11]) on maternal transcripts. We then believe that such conserved maternal transcripts, if exist, would suggest a heavily conserved mechanism of A-to-I editing on regulating clearance of maternal transcripts, possibly by inducing MBS. Nevertheless, this direction is out of the scope of our current work; therefore, we added this as a future direction to Discussion (Page 11, Lines 11-14 in main text). We also note that a similar role of RNA editing affecting the stability of translation of edited maternal mRNAs has been proposed, yet to be fully validated, in a previous study of zebrafish A-to-I editing [12]. However, given the primate-specific pattern of the Alu-dominated editing landscape, we currently cannot consider this a conserved functional role of RNA editing in human early embryos.

Remarks to the Author: Reproducibility

1. Comparison of the number of editing sites between this study and previous reports should be considered as quality control.

We thank for this helpful suggestion which helped us to investigate potential effects of technical differences on the identification.

We sectionized the detailed reply to make it clear. The final conclusions are:

1. While there does exist a certain previous study (the Qiu et al. [13]) to compare with, its identification pipeline could not be fully reimplemented even after several rounds of communication with the authors;
2. Based on our best reimplementation results, we made several attempts and can explain in all samples $\geq 75\%$ edits that were identified by Qiu et al. but not us (by our more stringent criteria), and in $\geq 75\%$ samples $\geq 66\%$ edits that were identified by us but not by Qiu et al. (by differences in the choice of sequence aligners in the first-round alignment and realignment);
3. Therefore, our identification could be considered passing the quality control.

Because the analysis below involves certain third-party software which cannot be made public (as requested by the authors of Qiu et al. [13]), we cannot include this analysis in the manuscript for publication. Therefore, we decided to present the details of this analysis (with Additional Fig. 11-19 and Additional Notes 1) in the current response letter only. We also provided all our codes but the key third-party software for this analysis on our GitHub repository (see Section "Additional files for the comparison with Qiu et al. 2016" on the README page of <https://github.com/gao-lab/HERE>). Users

who'd like to reproduce these results must request for the corresponding third-party software from the authors of Qiu et al. [13] by themselves first.

Choosing the previous study to compare

There are two previous reports concerning the distribution of RNA editing sites in human embryos, the Qiu et al. [13] and Li et al. [14]. While the former has focused on human embryonic edits and released a detailed list of editing sites identified, the latter focused mainly on pig embryonic edits and has no human editing sites released for comparison. Therefore, we chose Qiu et al. to compare.

Compare our edits directly with the released edits (with no per-sample information) from Qiu et al.

We first compared our identified edits with the released edits from Qiu et al.. In Qiu et al., the datasets GSE36552 and GSE44183 were considered, which were also used by us. We note that Qiu et al. discarded 81-68=13 samples, but the released edits were not annotated with the samples from which they were identified. Therefore, we can only use the union of our edits identified in all these 81 samples for this comparison. In addition, we noticed that Qiu et al. used hg19 as the reference genome, while we used hg38. Therefore, we used UCSC liftOver [15] to lift the edits from Qiu et al. to hg38 before comparing them to ours. Edits that could not be lifted to hg38 were discarded.

As shown below in Additional Fig. 11, 5,893 edits were identified in both, 2,920 in Qiu et al. only, and 70,980 in ours only.

Additional Figure 11. Direct comparison of edits overlaps between those reported by Qiu et al. [13] (liftOver-ed

to hg38) and ours.

Reimplement Qiu et al.'s identification pipeline to obtain per-sample edits

To explain why we identified much more edits than Qiu et al., we need to first obtain the per-sample edits by reimplementing Qiu et al.'s identification pipeline. Because Qiu et al. did not release any source codes publicly, we contacted them and requested the codes. We received codes starting from the GATK-recalibrated bam files; codes for early steps are currently unavailable from the authors. Therefore, we reimplemented the rest codes and tried to reproduce the results, with the exception that we switched to hg38 assembly to avoid artifacts introduced by liftOver-ing variants from hg19 to hg38 (see Additional Fig. 12 below for the modification we made to switch to hg38; also see <https://github.com/gao-lab/HERE/blob/master/README.org> for codes of the entire reimplementations). We also noted that we do not have access to the unpublished Dai and Mongolian population datasets mentioned in Qiu et al., so we did not use them.

Feature	original (hg19-based)	hg38-based reimplementation
Reference genome	hg19.fa	hg38.fa
Reference gene GTF	Ensembl 75 GRCh37	GENCODE v32 (Note: we did not find GRCh38 versions for Ensembl75)
dbSNP	dbSNP 138 GRCh37	dbSNP 151 GRCh38 (Note: we did not find GRCh38 versions for dbSNP 138)
1000Genomes variant VCF	UCSC hg19 version Phase 3 (the \$version for each chromosome is v5a for autosomes and v1b for chrX/chrY)	UCSC hg38 version
GoNL Phase 2	original file from (in hg19)	liftover-ed original file to hg38 by picard LiftoverVcf
Combined reference for bwa alignment	hg19.fa + Ensembl 75 GRCh37	hg38.fa + GENCODE v32
simple repeats	UCSC hg19 version (Simple Repeats track, simpleRepeat table)	UCSC hg38 version (Simple Repeats track, simpleRepeat table)
Alu	UCSC hg19 version (RepeatMasker track, rmsk table)	UCSC hg38 version (RepeatMasker track, rmsk table)
ANNOVAR refGene	hg19 version	hg38 version

Additional Figure 12. Modifications we made to Qiu et al.'s pipeline for switching to the hg38 assembly. Links for features in the table: 1000Genomes, UCSC hg19: <ftp://hgdownload.soe.ucsc.edu/gbdb/hg19/1000Genomes/phase3/> ; 1000Genomes, UCSC hg38: <https://hgdownload.soe.ucsc.edu/gbdb/hg38/1000Genomes/> ; GoNL Phase 2:

http://molgenis26.gcc.rug.nl/downloads/gonl_public/releases/release2_noContam_noChildren_with_A_N_AC_stripped.tgz .

Select samples passing the "uniquely mapped bases" filter

We then examined the number of samples passing the threshold that the sample should have "over 0.5Gb uniquely mapped bases", as described by Qiu et al. [13]. Of all 81 human embryonic samples from GSE36552 and GSE44183, 76 passed this threshold (Additional Fig. 13), more than the reported 68 samples in the original paper.

Additional Figure 13. The number of uniquely mapped bases per sample examined. The red dashed line indicates the threshold of 0.5Gb (i.e., $0.5 * 1E9$).

We asked the authors why there's such inconsistency but did not receive any reply yet. We then proceeded with the 76 samples.

Summarize the number of edits across all valid samples

By running the steps in RNA_editing.pl provided by the authors (see Additional Notes 1 for more details), we identified a median of 3,787.5 edits among these samples (Additional Fig. 14a) with a union size of 271,035, far exceeding the number reported by the authors (8,813). Furthermore, the overlap between the 271,035 edits and the hg38-liftOver version of the reported edits (which is also 8,813) contains only 3,776 edits (Additional Fig. 14b).

a

b

Additional Figure 14. Distribution of edits identified by a reimplementation of the pipeline directly.

(a), count of edits identified by reimplementation in each sample. (b), overlap between the edits identified in the reimplementation and the edits reported by Qiu et al. [13].

Therefore, we suspected that this script did not use all the filters. For example, it can only accept one sample at a time and thus is not likely to filter against edits found in one sample only (see Filter (7) of Qiu et al. [13]); in addition, it might have only computed the p -values for filtering, but did not really use these p -values to filter the editing sites. Therefore, we used the intermediate outputs to locate possible p -values and other metrics for each of these sites, and filtered them accordingly (for details, see Additional Notes 1). This resulted in a median of 49 edits among these samples (Additional Fig. 15a) with a union size of 2,700, far lower than the number reported by the authors (8,813). In addition, its overlap with the hg38-liftOver 8,813 reported edits contains only 896 edits (Additional Fig. 15b).

a

b

Additional Figure 15. Distribution of edits identified by reimplementation (after re-filtering) of the pipeline directly. (a), count of edits identified by reimplementation (re-filtered) in each sample. (b), overlap between the edits identified in the reimplementation (re-filtered) and the edits reported by Qiu et al. [13].

Determine the final set of edits identified by Qiu et al.'s pipeline

To find out why the number of kept edits after filtering was so low, we decided to make a per-sample comparison of edits identified with all but the p -value filter of Step 1 activated, whose reason is detailed below. The p -value computed in Step 1 is based on the background mismatch rate, which seemed to be computed from the recalibrated bam (using the MismatchStat software). Only those sites whose read distribution is an outlier to the overall binomial distribution of all sites in the bam file were kept. However, for an RNA-seq sample, quite a few (if not most) of its variant sites could arise from RNA editing. In fact, this has been allowed in the RNA-seq-alone RNA editing identification pipeline [16]. Carrying out such a binomial test, therefore, would discard many potential RNA editing sites and suffer from a high false negative rate. In addition, re-applying all but this filter also gave a total edit count more comparable to the reported than previous ones (median 449.5 and total 13,500) (Additional Fig. 16a); its overlap with the reported hg38-liftOver 8,813 edits also contains 3,335 edits, a number close to the 3,776 edits before re-applying any filters, suggesting that the p -value computed in Step 1 might have not been applied by the authors at all (Additional Fig. 16b).

a

b

Additional Figure 16. Distribution of edits identified by reimplementations (after re-filtering without Step 1's p -value filter) of the pipeline directly. (a), count of edits identified by reimplementations (re-filtered without Step 1's p -value filter) in each sample. (b), overlap between the edits identified in the reimplementations (re-filtered without Step 1's p -value filter) and the edits reported by Qiu et al. [13].

Compare per-sample edits between Qiu et al.'s pipeline and ours

We compared the identified editing sites by Qiu et al's pipeline (named 'Qiu2016') and those by ours in the corresponding samples (named 'ours'). We found that the overlap between these two (defined as the number of edits identified in both divided by the number of edits identified in at least one of them) was very poor, with a median of 5.85% and a maximum of 10.29% (Additional Fig. 17).

group FN: detected in Qiu et al. only FP: detected in ours only TP: detected in both

Additional Figure 17. Overlap of edits between those identified by ours, and those identified by reimplementation (after re-filtering without Step 1's p -value filter) of the pipeline directly.

We then decided to explain why there is such a huge discrepancy between the two pipelines.

Explain our 'false negatives' with respect to Qiu2016

We first examined our 'false negatives' with respect to Qiu2016 in each sample, defined as edits identified by Qiu2016 but not by ours. There are in total 9,798 edits (with a median of 251 in each sample) that are determined to be 'false negatives'. After a careful checking we can explain >75% false negatives in all samples with the following reasons: (1) we discarded sites with multiple alternative alleles identified (explaining a median of 3.91% false negatives); (2) we used a more comprehensive set of population genotypes from worldwide cohorts (explaining 54.58%); (3) we discarded sites with invalid sample occurrence (note that we required non-Alu sites to be identified at least twice in the same normal stage or the same abnormal stage, not across all samples; explaining 10.13%); and (4) we discarded sites whose event on the known transcripts does not contain the A-to-G case (explaining 20.28%) (Additional Fig. 18).

Reasons of FN

- 1. Multiple alleles detected
- 2. Overlapping with variants from worldwide cohort genotyping
- 3. Invalid sample occurrence
- 4. No A-to-G variant on known transcripts
- 5. Others

Additional Figure 18. Possible reasons and % explained false negatives (FN) for each reason in each sample. The intercept of the dashed line is 0.25.

Explain our 'false positives' with respect to Qiu2016

We then examined our 'false positives' with respect to Qiu2016 in each sample, defined as edits identified by ours but not by Qiu2016. There are in total 108,877 edits (with a median of 3,058 in each sample) that are determined to be 'false positives'. After a careful checking we can explain $\geq 66\%$ in $\geq 75\%$ samples with the following reasons: (1) some sites were not aligned to the genome by TopHat2 in the first round alignment (explaining a median of 2.81% false positives); (2) some sites were not supported by any reads at all in BWA realignment (explaining 34.96%); and (3) some sites were supported by at least one read in BWA realignment, but were still discarded anyway (explaining 40.48%) (Additional Fig. 19).

Reasons of FP

- 1. Not detected in first-round alignment
- 3. Detected in BWA realignment but discarded anyway
- 2. Not detected in BWA realignment
- 4. Others

Additional Figure 19. Possible reasons and % explained false positives (FP) for each reason in each sample. The intercept of the dashed line is 0.34.

Final conclusions

1. While there does exist a certain previous study (the Qiu et al. [13]) to compare with, its identification pipeline could not be fully reimplemented even after several rounds of communication with the authors;
2. Based on our best reimplementation results, we made several attempts and can explain in all samples $\geq 75\%$ edits that were identified by Qiu et al. but not us (by our more stringent criteria), and in $\geq 75\%$ samples $\geq 66\%$ edits that were identified by us but not by Qiu et al. (by differences in the choice of sequence aligners in the first-round alignment and realignment);
3. Therefore, our identification could be considered passing the quality control.

2. In Fig2a, only 23% sites were shared between normal and abnormal samples, what is the potential biological explanation?

We thank for the reviewer for pointing out this interesting observation, which helped us to investigate potential technical limitations of our identification. Currently, we have two possible reasons for this small overlap.

The first reason for such a small overlap is the fact that normal and abnormal samples in our collection are not aligned at the stage level. In particular, only the following stages have both normal and abnormal samples at the same time: zygote, zygote (2PN), 2-cell, 4-cell, 8-cell, and morula. Each of all other stages consists of only normal samples, or only abnormal samples, but not both types of samples at the same time, thus contributing to some of the normal/abnormal-specific edits.

The second reason might be the insufficient detectability of RNA edits in samples that were less deeply sequenced. As shown in Additional Fig. 20, for each of the stages with both normal and abnormal samples, the more bases were sequenced, the more edits would be identified.

Additional Figure 20. Scatterplot of the count of edits identified vs. count of total bases sequenced in each sample.

As a consequence of this issue, samples from the same group would be more likely to have a large overlap in their edits, if their sequence depths are more similar (Additional Fig. 21). Nevertheless, even for the most similar pairs, the overlap is still too low (0.3~0.4), suggesting that the current sequencing depth is still not large enough for a complete description of the editing events in human embryos.

Additional Figure 21. Sample pair scatterplot of the overlap ratio (defined as the number of edits identified in both samples divided by the number of edits identified in at least one of the

samples) vs. the distance of sequencing depth (defined as the absolute difference between the $\log_{10}(\text{count of total bases sequenced})$). Good/bad viab., predicted to be of good/bad viability (see the paper of GSE65481 [17] for more details).

This is one of the reasons why we currently focus on REEs, a subset of RNA editing patterns that occurs in at least half of embryos and is unlikely to be false positives upon observation due to insufficient sequencing depth. We note that this trend was inversed in certain groups of 8-cell and morula (i.e., 8-cell/abnormal/AG and morula/abnormal/AG); however, we note that in these stages the number of REEs is much lower compared to that in earlier stages (Fig. 2b), which suggests a lower editing activity (with respect to REEs) and might bias this trend anyway, and therefore we cannot conclude immediately that sequence depths in these samples are high enough for complete detection of RNA edits. Interestingly, this trend is stronger in normal samples than in abnormal samples, which might be due to the dysregulation of RNA editing in these abnormal samples.

Aside from the possible reasons above, we also note that, while individual variation might bring observable differences within normal or other samples, this difference did not seem to contribute the most to the overall variation, as otherwise all these samples should mix across stages (Additional Fig. 22).

UMAP plot

Additional Figure 22. Editing level-based Uniform Manifold Approximation and Projection (UMAP) plot for samples from stages with both normal and other samples available. The generation of the editing level table for dimensional reduction followed that of [11], except that only edits passing our editing level threshold (i.e., ≥ 0.1) were considered.

We did not include these in our manuscript, as the overlap of general edits between normal and abnormal samples is not our major focus here.

3. Does the number in Fig2b represent intersect or union number of sites from different replicates of the same sample? Also, number of reproducible sites from different replicates was expected as quality control.

We thank the reviewer for this comment and the suggestion.

For the question, the number in the old Fig. 2b (now renumbered as Supplementary Fig. 3) represents the union number of sites from different (biological) replicates, if any, of the same sample. Technically speaking, however, our curation was based on 18 different RNA-seq datasets from various studies, and therefore samples from the same stage are very likely to come from different individuals with different genetic backgrounds, and should not be regarded as replicates in the strict sense.

For the reviewer's suggestion on using the number of reproducible sites as quality control, we did not find any study of identifying A-to-I edits from large-scale RNA-seq human samples (including the famous GTEx-based identification [11]) that used this metric as quality control. Rather, the RNA-seq- alone identification developed by Ramaswami et al. [16] did consider Alu edits identified in only one sample as valid edits. On the other hand, we note that our focus on the human early embryonic editing sites, REE, could be regarded as a highly reproducible subset of edits, because they are defined to be observed across $\geq 50\%$ of samples.

We have added the union description for the number to the old Figure 2b (now renumbered as Supplementary Figure 3) legend (Page 4, Line 21 in main text; Pages 16-17 in supplementary material).

4. P7.I149 seems should be "median" instead of "mean".

We thank the reviewer for pointing out the misuse of these terms.

We corrected all misuses of "mean"/"median" (Page 27, Lines 8, 12, and Page 29, Lines 4, 8 in main text; Page 36, Line 6 in supplementary material).

Reviewer #2 information

Remarks to the Author: Overall significance

By curating and analyzing the largest human embryonic editome to date, Ding and coauthors showed that the human preimplantation embryos at various embryonic stages harbour thousands of REs that are preferably exonic and highly shared between stages at the editing site. They also proposed that these REs could potentially enhance maternal mRNA clearance, a process that is crucial for successful maternal-to-zygotic transition, by introducing more MBSs to clearance targets than to other maternal genes.

We thank the reviewer for the nice and concise summary.

Remarks to the Author: Impact

This study introduced the first large-scale A-to-I RNA editome for early human embryos, the analysis of which revealed a consistent early-stage editing pattern (of REs) with probable functional

importance in microRNA-based maternal mRNA clearance. The editome itself is a valuable resource for further examination of the interplay between maternal RNAs and early embryo development. However, many of the conclusions need to be experimentally confirmed. And some key hypotheses also need to be examined by bench works.

While experimental validation of results would not be necessary for further consideration at *Communications Biology*, the lack of new data should be explicitly stated as a limitation and all relevant conclusions regarding maternal RNA clearance pathways should be appropriately qualified.

We thank the reviewer for the praise regarding our editome, as well as one of the key drawbacks of our study (as also raised by the editor) that we lack experimental validation of our results.

We added the statement describing the limitation of lack of new data as discussed below in Comment #5 of the "Strength of the claims" section of Reviewer 2 (Page 10, Lines 10-12 in main text). We also addressed all relevant conclusions regarding maternal RNA clearance pathways (including those concerning the relationship between REEs and gene expression levels), as discussed in Comments #4- #8 of Reviewer 1, Comments #4 and #6 of the "Strength of the claims" section of Reviewer 2, and Comments #9, #14, #15, #18 of Reviewer 3. Please see the reply to each of these comments for more details.

Remarks to the Author: Strength of the

claims Specific comments:

The title of "Large-scale identification of recurrent RNA edits in human embryos" is unclear. While the authors only investigated A-to-I RNA editing in human preimplantation embryos, it is better for them to be more specific in defining the scope of research in the title.

We generally recommend that the title be written as a declarative statement (<15 words) that includes any key species, protein, or gene names.

We thank the reviewer and the editor for the comments that make the title more specific.

We have rewritten the title as "Recurrent RNA edits in human preimplantation potentially enhance maternal mRNA clearance" (11 words) (Page 1, Lines 1-2 in main text).

The reader would benefit from a more detailed description of A-to-I RNA editing (i.e., frequency, sequence preference, and distribution in the 5' and 3' untranslated region, coding sequences of mRNAs) and the biochemical pathways underlying this edit (i.e., enzymes and RNA-binding proteins that mediate this post-transcriptional modification) in the Introduction section.

We thank the reviewer for pointing out the lack of such background description that would benefit readers more on RNA editing.

We have detailed current knowledge of the landscape of human RNA edits in Introduction, as the reviewer has suggested (Page 3, Lines 5-9 in main text). Because our main focus is the potential consequences, rather than the mediators, of A-to-I editing, we briefed the *ADAR* enzyme and its target (double-stranded RNA) in the Introduction (Page 2, Lines 20-22 in main text).

The authors should be sure to define abbreviations in figure legends and in the main text and minimize the number of abbreviations. As the manuscript is now, it is difficult to follow the flow on the story due to so many abbreviations.

Please avoid abbreviating terms unless they are used five or more times. We ask that you avoid all non-standard 2 letter abbreviations.

We thank the reviewer and the editor for the comments on avoiding filling the manuscript with too many different abbreviations.

As described in the summary, we have replaced the term "RE" (Recurrent Edit) with "REE" (Recurrent Embryonic Edit).

In addition, we have replaced all abbreviations below with their full names or alternative descriptions: NCBI, GEO; AMA (for advanced maternal age); GO (for Gene Ontology); CDS (for coding sequence); scDNA-/RNA-seq (for single-cell DNA-/RNA-seq).

The following abbreviations were removed from the main text, but kept in the old Fig. 2b (now renumbered as Supplementary Fig. 3), the old Fig. 3b (now Fig. 2b), and the old Supplementary Fig. 8 (now Supplementary Fig. 5), because they are standard abbreviations used in embryonic studies and are needed to describe the number of samples per stage: oocyte (MI), PN, TE, ICM, hESC, CTB, STB, EVT, MTB.

The following standard abbreviation was kept in figures to save plotting space: BH-adjusted (for Benjamini-Hochberg-adjusted).

The following abbreviations were kept because they are standard and used ≥ 5 times: oocyte (GV), oocyte (MII); BI, AG, PG (used by [8]).

The following abbreviation was kept because they are used ≥ 5 times: MBS.

The authors showed that A-to-I RNA editing could potentially enhance maternal mRNA clearance, by introducing more MBSs to clearance targets than to other maternal genes. However, the role of an microRNA-based regulatory mechanism in maternal mRNA clearance has not been confirmed in mammalian early embryos. Therefore it is reluctant to explain the association of A-to-I RNA editing with maternal mRNA decay using this mechanism. Potential involvement of A-to-I RNA editing in PAN2-PAN3, CCR4-NOT, and exosome mediated RNA clearance pathways should be considered.

Please acknowledge these other pathways and appropriately qualify conclusions regarding maternal clearance, for further consideration at *Communications Biology*.

We are very grateful to the reviewer and the editor for pinpointing these key components of RNA degradation we had missed previously, and we detailed our analyses on them as shown below.

We first examined whether the A-to-I edits could affect the key gene in each of these pathways. Specifically, we examined whether REEs target the following genes:

1. PAN2-PAN3 [18] :
 - a. Core subunits: *PAN2*, *PAN3*
2. CCR4-NOT [19] :
 - a. Scaffold: *CNOT1*
 - b. Unknown but contributes to stabilization of the complex and RNA substrate recruitment: *CNOT2*, *CNOT10*, *CNOT11*
 - c. Interaction with ribosomes: *CNOT3*
 - d. Ubiquitin E3-ligase activity: *CNOT4*
 - e. Deadenylase: *CNOT6*, *CNOT6L*, *CNOT7*, *CNOT8*
 - f. Transcriptional cofactor: *CNOT9*
 - g. Multifunctional: *TNKS1BP1*
3. RNA exosome [20]:
 - a. Subunit of the 6-subunit ring of the 10-subunit core exosome: *EXOSC4*, *EXOSC5*, *EXOSC6*, *EXOSC7*, *EXOSC8*, *EXOSC9*
 - b. Subunit of the 3-subunit cap of the 10-subunit core exosome: *EXOSC1*, *EXOSC2*, *EXOSC3*
 - c. Ribonuclease/Catalytic subunit: *DIS3*
 - d. Riboexonuclease subunit: *EXOSC10*
 - e. Nuclear exosome cofactor: *TENT4B* (also known as *PAPD5*), *ZCCHC7*, *MTREX* (RNA helicase; also known as *MTR4*), *MPHOSPH6* (also known as *MPH6* and *MPP6*), *C1D*, *SETX*, *ZCCHC8*, *RBM7*
 - f. Cytoplasmic exosome cofactor: *SKIV2L* (RNA helicase), *TTC37* (whose yeast ortholog is *Ski3*), *WDR61* (whose yeast ortholog is *Ski8*), *HBS1L* (isoform 3; whose yeast ortholog is *Ski7*)

We found that multiple REEs target *EXOSC6* in oocyte (GV), oocyte (MII), and zygote, and a single REE targets *CNOT6* in 2-cell (Additional Fig. 23 and also Supplementary Fig. 24). While the function of *EXOSC6* has been rarely studied, the mouse ortholog of *EXOSC10* in the RNA exosome degradation pathway has been found to be required for the growth-to-maturation transition in oocytes [21] and eight-cell embryo/morula transition [22]; therefore, it is also possible that A-to-I editing on *EXOSC6* is a preferred regulation on the RNA exosome degradation pathway by oocytes (GV), oocytes (MII), and/or zygotes. On the other hand, the mouse ortholog of *CNOT6* gene has been found to regulate deadenylation of mRNAs in mouse oocyte growth and maturation [23], and the mouse CCR4-NOT complex itself is also found to be involved in regulating 2-cell-specific genes [24], suggesting the possibility that A-to-I editing on *CNOT6* might be preferred in the 2-cell stage as well.

a

b

c

REE editing profile on *CNOT6*

Additional Figure 23. *EXOSC6* and *CNOT6* are the REE-targeted genes of the RNA degradation pathways in question. (a), editing profile of REEs on *EXOSC6* across oocyte (GV), oocyte (MII), and zygote stages. (b), UCSC track of *EXOSC6*-targeting REEs. (c), editing profile of the single REE on *CNOT6* in the 2-cell stage.

Because the CCR4-NOT complex is known to target 3'-UTR of mRNAs with certain sequence motifs [25], we also examined whether REEs could lead to gain or loss of such motifs. Specifically, we examined the Pumilio-response element (PRE) UGUANAUW (where N is any nucleotide and W is either A or U) [26], and the AU-rich element (ARE) UUAUUUAUU [27]. In addition, we also examined the hnRNP A1 and A2/B1 binding site UAASUUAU (where S is either C or G) as discovered in [28]. We found that REEs in normal early stages did alter 6 motifs for 5 genes (Additional Fig. 24 and Supplementary Fig. 25), suggesting that A-to-I could indeed be able to affect (CCR4-NOT-based) RNA degradation at the level of CCR4-NOT sequence motif.

Additional Figure 24. Editing profile for normal early stage REEs that alter the CCR4-NOT sequence motifs.

We also note that the miRNA-based maternal mRNA decay pathway, if exists, might still be able to recruit PAN2-PAN3 and CCR4-NOT via the protein TRNC6A (also known as GW182) as discovered earlier [25,29–31]. Therefore, miRNA binding sites introduced by A-to-I edits might further enhance the degradation of target mRNAs by PAN2-PAN3 and CCR4-NOT-based pathways.

We added the above as Supplementary Notes 5 (Pages 4-6, 44-46 in supplementary material) to Discussion (Page 9, Lines 7-23 in main text).

The occurrence and functions of A-to-I RNA editing in human preimplantation embryos are completely based on high throughput data analyses. The results of analyses are no doubt valuable, but it is equally important to experimentally confirm the location of A-to-I RNA editing on some representative maternal transcripts, and provide experimental results that support the hypotheses that A-to-I RNA editing indeed affect the dynamics of maternal mRNA clearance.

We completely agree with the reviewer's concern about the lack of experimental support for our computational results.

Due to technical limitations, we were unable to carry out experimental validations on these transcripts by ourselves. We did not find any other experimental evidence from previous studies that support our hypotheses. We added this limitation of our work to Discussion (Page 10, Lines 10-12 in main text).

In Figure 7, there is an error about the dynamics of maternal mRNA clearance in human embryos: the major wave of maternal mRNA clearance occurs at the 8 cell stage, not after blastocyst formation as is shown in the left corner of the proposed model.

We thank the reviewer for highlighting this subtle yet serious problem.

We corrected this error (now renumbered as Fig. 6; Page 30, Lines 1-2 in main text).

Authors need to follow the guidelines of gene symbols, protein symbols, RNA symbols. Based on our understanding gene and RNA symbols should be italicized.

Please ensure all gene and RNA symbols are italicized in the text and figures.

We thank the reviewer and the editor for reminding us of following the guidelines.

We italicized all gene and RNA symbols. For cases where the entity referred to is a protein (e.g., "ADAR" in "ADAR-binding motif"), we did not italicize them.

Reviewer #3 comments

Remarks to the Author: Overall significance

In the submitted manuscript, the authors reported their analysis of A-to-I RNA editing in human embryos. They found that editing is diminished in abnormal embryos and that it can introduce microRNA binding sites in 3'UTR of maternal transcripts to facilitate their clearance. The topic of the study is interesting and might be clinically relevant to reproduction and fertility. Nevertheless, I think the work can be further improved.

We thank the reviewer for the encouraging comments on our study.

Remarks to the Author: Impact

Communications Biology would be most appropriate for the manuscript.

We thank the reviewer for the recommendation of our study to *Communications Biology*.

Remarks to the Author: Strength of the

claims Major comments:

1) The authors should elaborate more about their analysis steps in the main text. Details of any additional filters applied (besides SNP removal) are missing from Figure 1b. As written, there seems to be nothing special about the authors' computational pipeline.

A similar point was raised by Reviewer #1.

We thank the reviewer and the editor for pointing out the lack of enough details in our writing of methodology.

As discussed in the reply to the first question of Reviewer #1, we had detailed these steps in the previous Supplementary Notes 1 (Supplementary Notes 6 in the revision) in the original manuscript, and in the revised manuscript we reorganized this part to make each filter clear to readers (Pages 8-12 in supplementary material). Because we had adapted a well-established pipeline from Ramaswami et al. [16], most of our steps are identical to its and we had only highlighted the adaptations we made to this pipeline in Figure 1b. The major change to the pipeline in this adaptation is the usage of additional genotypes from worldwide cohorts that span hundreds of thousands of people [3,4]. This adaptation thus stringently filters against false positives from those genomic variants that can emerge at this size of human population.

2) Figure 1 should include the distribution of all 12 possible types of nucleotide changes (A-to-C, A-to-G, A-to-T etc) after variant calling with or without application of additional filters.

A similar point was raised by Reviewer #1.

We thank the reviewer and the editor for this nice suggestion to further solidify the reliability of our identification pipeline.

We have updated Figure 1 (which we assumed to be Figure 1d) with the distribution plot across all possible 12 types of nucleotide changes (Additional Fig. 25 and Supplementary Fig. 1; it is too big to be put into the main figure) (Page 21, Lines 5-6 in main text; Page 13 in supplementary material). As discussed in Supplementary Notes 6, previous studies have identified potentially functional RNA edits that are located on transcripts of different orientations [5-7], and therefore we also plotted them separately. From the plot, it is clear that our pipeline did filter out false positives effectively, and the remaining variants are mostly A-to-G and A-to-G-or-T-to-C variants, consistent with the expected variant type distribution of RNA editing pipeline identification.

Additional Figure 25. Distribution of all possible types of nucleotide changes (see also Figure 1d). Because it is impossible to tell what type of RNA variant is when the variant location is on two transcripts of different orientations (e.g., "A-to-G" and "T-to-C"), and because previous studies have observed such edits with potential functions (see discussion in the subsection "Filter against unreliable variants" in Supplementary Notes 6) [5–7], we plotted such variants separately rather than treated them as the "mixed variants", where multiple types of variants were observed at the same location.

3) Figure 1D: How was the false positive (FP) rate calculated? Please describe in main text.

We thank the reviewer for locating this lack of explanation of this critical metric.

We have added the following description of FP (Page 4, Lines 13-15, and Page 21, Lines 3-4 in main text). The FP rate is the ratio of A-to-I RNA edits identified that matched the DNA variants in the same cell (note that this is a DNA-RNA-paired single-cell sequencing dataset).

4) The A-to-G percentage should be given for non-Alu and Alu sites separately.

We thank the reviewer for the suggestion of examining the A-to-G percentage on different types of edits separately.

We have updated all plots with A-to-G percentage in Alu- and non-Alu-subsets (Additional Fig. 26 and Supplementary Fig. 6) (Page 21, Lines 11-12 in main text; Page 21 in supplementary material).

Additional Figure 26. Distribution of percentage of A-to-G variants for Alu and non-Alu sites separately.

5) The authors should calculate the Alu editing index (AEI) for each sample. Does the extent of editing correlate with ADAR expression levels?

This point would be necessary for further consideration at *Communications Biology*.

We thank the reviewer and the editor for reminding this useful metric of RNA editing.

The original AEI computed by <https://github.com/a2iEditing/RNAEditingIndexer> as published by Roth et al. [32] starts from RNA-seq read alignments (in BAM format) and calls the RNA edits by its own pipeline. This pipeline does not consider the various filters we used in our pipeline. To compute the AEI more accurately, we re-implemented AEI following the definition of Roth et al., i.e., "weighted average [of editing index] over [all] tens of millions of genomic adenosine sites located within Alu sequences, where the weights are the coverage of each site", where "editing index" for a given adenosine site is defined as "the ratio of the number of A-to-G mismatches to the total coverage of adenines (that is, the sum of the number of A-to-G RNA-DNA mismatches and A-A matches in these regions)".

In practice, we skipped the Steps 1-3 in Supplementary Notes 4 (Pseudocode) of Roth et al., and started from Step 4 (see the code at https://github.com/gao-lab/HERE/blob/master/scripts/A02_10_compute_AEI/run.R). Specifically, (a) because Roth et al. have demonstrated that their strand selection strategy (Fig. 1d and Methods in Roth et al. [32]) "has a high agreement between the [Alu-editing] indexes calculated both with and without taking into account the correct strand information (Fig. 2a, Supplementary Fig. 1 and Supplementary Notes 1)", for simplicity we assumed all datasets were unstranded, and followed the "Strand assignment" section in Methods of Roth et al. to determine the strandness for each Alu element in each sample; (b) to make the results consistent with our pipeline, we used GENCODE version 32 instead of RefSeq, and for (3) of "Strand assignment", we used for each sample its own expression profile to determine the expressed strand. We then took the signal-to-noise ratio (as suggested by legend of Fig. 1d in Roth et al., with the 'mismatches of other sources' chosen to be C-to-T index as suggested by Roth et al.), and used it as the AEI.

The overall AEI across all stages is slightly lower than that in adult tissues reported by Roth et al. (Additional Fig. 27 and Supplementary Fig. 20, boxplot part). We also noted that a number of samples do not have valid AEI due to their control index (which is used to divide the raw AEI to get the final AEI) being 0 (Additional Fig. 27 and Supplementary Fig. 20, barplot part).

Additional Figure 27. Distribution of AEI across different stages of embryonic samples.

Their correlations with *ADAR* expression are not statistically significant (Additional Fig. 28 and Supplementary Fig. 21), which is consistent with the observation that such correlation is rather weak in adult human tissues [32].

Additional Figure 28. Histogram of Spearman's correlation coefficients between AEI and *ADAR* FPKM for each normal stage with ≥ 10 normal samples. All those that have a Benjamini-Hochberg-adjusted p -value for the correlation coefficient test (with the alternative hypothesis being that the correlation coefficient is larger than 0) that is no less than 0.05 were considered not significant (NS).

We added the above analysis to Supplementary Notes 2 (Page 4, Line 22 in main text; Pages 2-3, 41- 42 in supplementary material). Nevertheless, we cannot conclude immediately that these results support or disqualify our identification, because while these results might reflect the true underlying biological mechanism, the number of edits identified is still associated with the sequencing depth (see the reply to Comment #2 of the "Reproducibility" section by Reviewer #1 above), thus biasing the results observed here.

6) Figure 1 & 2 can be combined.

We thank the reviewer for the recommendation of combining the two figures, and we combined them into one (Pages 20-21 in main text), with the original Figure 2b subtracted and renamed as Supplementary Figure 3 (Pages 16-17 in supplementary material).

7) What is the number of samples for each developmental stage (Figure 3b)?

We thank the reviewer for pointing out the lack of this important methodological detail.

Based on the original per-sample table (Supplementary Table 7; see reply to Comment #2 of Reviewer #1), we have added Supplementary Table 8 and Supplementary Figure 5 (also shown below as Additional Fig. 29) (Page 12, Lines 11-12 in main text, and Pages 20-21, 46-47 in supplementary material) describing the number of samples for each developmental stage.

Additional Figure 29. Count of normal or abnormal samples (in log10 scale) in each stage. Groups labeled with * have only one sample. GV, germinal vesicle. MI, metaphase of first meiosis. MII, metaphase of second meiosis. PN, pronuclear. TE, trophoctoderm. ICM, inner cell mass. hESC, human embryonic stem cell. CTB, cytotrophoblast. STB, syncytiotrophoblast. EVT, extravillous trophoblast. MTB, migratory trophoblast.

8) Figure 3c,d: It would be useful to show similar graphs for later developmental stages (morula and afterwards) as well. Is the preponderance of sites in 3'UTR observed only in the earlier developmental stages?

This point would be necessary for further consideration at *Communications Biology*.

We thank the reviewer and the editor for the suggestion of completing the analysis of RNA editing in both earlier and later developmental stages.

We have added similar graphs of the old Fig. 3c and 3d (now renumbered as Fig. 2c and 2d, respectively) for these samples (Additional Fig. 30 and Supplementary Fig. 9) (Page 23, Lines 6-8 in main text; Pages 22-23 in supplementary material). We found that the preponderance of sites in 3'-UTR was also observed in samples from later developmental stages, though with a limited number of REEs.

a

b

Additional Figure 30. Percentage of exonic edits and REEs (a) and percentage of 3'-UTR REEs (b) in the late stages of embryogenesis. In (a), each stage label is appended with an additional label describing the number of exonic edits (or REEs) / the number of total edits (or REEs).

9) Genes with recurrent edits (REs) in early stages - do their expression levels go down in later developmental stages (morula and afterwards), indicative of transcript degradation?

We thank the reviewer for proposing this potentially important role of RNA editing that might affect post-implantation embryo development by editing pre-implantation transcripts, which we had not considered previously.

Because the reviewer did not explicitly state which specific early stage(s) we should compare with the "morula and afterwards" stages, we assumed that the reviewer was referring to the 8-cell stage, the only stage that precedes morula immediately. In 8-cell, however, most maternal mRNA clearance ends [33], and a wave of zygotic genome activation can be observed [34]. Under such drastic transcriptomic changes, it is technically difficult to tell whether 8-cell REEs (which are very few (~100, see Fig. 2b)) are a major contributor to transcript degradation.

Nevertheless, we examined whether the editing level of these 8-cell REEs is negatively correlated with the expression level of their targeted genes in morula and subsequent stages, as suggested by the reviewer. We did not add this to our manuscript, because currently, we cannot tell whether the result was biased by the end of maternal mRNA clearance and the wave of zygotic genomic activation mentioned above. As shown below, this negative correlation did exist for these 8-cell-REE-targeted genes (Additional Fig. 31), and we could conclude in the sense of computational correlation (but not biological causality) that these REEs might be indicative of transcript degradation.

Additional Figure 31. Editing level of 8-cell REEs was negatively correlated with the morula, blastocyst (late), and ICM expression levels of their target genes. Only normal samples from early stages were considered. For each stage pair, only genes with FPKM > 0.001 were plotted. The trend of association is visualized by linear regression (with orange line denoting the fitted line and grey shadow denoting the 95% confidence interval).

10) The authors wrote that "An initial scan revealed 107 edits on 76 genes that were REs in normal embryos, but completely lost in the same stage in pathological embryos and embryos from elder mothers." What is the initial scan? Please provide more details on how the 107 edits on 76 genes were shortlisted.

We thank the reviewer for pointing out the lack of methodological clarity in this analysis. We added the following to Methods (Page 14, Lines 1-24, and Page 15, Lines 1-4 in main text) to make it clear:

We started with all REEs identified in oocytes (GV), oocytes (MII), zygotes, 2-cells, 4-cells, 8-cells, and morula. Then, we selected the union of the following REEs as the set of 107 edits:

1. For REEs identified in oocytes (GV), we selected those that could not be detected in any oocytes (GV) from elder mothers from GSE95477 [35];
2. For REEs identified in oocytes (MII), we selected those that could not be detected in any oocytes (MII) from elder mothers from GSE95477 [35];
3. For REEs identified in zygotes, we selected those that could not be detected in any androgenetic (AG) zygotes from GSE133854 [8];
4. For REEs identified in zygotes, we selected those that could not be detected in any parthenogenetic (PG) zygotes from GSE133854 [8];
5. For REEs identified in 2-cells, we selected those that could not be detected in any AG 2-cells from GSE133854 [8];
6. For REEs identified in 2-cells, we selected those that could not be detected in any PG 2-cells from GSE133854 [8];
7. For REEs identified in 4-cells, we selected those that could not be detected in any AG 4-cells from GSE133854 [8];
8. For REEs identified in 4-cells, we selected those that could not be detected in any PG 4-cells from GSE133854 [8];
9. For REEs identified in 8-cells, we selected those that could not be detected in any AG 8-cells from GSE133854 [8];
10. For REEs identified in 8-cells, we selected those that could not be detected in any PG 8-cells from GSE133854 [8];
11. For REEs identified in morulae, we selected those that could not be detected in any AG morulae from GSE133854 [8];
12. For REEs identified in morulae, we selected those that could not be detected in any PG morulae from GSE133854 [8].

For each site, was there sufficient sequencing coverage in the abnormal embryos? What were the editing levels of these 107 sites in the normal embryos?

We thank the reviewer for the suggestion of a more detailed examination of these edits, which would help readers better identify those edits of interest.

For the first question, we examined the sequencing coverage and editing level on these sites (Additional Fig. 32 and Supplementary Fig. 10) (Page 27, Lines 1-2 in main text; Pages 23-26 in supplementary material) and found that, while most of these sites have been sequenced (i.e., with non- zero coverage), in a number of GSE133854 samples the coverage did not pass the 10 threshold in our identification pipeline, suggesting that the absence of these edits might be potentially false negatives.

b

Coverage 0 [1, 9] [10, 99] >=100

c

Additional Figure 32. Coverage of each of the 107 REE in the samples from elder mothers (a), androgenetic/parthenogenetic zygotes (b), or the parthenogenetic 2-cells (c). For each of these embryo types, only those REEs determined to be completely lost were displayed.

For the second question, we plotted the editing level of these REEs in the corresponding stages (Additional Fig. 33 and Supplementary Fig. 11) (Page 27, Lines 2-3 in main text; Pages 26-30 in supplementary material).

a

b

Additional Figure 33. Distribution of normal sample editing level of the 107 REEs lost in oocytes from elder mothers (a), androgenetic embryos (b), or the parthenogenetic embryos of the corresponding stages (c).

11) Figure 5a,c: What are AG embryos and PG embryos? Also, what is "BI"? Please explain in the figure legend.

We thank the reviewer for highlighting the lack of explanation of these important abbreviations.

"AG" and "PG" embryos are two types of embryos with uniparental disomy. "AG" (short for "AndroGenetic") means that both of the two copies of chromosomes of the given embryo come from the father. Similarly, "PG" (short for "ParthenoGenetic") means that both come from the mother. "BI" stands for "Biparental", which means that for each pair of chromosomes, one comes from the father and one comes from the mother (i.e., a normal embryo). We borrowed the usage of these abbreviations from Leng et al. [8]. We have added these explanations to the figure legend (now renumbered as Fig. 4a and 4c; Page 26, Line 4, Page 27, Lines 1, 9 in main text).

12) I see "amanitin" in Figure 5c. Amanitins are compounds that block RNA polymerase II, thereby inhibiting transcription. How is this drug treatment relevant to the manuscript?

We thank the reviewer for reminding us to rethink the relevance of this drug here in our study.

After literature survey, we found that treatment with (alpha-)amanitin (as described by the paper accompanying the dataset GSE101571 [9]) will block RNA Pol II-based transcription, and has been shown to stop mouse early embryo development [36]. However, being applied with alpha-amanitin, a toxin from mushrooms, is not a low-quality-embryo case that has been observed in natural conditions previously. Therefore, this analysis is irrelevant to our story, and we removed it from our manuscript.

13) Figure 5c: Data from GSE133854 don't look convincing. This seems to invalidate the authors' claim that there are "fewer RE-matching edits in protein-coding genes of abnormal embryos and embryos from elder mothers".

This point would be necessary for further consideration at *Communications Biology*.

We are grateful to the reviewer and the editor for pointing out our misinterpretation of the results here.

We are sorry that we misread this result; not all abnormal (AG or PG) embryos support this claim. In particular, only AG (zygote), PG (2-cell), AG (4-cell), PG (4-cell), AG (8-cell), and PG (8-cell) have their median less than that of paired BI, and only PG (8-cell) is statistically significantly fewer. We have updated the description here (Page 6, Lines 21-25; Page 7, Line 1 in main text) as below:

" ... the number of REE-matching edits in abnormal embryos (or embryos from elder mothers) is smaller (though mostly statistically insignificantly) than that in normal embryos (or embryos from young mothers) in many but not all cases (Fig. 4c). However, we noticed that the difference is indeed larger in GSE95477. Therefore, we determined to examine whether there are certain subsets of

protein-coding genes actively regulated by REEs that underwent statistically significant loss in abnormal embryos or embryos from elder mothers. "

14) MicroRNA prediction programs are often inaccurate. Instead of relying solely on TargetScan, can the authors use at least two different software and take the intersection?

Please validate any miRNA predictions on at least one other tool, for further consideration at

Communications Biology.

We thank the reviewer and the editor for the suggestion on further validating our microRNA-related analysis with more stringent computational criteria.

In addition to TargetScan, we used a second MBS-predicting tool, miRanda [37], took the intersection of their predictions, and tried to use this intersection to validate the MBS-related results one by one. We have updated all figures related to this section in the main text and supplementary files (i.e., the old Figures 6b (now renumbered as Fig. 5b), 6c (now Fig. 5c), 6d (now Fig. 5d), and Supplementary Figure 5 (now Supplementary Fig. 15); we removed the old Figure 6e (see reasons in the reply to Comment #4 by Reviewer 1)).

How to take the intersection of the two tools

During the intersection, we noted that the predicted MBSs were defined at different levels for these two tools (Additional Fig. 34 and also Supplementary Fig. 18). For TargetScan, its predicted MBS is an alignment of a given miRNA family, denoted by its seed region sequence (i.e., the 2-8 nucleotides on the mature miRNA sequence), onto the given 3'-UTR sequence. For miRanda, its predicted MBS is an alignment of a given mature miRNA sequence onto the given 3'-UTR sequence. Because multiple mature miRNA sequences can share the same seed region sequence (and thus belong to the same 'miRNA family'), we need to take the intersection at the miRNA family level.

Additional Figure 34. How TargetScan and miRanda make their predictions.

Therefore, we collapsed the miRanda predictions, all of which were predicted with default parameters, to the miRNA family level before taking the intersection. Specifically, we collapsed into a single prediction all those miRanda predictions that share (1) the same miRNA family, (2) the same target 3'-UTR sequence, (3) the same seed region site type (as specified by TargetScan), and (4) the same start and end positions on the 3'-UTR sequence the seed region aligns to (Additional Fig. 35 and also Supplementary Fig. 19). miRanda predictions that do not share all of these four properties were considered as different predictions. As required by TargetScan, during the computation of the site type, we only considered exact matches (i.e., A-U/T and C-G), and wobble pairs (e.g., G-U/T) were excluded. We then took the intersection of TargetScan predictions and the collapsed miRanda predictions. Similar to the collapsing pipeline above, a MBS was considered in this intersection (i.e., shared by both tools), if its TargetScan prediction and miRanda prediction share all the four properties above.

Additional Figure 35. Criteria for collapsing miRanda predictions and also for determining the intersection of TargetScan and (collapsed) miRanda predictions.

We updated these methodological details to Methods (Page 15, Lines 11-24, Page 16, Lines 1-24 in main text; Pages 40-41 in supplementary material).

Update of MBS-related results, Figure 6b (now renumbered as Fig. 5b)

We note that we had mistakenly considered all non-REE edits as "no overlap" in the original code producing the original Figure 6b. Therefore, we re-computed the edit effects on MBS across all edits in normal early stage samples, and plotted the old Figure 6b again. It turned out that, while REEs do not distinguish themselves from general edits in the proportion of MBS-overlapping edits (Additional Fig. 36a and Supplementary Fig. 14), they did display a strong preference for MBS-gaining edits (Additional Fig. 36b and the new Fig. 6b (renumbered as Fig. 5b)). We have also updated the corresponding text and figures (Page 7, Lines 20-22, Page 28, Lines 1, 3-4, Page 29, Lines 1-2 in main text; Page 35 in supplementary material).

a

All edits

REEs

b

All MBS-overlapping edits

MBS-overlapping REEs

P-value for chi-square test on frequency of MBS-gaining edits between MBS-overlapping REEs and all MBS-overlapping edits, BH-adjusted

c

Additional Figure 36. Ratio of MBS-altering edits in 3'-UTR REEs and all 3'-UTR edits, based on the intersection of TargetScan and miRanda (a and b) or TargetScan only (c, which is also the previous uncorrected Figure 6b). (a) and (c), with no-overlap edits considered. (b), without no-overlaps considered. In (b), the BH-adjusted p -values for the chi-square test on the frequency of MBS-gaining edits between MBS-overlapping REEs and all MBS-overlapping edits were also shown.

We also noted that the number of MBS-overlapping REEs dropped to a very low ratio (Additional Fig. 36a, compared with Additional Fig. 36c (which is also the previous uncorrected Figure 6b)), which might arise from the poor intersection between TargetScan and miRanda predictions (Additional Fig. 37). Because the overlap between TargetScan and miRanda predictions is not our main focus, we did not put this figure into our manuscript.

Additional Figure 37. Overlaps between TargetScan predictions and (collapsed) miRanda predictions on all unedited transcripts. Each prediction is a 4-tuple of (transcript ID, miRNA family ID (equivalent to sequence of seed region), the start coordinate of the seed region alignment on the transcript, the end coordinate of the seed region alignment on the transcript).

Validation of MBS-related results, the old Figure 6c (now renumbered as Fig. 5c)

For the old Figure 6c, we successfully replicated the statistical significance of observing more MBS- gaining REEs than MBS-losing REEs per gene per sample, though (again) we did not reproduce the 1 median of the MBS gain group (Additional Fig. 38a and the new Fig. 6c (now renumbered as Fig. 5c)). This failure of reproducing the median might be due to (again) a decrease in the total number of MBS identified after taking the intersection.

We updated the figure accordingly (Page 7, Line 23; Page 28, Line 1 in main text).

a

b

Additional Figure 38. Plots for the count of REEs per (gene, sample) that affect MBS's in different ways, based on the intersection of TargetScan and miRanda (a, which is also the new Figure 5c) or TargetScan predictions only (b, which is also the previous Figure 6c). The p -value was derived from one-tailed Wilcoxon's Rank Sum test, with the alternative hypothesis being that median value for the

"MBS gain" group would be greater than that for the "MBS lost" group.

Validation of MBS-related results, the old Figure 6d (now renumbered as Fig. 5d)

and the old Supplementary Figure 5 (now Supplementary Fig. 15)

For the old Figure 6d, we successfully replicated the statistical significance of target genes of maternal mRNA clearance (the "decay at 8-cell" group) obtaining more MBS-gaining REEs than are other maternal genes (and also compared to the baseline, where there're no MBS-gaining REEs) (Additional Fig. 39a and the new Fig. 6d (now renumbered as Fig. 5d)).

We updated the figure accordingly (Page 8, Line 4; Page 28, Line 1 in main text).

a

b

Additional Figure 39. Barplots for distribution of count of MBS-gaining REEs per (gene, sample) on target genes of maternal mRNA clearance (denoted as "decay at 8-cell"), on other maternal genes (denoted as "others"), and on the expected baseline which is just a value of 0 (i.e., no MBS-gaining REEs would be found), based on the intersection of TargetScan and miRanda (A, which is also the new Figure 5d) or TargetScan predictions only (B, which is also the previous Figure 6d). All *p*-values were unadjusted and were derived from one-tailed Wilcoxon's Rank Sum test, with the alternative hypothesis being that the median value for the left (decay at 8-cell) group would be greater than that for the right (others/baseline) group.

When we examined the net change, however, we failed to reproduce the statistical significance (Additional Fig. 40a and the new Supplementary Fig. 5 (now renumbered as Supplementary Fig. 15)).

We updated the figure accordingly (Page 8, Lines 4-5, Page 28, Line 1, Page 29, Lines 6-7 in main text; Page 36 in supplementary material).

a

Net change of MBS by REE on the gene (defined as # MBS-gaining REE - # MBS-losing REE)

<=-3	1
-2	2
-1	>=3
0	

b

Additional Figure 40. Barplots for distribution of "count of MBS-gaining REEs minus count of MBS-losing REEs" per (gene, sample) on target genes of maternal mRNA clearance (denoted as "decay at 8-cell"), on other maternal genes (denoted as "others"), and on the expected baseline which is just a value of 0 (i.e., no MBS-gaining REEs would be found), based on the intersection of TargetScan and miRanda (a, which is the new Supplementary Figure 5) or TargetScan predictions only (b, which is also the previous Supplementary Figure 4). All p -values were unadjusted and were derived from one-tailed Wilcoxon's Rank Sum test, with the alternative hypothesis being that the median value for the left (decay at 8-cell) group would be greater than that for the right (others/baseline) group.

This failure of reproducing the median might be due to (again) a decrease in the total number of MBS identified after taking the intersection.

15) Figure 6d: Where did the authors obtain the genes that "decay at 8-cell" from? I don't see any gene expression analysis. (Similar to comment #9 above.)

We thank the reviewer for raising this confusion of lack of methodological details.

We are sorry for not making this clear. These genes are genes that undergo maternal clearance, and the way we obtain these genes had been described in the "Annotation of maternal genes and targets of maternal mRNA clearance" Subsection of Methods Section (Page 17, Lines 20-24 in main text).

Minor comments:

16) In the introduction, the authors wrote "The successful development of human embryos is based on a well-regulated network that spans multiple omic layers." It is unclear what "multiple omic layers" mean.

We thank the reviewer for pointing out our lack of clarity in writing here.

We had previously decided to express the idea that the embryonic development must be well-regulated at multiple levels, including (but not limited to) the DNA and chromatin level (e.g., DNA duplication and DNA repair), the RNA level (e.g., transcriptional regulation by transcription factors, alternative splicing, posttranscriptional modification, and RNA degradation), and protein level (e.g., translation, posttranslational modification, and protein degradation). To quickly introduce the major topic of our story (role of A-to-I editing in human early embryonic development), we have decided to switch to the phrase "the stringent gene regulation across the central dogma" (Page 2, Lines 14-15 in main text).

17) Figure 3a is redundant.

We thank the reviewer for the comment on the potentially misleading presentation of the key items to analyze in Figure 3 and Figure 4 (now renumbered as Fig. 2 and 3, respectively).

We assumed that the redundancy the reviewer refers to is between Figure 3a (now renumbered as Fig. 2a) and Figure 4a (now Fig. 3a). These two subfigures refer to different levels of editing patterns to examine. In the old Figure 3a, we focus on an edit that is REE, i.e., observed in $\geq 50\%$ of samples in a given stage. In the old Figure 4a, we focus on a gene that is frequently targeted by REEs, i.e., targeted by at least one REE in $\geq 80\%$ samples of a given stage. Therefore the old Figure 3a is not redundant.

We revised the old Figure 3a and the old Figure 4a to better distinguish them from each other (now renumbered as Fig. 2a and 3a, respectively; Page 22, Line 1, and Page 24, Line 1 in main text).

18) At the end of the results section, the authors wrote that "This hypothesis was immediately validated by the observation that ..." I suggest replacing "immediately validated" with "supported". The authors did not perform any independent experimental validations.

We thank the reviewer for the comment on our overclaim. We

have revised accordingly (Page 8, Line 3 in main text).

19) In the discussion, the authors wrote that "The role of A-to-I RNA editing in human has been ambiguous for a long time." This is not true. Multiple papers have clarified the diverse roles of editing in human.

We thank the reviewer for pointing out our confusing and misleading writing here.

We are sorry for using the incorrect word ("ambiguous") here. What we'd like to stress here is that (as stated in the succeeding sentence) the role of RNA editing in human embryonic development is largely unclear, in contrast to those in adult human tissues. We have deleted this sentence (Page 8, Line 16 in main text).

20) In the methods section, the authors wrote "For single-cell RNA-Seq datasets, we required that the sequencing technology not be based on cell barcoding." Can the authors explain why?

We thank the reviewer for reminding us of the lack of necessary explanation in this methodological setting.

As stated in the subsection "Compilation of human embryonic RNA-seq datasets" of Methods, We only considered datasets with paired-end sequencing (except for the dataset GSE36552 whose RNA editome has been studied [13,14]) to ensure more accurate sequence alignment [1]. Therefore, we excluded these cell-barcoded datasets, because they are essentially single-ended RNA-seq sequencing for transcripts; the other end is used for barcoding cells and contains no information on transcript sequences. We have added this explanation to Methods (Page 12, Lines 8-10 in main text).

21) One of the supplementary datasets (the largest one) appears to be filled with nonsensical characters.

Please provide a new version of Supp Table 6; it appears to be corrupted to editors as well.

We thank the reviewer and the editor for the reminder of the corruption of one of the supplementary files.

It is possible that the previous Supplementary Table 6 (the list of recoding edits in each sample), whose file size is about 80MB, is too large to be correctly handled by the manuscript submission system. We decided to replace this table (now renumbered as Supplementary Table 5) with the list of recoding edits without sample-specific information, which is much smaller (237KB in gzipped format). Because this table is very small, we did not include it in the Zenodo repository; instead, we put it in the GitHub repository (<https://github.com/gao-lab/HERE/blob/master/report.ver2/210215-sixth-dataset/201221-fifth-phenotype-collection/all.normal.samples/normal.recoding.edits.dt.csv.gz>).

22) There are scattered English language errors. Please proofread.

Please carefully proofread the manuscript for clarity and grammar. If you would like the assistance of paid editing services to do this, we can recommend our affiliates, Nature Research Editing Service and American Journal Experts. However, please note that use of an editing service is neither a requirement nor a guarantee of publication. Free assistance is available from our resources page.

We thank the reviewer and the editor for the suggestion of proofreading our manuscript. We have proofread the text carefully and corrected errors.

Remarks to the Author: Reproducibility

Please see comments under "Strength of the claims", for example, comments #13 and #14. We

thank the reviewer for their effort in helping better our manuscript.

We have replied to the corresponding comments above.

References

1. Lo Giudice C, Tangaro MA, Pesole G, Picardi E. Investigating RNA editing in deep transcriptome datasets with REDIttools and REDlportal. *Nat Protoc.* 2020;15:1098–131.
2. 1000 Genomes Project Consortium, Auton A, Brooks LD, Durbin RM, Garrison EP, Kang HM, et al. A global reference for human genetic variation. *Nature.* 2015;526:68–74.
3. Karczewski KJ, Francioli LC, Tiao G, Cummings BB, Alföldi J, Wang Q, et al. The mutational constraint spectrum quantified from variation in 141,456 humans. *Nature.* 2020;581:434–43.
4. Phan L, Jin Y, Zhang H, Qiang W, Shekhtman D, Shao D, et al. ALFA: Allele Frequency Aggregator [Internet]. National Center for Biotechnology Information, U.S. National Library of Medicine; 2020. Available from: <https://www.ncbi.nlm.nih.gov/snp/docs/gsr/alfa/>
5. Levanon EY, Hallegger M, Kinar Y, Shemesh R, DjinoVIC-Carugo K, Rechavi G, et al. Evolutionarily conserved human targets of adenosine to inosine RNA editing. *Nucleic Acids Res.* 2005;33:1162–8.

6. Shtrichman R, Germanguz I, Mandel R, Ziskind A, Nahor I, Safran M, et al. Altered A-to-I RNA editing in human embryogenesis. *PLoS One*. 2012;7:e41576.
7. Hu X, Wan S, Ou Y, Zhou B, Zhu J, Yi X, et al. RNA over-editing of BLCAP contributes to hepatocarcinogenesis identified by whole-genome and transcriptome sequencing. *Cancer Lett*. 2015;357:510–9.
8. Leng L, Sun J, Huang J, Gong F, Yang L, Zhang S, et al. Single-Cell Transcriptome Analysis of Uniparental Embryos Reveals Parent-of-Origin Effects on Human Preimplantation Development. *Cell Stem Cell*. 2019;25:697–712.e6.
9. Wu J, Xu J, Liu B, Yao G, Wang P, Lin Z, et al. Chromatin analysis in human early development reveals epigenetic transition during ZGA. *Nature*. 2018;557:256–60.
10. Kim DD, Kim TT, Walsh T, Kobayashi Y, Matisse TC, Buyske S, et al. Widespread RNA editing of embedded alu elements in the human transcriptome. *Genome Res*. 2004;14:1719–25.
11. Tan MH, Li Q, Shanmugam R, Piskol R, Kohler J, Young AN, et al. Dynamic landscape and regulation of RNA editing in mammals. *Nature*. 2017;550:249–54.
12. Liu G, Kong L, Baweja R, Ba D, Saunders EFH. Gender disparity in bipolar disorder diagnosis in the United States: A retrospective analysis of the 2005-2017 MarketScan Commercial Claims database. *Bipolar Disord*. 2022;24:48–58.
13. Qiu S, Li W, Xiong H, Liu D, Bai Y, Wu K, et al. Single-cell RNA sequencing reveals dynamic changes in A-to-I RNA editome during early human embryogenesis. *BMC Genomics*. 2016;17:766.
14. Li T, Li Q, Li H, Xiao X, Ahmad Warraich D, Zhang N, et al. Pig-specific RNA editing during early embryo development revealed by genome-wide comparisons. *FEBS Open Bio*. 2020;10:1389–402.
15. Hinrichs AS, Karolchik D, Baertsch R, Barber GP, Bejerano G, Clawson H, et al. The UCSC Genome Browser Database: update 2006. *Nucleic Acids Res*. 2006;34:D590–8.

16. Ramaswami G, Zhang R, Piskol R, Keegan LP, Deng P, O'Connell MA, et al. Identifying RNA editing sites using RNA sequencing data alone. *Nat Methods*. 2013;10:128–32.
17. Yanez LZ, Han J, Behr BB, Pera RAR, Camarillo DB. Human oocyte developmental potential is predicted by mechanical properties within hours after fertilization. *Nat Commun*. 2016;7:10809.
18. Ma J, Fukuda Y, Schultz RM. Mobilization of Dormant Cnot7 mRNA Promotes Deadenylation of Maternal Transcripts During Mouse Oocyte Maturation. *Biol Reprod*. 2015;93:48.
19. Chalabi Hagkarim N, Grand RJ. The Regulatory Properties of the Ccr4-Not Complex. *Cells*. 2020;9.
20. Fasken MB, Morton DJ, Kuiper EG, Jones SK, Leung SW, Corbett AH. The RNA Exosome and Human Disease. *Methods Mol Biol*. 2020;2062:3–3.
21. Wu D, Dean J. EXOSC10 sculpts the transcriptome during the growth-to-maturation transition in mouse oocytes. *Nucleic Acids Res*. 2020;48:5349–65.
22. Petit FG, Jamin SP, Kernanec PY, Becker E, Halet G, Primig M. EXOSC10/Rrp6 is essential for the eight-cell embryo/morula transition. *Dev Biol*. 2022;483:58–65.
23. Vieux KF, Clarke HJ. CNOT6 regulates a novel pattern of mRNA deadenylation during oocyte meiotic maturation. *Sci Rep*. 2018;8:6812.
24. Du H, Chen C, Wang Y, Yang Y, Che Z, Liu X, et al. RNF219 interacts with CCR4-NOT in regulating stem cell differentiation. *J Mol Cell Biol*. 2020;12:894–905.
25. Wahle E, Winkler GS. RNA decay machines: deadenylation by the Ccr4-not and Pan2-Pan3 complexes. *Biochim Biophys Acta*. 2013;1829:561–70.
26. Bohn JA, Van Etten JL, Schagat TL, Bowman BM, McEachin RC, Freddolino PL, et al. Identification of diverse target RNAs that are functionally regulated by human Pumilio proteins. *Nucleic*

Acids Res. 2018;46:362–86.

27. Bakheet T, Hitti E, Khabar KSA. ARED-Plus: an updated and expanded database of AU-rich element-containing mRNAs and pre-mRNAs. *Nucleic Acids Res.* 2018;46:D218–20.
28. Geissler R, Simkin A, Floss D, Patel R, Fogarty EA, Scheller J, et al. A widespread sequence-specific mRNA decay pathway mediated by hnRNPs A1 and A2/B1. *Genes Dev.* 2016;30:1070–85.
29. Braun JE, Huntzinger E, Fauser M, Izaurralde E. GW182 proteins directly recruit cytoplasmic deadenylase complexes to miRNA targets. *Mol Cell.* 2011;44:120–33.
30. Fabian MR, Cieplak MK, Frank F, Morita M, Green J, Srikumar T, et al. miRNA-mediated deadenylation is orchestrated by GW182 through two conserved motifs that interact with CCR4-NOT. *Nat Struct Mol Biol.* 2011;18:1211–7.
31. Chekulaeva M, Mathys H, Zipprich JT, Attig J, Colic M, Parker R, et al. miRNA repression involves GW182-mediated recruitment of CCR4-NOT through conserved W-containing motifs. *Nat Struct Mol Biol.* 2011;18:1218–26.
32. Roth SH, Levanon EY, Eisenberg E. Genome-wide quantification of ADAR adenosine-to-inosine RNA editing activity. *Nat Methods.* 2019;16:1131–8.
33. Sha QQ, Zheng W, Wu YW, Li S, Guo L, Zhang S, et al. Dynamics and clinical relevance of maternal mRNA clearance during the oocyte-to-embryo transition in humans. *Nat Commun.* 2020;11:4917.
34. Vassena R, Boué S, González-Roca E, Aran B, Auer H, Veiga A, et al. Waves of early transcriptional activation and pluripotency program initiation during human preimplantation development. *Development.* 2011;138:3699–709.
35. Reyes JM, Silva E, Chitwood JL, Schoolcraft WB, Krisher RL, Ross PJ. Differing molecular response of young and advanced maternal age human oocytes to IVM. *Hum Reprod.* 2017;32:2199–208.

36. Warner CM, Versteegh LR. In vivo and in vitro effect of alpha-amanitin on preimplantation mouse embryo RNA polymerase. *Nature*. 1974;248:678–80.

37. Enright AJ, John B, Gaul U, Tuschl T, Sander C, Marks DS. MicroRNA targets in *Drosophila*.

Genome Biol. 2003;5:R1.

Reviewer comments:

Reviewer #1 (Remarks to the Author: Overall significance):

In this revised manuscript Ding. et,al. performed additional analysis and provided more details in Methods and additional notes. In general they provided elaborative response for each comment, and adjust their conclusions in result and discussion part to make the manuscript more reliable. Here several questions remained to be solved.

Reviewer #1 (Remarks to the Author: Strength of the claims):

1. In Fig1e for the identification of normal and abnormal points, as they were defined from multiple samples, will they be calculated when appeared once or they will need to appear in $\geq 20\%$ of indicated samples?
2. As shown in Additional Figure 2b, more 3'-UTR/intron-enriched in overlapped sites, is it because the overlapped sites were more reliable? How about the unique pattern in 8cell and morula stages?
3. It is a pity that GSE95477 could not provided the solidity of the conclusions between A-I editing induced MBS and maternal RNA clearance. In fact I could not understand the implication of the P-value between the correlation coefficient between REE number and gene expression in young and old mothers. The correlation coefficient in young mother is almost near 0 which means REE is not related with gene expression, how they could conclude that more REEs on a given maternal gene is more likely to lead to lower FPKM ? Similarly, the cross-stage correlation in Additional Figure 6 could hardly support anything in my view.
4. In general the analysis results in Fig 4c,4e and Additional Figure 5-6 are wanting and could hardly support the indicating conclusions they made. Therefore it is better to tone down and keep them in discussion part.
5. The correlation score in Sup fig8, which is mostly between -0.2 and -0.08, indicating very weak negative correlation, and the author need to adjust their statement.
6. The manuscript could be improved after substantial editing for both language and logical structure. Particularly the individual pieces of results were not tightly strung together in a very

logical fashion. One of the examples is in Page6 line24 “other REEs also undergo.....”, what is the meaning of other REE in this sentence? Why they take the loss of one REE but not the reduced amount of REEs in Fig 4a, as a logical reason to propose a systematic loss of REEs?

Reviewer #1 (Remarks to the Author: Reproducibility):

For the Reproducibility part. In fact the author only have to compare RNA editing characters defined in their study with published results in Qiu et al. and show some data like the number, genome distribution and specific preference like Alu, and explain if there are any discrepant results. If they want to be more strict they can analysis the RNA-seq data provided, but this is essential, and there is also no need to run their pipeline. PS.: The IGV of A-I mismatched site on Alu can be shown here.

Reviewer #2 (Remarks to the Author: Overall significance):

My comments on overall significance and impact are similar to my previous comments. In my opinion, the authors have adequately addressed all reviewers' comments with additional information.

Reviewer #3 (Remarks to the Author: Overall significance):

The authors submitted a revised manuscript on their characterization of the A-to-I RNA editome in human embryos. They have made considerable efforts to address the previous reviewer comments. Nevertheless, there are still areas that need to be addressed before the work can be published in Communications Biology.

Major comments:

1) The various distributions of editing levels (Supplementary Figure 4) look odd. Editing level distributions are typically heavily right-skewed. But the authors' plots show numerous sites with high editing levels. There is also an abnormal peak at an editing rate of 1 in all the plots, which strongly suggests the presence of artifacts (rightly acknowledged in the figure legend). As a sanity check, can the authors also examine published RNA-seq on zebrafish development (PMID: 33872356) and see if they have observed similar editing level distributions?

2) Show how the A>G percentage changes with each filter imposed. Can the authors also provide a more detailed analysis flowchart as an overview? Figure 1B is sparse in details and does not contain many of the filters used.

3) There are 2920 sites identified in Qiu et al but not in the current manuscript - why?

4) The overlap between normal and abnormal embryos, even at matched stages, is relatively low. This could be either real biology or noise. Sanity check: Randomly split normal embryos only into two sets and check extent of overlap. Repeat multiple times to get error bar. The authors can also do the same for abnormal embryos.

5) My previous comment #2 - show distribution of all 12 possible types of nucleotide changes (A-to-C, A-to-G, A-to-T etc) - has not been satisfactorily addressed. Supplementary Figure 1 is not the standard way of presenting the distribution (what are the FP/TP in the figure anyway?). Please present in the same way as other RNA editing papers.

6) The authors have defined FP wrongly (authors' response to my previous comment #3). I suggest them to look up how FP is defined in other RNA editing papers.

7) AEI is odd. Editing in hESCs is known to be generally low. But yet, in authors' AEI plot (Supplementary Figure 20), the AEI in hESCs is similar to that in embryonic samples, where there are supposed to be numerous recurrent edits. This suggests analysis artifacts. In addition, AEI often correlates reasonably well with ADAR1 expression, which the authors do not observe.

8) The authors did not address my previous comment #9 satisfactorily. (There is a similar comment from reviewer 1 as well). A big claim in the manuscript is that recurrent edits generate miRNA-binding sites, leading to clearance of maternal transcripts. To support the claim, the authors must show the expression profiles of such genes (e.g., in the form of a heatmap). Instead, the authors tried to skirt the issue and produced plots with "editing level in previous stage" on the x-axis and "expression in the current stage" on the y-axis. Firstly, the Spearman correlation coefficients do not look good in general. Secondly and more importantly, those plots are not supporting evidence of the authors' claim. The authors should not be comparing across genes. So what if Gene A with higher edits in previous stage and lower expression in current stage than Gene B? The expression of Gene A could be constantly lower all the time! Instead, I think readers would want to see that RNA transcripts with recurrent edits indeed have a drop in their abundance as development progresses.

Response to reviewers' comments

We thank the reviewers for their further valuable comments, and have revised the manuscript accordingly. Specifically,

1. We moved the analysis of change in total REE counts for abnormal embryos and embryos from elder mothers to Discussion and Supplementary Notes 7 (and Fig. 4c-e to Supplementary Fig. 26-27) because of a lack of strong evidence;
2. We added to Results a brief description that our observation is also consistent with a previous pilot study on human early embryos by Qiu et al.¹ in terms of a high proportion of A-to-G mismatches and a high proportion of Alu edits among all edits, as well as an updated set of AEI analysis results (**Supplementary Figures 20 and 21**) and an additional layer of evidence supporting that transcripts with REEs did have a drop in median abundance as development progresses (**Supplementary Figures 29 and 30**);
3. We revised the text and figures to make them clear and precise; in particular, we provided a flowchart for the identification of A-to-I RNA editing (**Supplementary Figure 28**), avoided inappropriately using the terms “false positive”, “true positive”, and “false negative” for the identification, and re-plotted the **Supplementary Figure 1** to make its presentation consistent with previous studies.

All the changes in the manuscript and supplementary material are high- lighted in blue font. Please see the replies to the specific reviewer comments below for further details. We thank again for the editors and reviewers' efforts.

Reviewer comments:

Reviewer #1 (Remarks to the Author: Overall significance):

In this revised manuscript Ding. et.al. performed additional analysis and provided more details in Methods and additional notes. In general they provided elaborative response for each comment, and adjust their conclusions in result and discussion part to make the manuscript more reliable. Here several questions remained to be solved.

We thank the reviewer for the acknowledgement of our previous response.

Reviewer #1 (Remarks to the Author: Strength of the claims):

1. In Fig1e for the identification of normal and abnormal points, as they were defined from multiple samples, will they be calculated when appeared once or they will need to appear in $\geq 20\%$ of indicated samples?

We thank the reviewer for pointing out this methodological detail.

As stated in Step (12) of **Supplementary Notes 6**, edits located in Alu elements will be included in the summary in Fig. 1e as long as detected in at least one sample, while edits located not in Alu elements will be included only if detected in at least two normal samples (or two abnormal samples) of the same stage (see **Supplementary Table 8** for the details of each stage). We made this detail more prominent by repeating it in the Methods section of the main text (Page 14, Lines 326-330 in main text).

2. As shown in Additional Figure 2b, more 3'-UTR/intron-enriched in overlapped sites, is it because the overlapped sites were more reliable? How about the unique pattern in 8cell and morula stages?

We thank the reviewer for helping us further dissecting the possible causes of such statistical significance before determining its biological importance.

We first put the previous Additional Figure 2b (renumbered as **Additional Fig. 1**) below for readability.

Additional Fig. 1. (The previous Additional Figure 2b) The deviation of 3'-UTR-or-intron edit ratio of the overlap edits from that of the unique edits. Only stages with both types of samples were shown. All p -values were from chi-square test and BH-adjusted.

For the first question, we noted that the more reads covered on a given

editing site, the more reliable its identification (as an A-to-I editing with an editing level) would be. In particular, being covered by at least 10 reads is a prerequisite for a candidate site to be identified as an A-to-I edit (Step (11) in **Supplementary Notes 6**). Therefore, we addressed this question preliminarily by examining whether in general the overlap edits were more deeply sequenced than were the unique edits.

We found that the overlap edits do have a much higher sequencing depth compared to the unique edits (**Additional Fig. 2**; also the new **Supplementary Fig. 2C**) which is statistically significant in all comparisons (i.e., all unadjusted p -values are less than 2.2^{-16} for Wilcoxon's Rank Sum test, with the null hypothesis being that the median sequencing depth of the overlap edits is equal to or less than that of the normal- or abnormal-only edits in a given stage). Therefore, the genomic enrichment in 3'-UTR/intron for the overlap edits might be at least partially due to a better reliability of these sites.

Additional Fig. 2. The distribution of sequencing depth on the overlap edits and the unique edits.

For the second question, we assumed that the “unique pattern” the re-viewer refers to is the drop of $-\log_{10}(p)$ in 8-cell and morula stages. We currently do not have a clear explanation for this unique pattern. Apart from a possible loss of a stable editing pattern indicated by REE upon entry

into the 8-cell stage (Fig. 2b), other possible technical and biological factors (e.g., the end of maternal mRNA clearance² and the wave of zygotic genomic activation³) remain to be investigated.

We added these results and discussion to **Supplementary Figure 2** and its legend (Page 17, Lines 331-341 in Supplementary Information).

3. It is a pity that GSE95477 could not provide the solidity of the conclusions between A-I editing induced MBS and maternal RNA clearance. In fact I could not understand the implication of the P-value between the correlation coefficient between REE number and gene expression in young and old mothers. The correlation coefficient in young mother is almost near 0 which means REE is not related with gene expression, how they could conclude that more REEs on a given maternal gene is more likely to lead to lower FPKM? Similarly, the cross-stage correlation in Additional Figure 6 could hardly support anything in my view.

We thank the reviewer for pointing out this mistake.

For clarity, we first put the previous Additional Figure 5 and 6 (renumbered as **Additional Fig. 3** and **Additional Fig. 4**, respectively) below.

Additional Fig. 3. (The previous Additional Figure 5) Distribution of Spearman's correlation coefficient between the number of REE-matching ed-its identified on a gene, and its FPKM in the same sample. The sample groups are consistent with those in Figure 4e. All p -values were unadjusted and were derived from one-tailed Wilcoxon's Rank Sum tests, with the alternative hypothesis being that the median of the correlation from young mothers is smaller than that from old mothers.

Additional Fig. 4. (The previous Additional Figure 6) Distribution of Spearman's correlation coefficient between the number of REE-matching edits identified on a gene in an oocyte (GV), and its FPKM in the oocyte (MII) of the paired patient. All p -values were unadjusted and were derived from one-tailed Wilcoxon's Rank Sum tests, with the alternative hypothesis being that the median of the correlation from young mothers is larger than that from old mothers.

As the reviewer has commented, we have previously mistaken the drop in the correlation (between the number of REE-matching edits and gene expression) for the negative association between the number of REE-matching edits and gene expression. Unfortunately, we currently cannot arrive at any

solid, biologically meaningful conclusion from the results.

Because these results were not put into the manuscript in the previous response, we did not make any new revision concerning this problem to our manuscript.

4. In general the analysis results in Fig 4c, 4e and Additional Figure 5-6 are wanting and could hardly support the indicating conclusions they made. Therefore it is better to tone down and keep them in discussion part.

We thank the reviewer for pinpointing our overclaim.

We have toned down the part covering Fig. 4c and 4e, as well as the title of this result subsection (we did not include the previous Additional Figures 5 and 6 into our manuscript, as described in the reply to the third question of the same reviewer above), and moved it to Discussion and **Supplementary Notes 7** (Page 7, Lines 150, Page 12, Lines 277-279, Page 19, Lines 465 and 468 in main text; Page 3, Lines 46 and 58, Pages 13-14, Lines 287-314, Page 32, Line 420, Page 34, Line 430, Pages 46-48, Lines 518-539 in Supplementary Information).

5. The correlation score in Sup fig8, which is mostly between -0.2 and -0.08, indicating very weak negative correlation, and the author need to adjust their statement.

We thank the reviewer for pointing out the fact that the correlation is weak, and we have revised the text accordingly (Page 6, Lines 132-133 in main text).

6. The manuscript could be improved after substantial editing for both language and logical structure. Particularly the individual pieces of results were not tightly strung together in a very logical fashion. One of the examples is in Page 6 line 24 “other REEs also undergo……”, what is the meaning of other REE in this sentence? Why they take the loss of one REE but not the reduced amount of REEs in Fig 4a, as a logical reason to propose a systematic loss of REEs?

We thank the reviewer for the suggestion on editing language and logical structure, and particularly on the case discussed.

The “other REEs” are those REEs that are not in the 107 edits (Supplementary Table 2) analyzed in Fig. 4a and 4b. The reason why we had

proposed [the possible existence of] a systematic loss of REE-matching edits based on the loss of one [or several] REE-matching edits (rather than the reduced amount of REE-matching edits) in Fig. 4a is: the (almost complete) loss of some REE-matching edits in certain abnormal embryos (or embryos from elder mothers) might be the extreme cases – the most affected REE-matching edits – of a wave of systematic loss of many (if not most) REE-matching edits in these embryos.

We added this explanation to this part. As discussed above in the fourth question of the same reviewer, we have also moved it to Discussion and **Supplementary Notes 7**.

Reviewer #1 (Remarks to the Author: Reproducibility):

For the Reproducibility part. In fact the author only have to compare RNA editing characters defined in their study with published results in Qiu et al. and show some data like the number, genome distribution and specific preference like Alu, and explain if there are any discrepant results. If they want to be more strict they can analysis the RNA-seq data provided, but this is essential, and there is also no need to run their pipeline. PS.: The IGV of A-I mismatched site on Alu can be shown here.

We thank the reviewer for clarifying the major analyses we need to perform, and below we detail the comparisons one by one.

For the number of edits identified: this has been analyzed extensively in the previous reply to this question. Briefly, an initial comparison of total edits identified revealed a poor overlap (5,893 edits) between ours (with 70,980 exclusively identified edits) and theirs reported in Table S1 of Qiu et al. (with 2,920 exclusively identified edits), and after re-run their pipeline we arrived at the conclusion that most of the difference could be attributed to the choice of different filters and algorithms.

For the genomic distribution of edits, both we and Qiu et al. observed that most (~75% by us (**Supplementary Fig. 2a**) and ~80% by Qiu et al.) edits are located in 3'-UTR and intron, and both studies observed the extremely strong Alu-preference (boxplots of gametes and pre-implantation embryos in **Fig. 1g** of ours, and the 97.84% percentage reported by Qiu et al.). We updated these comparison to main text (Page 5, Lines 113-116 in main text).

As Qiu et al. did not provide any IGV track (or other Genome Browser- style track) to compare with, we picked the *TTF1* gene as an example and plotted the IGV track of some of *TTF1*'s Alu edits identified by Qiu et al. and/or us (**Additional Fig. 5**). We did not include it into our manuscript.

Additional Fig. 5. An example IGV plot-based comparison of Alu edits identified by Qiu et al. and by us.

Reviewer #2 (Remarks to the Author: Overall significance):

My comments on overall significance and impact are similar to my previous comments. In my opinion, the authors have adequately addressed all reviewers' comments with additional information.

We thank again the reviewer for all the efforts and valuable suggestions.

Reviewer #3 (Remarks to the Author: Overall significance):

The authors submitted a revised manuscript on their characterization of the A-to-I RNA editome in human embryos. They have made considerable efforts to address the previous reviewer comments. Nevertheless, there are still areas that need to be addressed before the work can be published in Communications Biology.

We thank the reviewer for the acknowledgement of our previous response.

Major comments:

1) The various distributions of editing levels (Supplementary Figure 4) look odd. Editing level distributions are typically heavily right-skewed. But the authors' plots show numerous sites with high editing levels. There is also an abnormal peak at an editing rate of 1 in all the plots, which strongly suggests the presence of artifacts (rightly acknowledged in the figure legend). As a sanity check, can the authors also examine published RNA-seq on zebrafish development (PMID: 33872356) and see if they have observed similar editing level distributions?

We thank the reviewer for suggesting a further examination on the distribution of editing levels.

The current presentation of the distribution of editing levels in the previous Supplementary Fig. 4 (also shown below as **Additional Fig. 6**) might be a little misleading due to the usage of a log₁₀-scale y-axis. We had initially chosen this scale to make the number of high-editing-level edits discernible. If we plot the number of edits per editing level bin as-is (i.e., no log₁₀-transformation) without considering those edits with their editing level being 1 (as acknowledged by us previously and pointed out by the reviewer) (**Additional Fig. 7**), in most stages (in particular, all gametes and pre-implantation embryos) the number of edits with high editing levels are not numerous any more, and the overall distribution in these stages becomes heavily right-skewed, as the reviewer has commented. We note that a peak close to 1 was still observed in some stages, particularly epiblasts and hypoblasts and less obviously in some blastocytes, ICM, CTB, STB, EVT.

Whether these are mostly non-biological artifacts or some truly biological signals involving RNA editing remains to be investigated.

Additional Fig. 6. (The previous Supplementary Figure 4) Distribution of editing level in each sample group (previous Supplementary Fig. 4, now Supplementary Fig. 4a). Note that although we have used a very stringent pipeline (see Methods), we still found a number of editing sites with an editing level of 1. These editing sites should be examined with caution. The number of editing sites (y-axis) is shown in log₁₀-scale.

Additional Fig. 7. Distribution of editing level (without taking into consideration those edits with an editing level of 1) in each sample group. The number of editing sites (y-axis) is shown as-is.

We renamed the previous Supplementary Fig. 4 as Supplementary Fig. 4a, and added **Additional Fig. 7** as Supplementary Fig. 4b to Supplementary Information (Pages 19-20, Lines 351, 353-354, 358-360 in Supplementary Information).

For the sanity check on zebrafish embryos, we noticed that the paper (PMID: 33872356⁴) did not release the full list of per-sample editing level of zebrafish edits identified; the only publicly available list is Supplementary Table S2, the mean editing level of 149 “coding sites that appeared in at least two out of the four samples”. We had asked the authors for the full list, but we have not received the list yet before submitting the revision; therefore, we used the 149 list for analysis. Examining this list did give a heavily right-skewed distribution (**Additional Fig. 8**).

Additional Fig. 8. Distribution of mean editing level in Supplementary Table S2 of Buchumenski et al..

2) Show how the A>G percentage changes with each filter imposed. Can the authors also provide a more detailed analysis flowchart as an overview? Figure 1B is sparse in details and does not contain many of the filters used.

We thank the reviewer for suggesting to make the pipeline more straight-forward to readers.

We note that we did not know for each candidate site which genomic strand would be transcribed until Step (13). Therefore, we drew a more detailed analysis flowchart with the per-filter A>G or T>C percentage (except for Step (13) where the A>G or “both A>G and T>C” percentage is used instead) of an example sample (GSM2706237) listed as **Supplementary Figure 28** (also shown below as **Additional Fig. 9**), and updated our manuscript accordingly (Page 13, Line 319 in main text; Page 6, Line 130, Pages 49-50, Lines 540-544 in Supplementary Information).

Additional Fig. 9. The more detailed analysis pipeline flowchart, with the per-filter A>G or T>C percentage (except for Step (13) where the A>G or “both A>G and T>C” percentage is used instead) of an example sample (GSM2706237) listed.

3) There are 2920 sites identified in Qiu et al but not in the current manuscript - why?

We thank the reviewer for commenting on this.

This has been addressed extensively in the previous reply (to the first question of the Reproducibility part of the first reviewer’s comments). Briefly,

1. Because Qiu et al. did not provide the per-sample state of their exclusive 2920 sites, we had to re-run their pipeline to find out why these edits were only identified by them.
2. We tried to reproduce Qiu et al.’s identification pipeline, but found out that this pipeline could not be fully reimplemented even after several rounds of communication with the authors.
3. We then used our best reimplementation results and made several attempts to explain the possible sources of Qiu et al.-exclusive edits. In the end we found the following reasons that can explain $\geq 75\%$ such edits in all samples:
 - (a) We discarded sites with multiple alternative alleles identified (explaining a median of 3.91% false negatives across samples);
 - (b) We used a more comprehensive set of population genotypes from worldwide cohorts (explaining 54.58%);
 - (c) We discarded sites with invalid sample occurrence (note that we required non-Alu sites to be identified at least twice in the same normal stage or the same abnormal stage, not across all samples; explaining 10.13%); and
 - (d) We discarded sites whose event on the known transcripts does not contain the A-to-G case (explaining 20.28%).

4)The overlap between normal and abnormal embryos, even at matched stages, is relatively low. This could be either real biology or noise. Sanity check: Randomly split normal embryos only into two sets and check extent of overlap. Repeat multiple times to get error bar. The authors can also do the same for abnormal embryos.

We thank the reviewer for this comment, and carried out the sanity check accordingly.

We first put the previous Additional Figure 21 (renumbered as **Additional Fig. 10**) below for readability.

Additional Fig. 10. (The previous Additional Figure 21) Sample pair scatterplot of the overlap ratio (defined as the number of edits identified in both samples divided by the number of edits identified in at least one of the samples) vs. the distance of sequencing depth (defined as the absolute difference between the \log_{10} (count of total bases sequenced)). Good/bad viab., predicted to be of good/bad viability (see the paper of GSE65481 for more details).

As shown in **Additional Fig. 10**, we had not compared the overlap between normal and abnormal samples at matched stages. Instead, for each sample group (defined by the combination of stage, whether normal or not, and (if the embryo is abnormal) the particular phenotype), we computed the overlap for each of all possible sample pairs within this sample group.

To confirm whether the overlap between normal and abnormal samples is still relative low at each of the matched stages, we computed the overlap ratio, and found that they are around 8-37%, either lower or higher than the 23% reported in Fig. 1e (**Additional Fig. 11**).

Additional Fig. 11. Overlap ratio between normal and abnormal samples at the level of matched stages.

Nevertheless, such overlap ratio is still low in absolute values; therefore, we carried out the sanity check suggested by the reviewer. By randomly splitting into two each of the normal and abnormal groups used in **Additional Fig. 10** and examining the percentage of overlap, we observed a low overlap (boxplots) that is a little lower than, but yet similar to, the overlap ratio between normal and abnormal samples (red dashed lines) in each group (**Additional Fig. 12**); the only exceptions are the zygote/abnormal/AG (or PG) cases, where the overlap ratio is even lower, possibly due to their small sample size (4 for both). Note that a statistical significance of difference is not important here, as most of the absolute ratio differences are small

compared to the overlap ratio themselves.

We then concluded that the low overlap ratio between normal and abnormal samples cannot be fully attributed to pure biological difference, because otherwise we should have observed a much higher overlap ratio in the within-normal/within-abnormal samples of random splits.

Additional Fig. 12. Boxplot of percentage-of-overlaps from ten different random splits for each group. The red dashed line for each stage denotes the normal-abnormal overlap ratio shown in **Additional Fig. 11**.

We did not include these analyses in our manuscript, as the overlap of general edits between normal and abnormal samples is not our major focus here.

5) My previous comment #2 - show distribution of all 12 possible types of nucleotide changes (A-to-C, A-to-G, A-to-T etc) - has not been satisfactorily addressed. Supplementary Figure 1 is not the standard way of presenting the distribution (what are the FP/TP in the figure anyway?). Please present in the same way as other RNA editing papers.

We thank the reviewer for pointing out our inappropriate way of presenting our results.

We surveyed paper and found that the standard way to represent such distribution is to plot a barplot, with one bar describing the number of edits for each type of nucleotide change (**Additional Fig. 13**)⁵. We then updated the Supplementary Fig. 1 accordingly (**Additional Fig. 14**) (Page 14, Line 317 in Supplementary Information).

In addition, we stopped using the FP/TP which brought about confusion (as described in the reply to the next comment of the same reviewer) in Fig. 1d (shown as **Additional Fig. 15** here; Page 36, Line 855 in main text) and Supplementary Fig. 1.

d

Additional Fig. 13. An example plot of all 12 types of nucleotide changes, adapted from Fig. 1d of Ramaswami et al., 2013.

Additional Fig. 14. The updated Supplementary Fig. 1.

Additional Fig. 15. The updated Fig. 1d.

6) The authors have defined FP wrongly (authors' response to my previous comment #3). I suggest them to look up how FP is defined in other RNA editing papers.

We thank the reviewer for pinpointing this misuse.

We surveyed papers for the definition of FP (False Positives) in identification of RNA A-to-I edits, and found that this was first described by the following references:

1. Pickrell et al., 2012⁶:

- (a) "This pattern [that mismatches to the genome at RDD (RNA-DNA differences) sites are dramatically enriched at the ends of RNA sequencing reads] is evidence that many of the RDD sites are false positives due to mapping or sequencing errors."

2. Lin et al., 2012⁷:

- (a) "Taken together, we identified 9037 (~89%) of 10,210 RDD sites that may be false positives derived from gene duplications, read-end misalignment, and/or SNPs."

3. Kleiman and Majewski, 2012⁸:

- (a) “Based on this analysis, we estimate that false positive results identified by their presence in only the first or last positions of sequencing reads, along with those that are supported only by unidirectional reads, account for more than half of the entire dataset (Table 1).”
- (b) “This specific case illustrates all the typical features of a false positive alignment and sequencing problem: (i) all the reads align in one direction only; (ii) the variant site is present at the extremity of the read; (iii) it is directly adjacent to another variant; and (iv) it flanks a splice junction and no supporting reads extend past the 5th nucleotide of the exon.”
- (c) “False positives are calculated using the tails of the distributions (where the RDD and control distributions intersect), as the number of RDD sites minus the expected number of sites [single-nucleotide polymorphisms (SNPs)].”

Consistent with the reviewer’s comments, it appears that the “false positive” was not defined as what we have defined (“those candidate variants from RNA-Seq that overlap with genomic variants”) in these references, but was defined as a wide range of different technical artifacts including read-end misalignments and genomic variants.

We thus decided to stop using “false positive/FP” in the identification of RNA editing and describe it explicitly, i.e., “those candidate variants from RNA-Seq that overlap with genomic variants”. Similarly, we decided to stop using “true positive/TP” in the identification of RNA editing and describe it explicitly, i.e., “those candidate variants from RNA-Seq that do not overlap with genomic variants”. We updated these to the manuscript (Page 5, Lines 96, 100-101, Page 11, Line 264, Page 14, Lines 333, 340, Page 37, Lines 858-859, Page 40, Line 888 in main text; Page 1, Line 10, Page 2, Lines 27-28, Page 20, Lines 356 (where we deleted the term) in Supplementary Information).

7) AEI is odd. Editing in hESCs is known to be generally low. But yet, in authors’ AEI plot (Supplementary Figure 20), the AEI in hESCs is similar to that in embryonic samples, where there are supposed to be numerous recurrent edits. This suggests analysis artifacts. In addition, AEI often

correlates reasonably well with ADAR1 expression, which the authors do not observe.

We thank the reviewer for this comment.

After carefully checking the result, we found that what we had plotted is signal-to-noise ratio in the original AEI paper⁹, not the AEI itself (the 'unnormalized AEI' in our results). When we switched to the AEI itself, we found that they are generally above 10, far higher than those in adult tissues reported by Roth et al.. To make our AEI estimates in embryos as much comparable to other reported AEI as possible, we eliminated biases at implementation level by re-running the official AEI software (<https://github.com/a2iEditing/RNAEditingIndexer>) on our realigned bam files with default parameters. This gave us the following distribution (**Additional Fig. 16**) and correlation with ADAR expression (**Additional Fig. 17**). We note that, during the recomputation of the correlation, we found that Roth et al. used a two-sided test⁹, while we had used the one-sided test previously; therefore, we switched to two-sided tests in the recomputation.

Additional Fig. 16. Distribution of AEI across different stages of embryonic samples, computed by the official AEI software.

Additional Fig. 17. Histogram of Spearman’s correlation coefficients between AEI and *ADAR* FPKM for each normal stage with ≥ 10 normal samples, computed by the official AEI software. All those that have a Benjamini-Hochberg-adjusted p -value for the correlation coefficient test (with the alternative hypothesis being that the correlation coefficient is not equal to 0) that is no less than 0.05 were considered not significant (NS).

Now we address the comments.

For the hESC comment, while its AEI was considerably lower than those in early embryos in the updated AEI distribution (**Additional Fig. 16**) and is consistent with the reviewer’s comment that “Editing in hESCs is known to be generally low”, we note that the generally low editing in hESC was not fully supported by previous studies. In particular, while some studies have found that the editing in hESCs is generally low in some genes

when compared to adult human tissues^{10–12}, they did not give a global (i.e., genome-wide) description of editing level (as AEI does) for hESCs, and they also found some other genes with a high editing level in hESCs; in fact, one of the major conclusion of Osenberg et al. is “We identified high editing levels of Alu repetitive elements in hESCs”¹⁰, conflicting with the reviewer’s comment. Whether the hESC editing level of these genes could represent the general editing pattern of hESC remains to be investigated, which is beyond the scope of this manuscript. We also examined all studies^{4,13–46} that cite the AEI paper⁹, but did not find any AEI reported in hESC.

For the correlation with ADAR1 expression, most of the correlations were still not statistically significant (**Additional Fig. 17**). However, the reviewer’s claim that “AEI often correlates reasonably well with ADAR1 expression” is not well-established, as Roth et al. has pointed out: “... However, more often than not, the correlation between ADAR expression and editing activity [in human adult tissues] is rather weak (Fig. 3b and Supplementary Figs. 6 and 7), presumably due to additional layers of ADARs regulation at the protein level.”⁹. Therefore, the poor correlation does not seem to be a sign of analysis artifacts.

We have updated **Additional Fig. 16** and **Additional Fig. 17** to **Supplementary Fig. 20 and 21**, respectively (Page 41, Line 476, Page 42, Line 480 in Supplementary Information), and have also updated the corresponding text describing how we computed the AEI (Page 2, Lines 32- 41, Page 43, Lines 484-485 in Supplementary Information).

8) The authors did not address my previous comment #9 satisfactorily. (There is a similar comment from reviewer 1 as well). A big claim in the manuscript is that recurrent edits generate miRNA-binding sites, leading to clearance of maternal transcripts. To support the claim, the authors must show the expression profiles of such genes (e.g., in the form of a heatmap). Instead, the authors tried to skirt the issue and produced plots with “editing level in previous stage” on the x-axis and “expression in the current stage” on the y-axis. Firstly, the Spearman correlation coefficients do not look good in general. Secondly and more importantly, those plots are not supporting evidence of the authors’ claim. The authors should not be comparing across genes. So what if Gene A with higher edits in previous stage and lower expression in current stage than Gene B? The expression of Gene A could be constantly lower all the time! Instead, I think readers would want to see that RNA transcripts with recurrent edits indeed have a drop in their abundance

as development progresses.

We thank the reviewer for this comment and the nice suggestion. Firstly, while we agree that the Spearman's correlation is poor (as discussed in the reply to the 5th comment of the first viewer), we note that we did not attempt to support the claim that "REEs are likely to enhance maternal mRNA clearance, a possible mechanism of which could be introducing more microRNA binding sites to the 3'-untranslated regions of clearance targets" (as stated in the previously revised Abstract) with the observation of correlation between "editing level in previous stage" and "expression in the current stage" (the previous Additional Fig. 7 and Supplementary Fig. 8, also shown as **Additional Fig. 18** below). This figure had been generated to address the sixth comment of Reviewer 1 in the previous reply, and we found that it would be more appropriate to regard it as a weak suggestion of a possible relationship between REE and the expression level of its targeted gene (no matter whether it drives the drop of expression level in the next stage or is just a byproduct of this drop) and thus its relationship with the embryonic development, rather than using it to support the claim above. Therefore, in the previous revision, we put it in the section "Detection of thousands of organized REEs throughout early embryonic development" as an auxiliary evidence (i.e. the Supplementary Fig. 8), and did not relate it to the claim above.

Additional Fig. 18. (The previous Additional Figure 7 / Supplementary Figure 8) Spearman's correlation coefficient between median editing level of a REE in a given stage and median FPKM of the gene hosting that REE in the latter stage. Only normal samples from early stages were considered. For each stage pair, only genes with FPKM > 0.001 were plotted. The trend of association is visualized by linear regression (with orange line denoting the fitted line and grey shadow denoting the 95% confidence interval). The p -value shown in each scatterplot is the Benjamini-Hochberg-adjusted p -value for two-sided correlation test (computed by R's `cor.test(method="spearman")`).

On the other hand, we thank the reviewer for providing a straightforward way to examine this correlation, and we examined this for each stage of observing REE separately (**Additional Fig. 19**). We note that it is infeasible to plot a heatmap for all tens of thousands of genes versus stages, because typical computer screens generally have only thousands of pixels per row or per column, far fewer than that needed for displaying the information of all

genes completely in the same screen. Therefore, we made the following plots instead.

We first made boxplots of FPKM medians for each stage (**Additional Fig. 19**). It appears that RNA transcripts with REEs, when regarded as a whole, did have a drop in their abundance (here quantified as the median of all gene-level FPKM medians) as development progresses from the stage where the REE is observed (i.e., the “starting stage”).

a

b

c

d

e

Additional Fig. 19. The developmental expression pattern of genes with and without REEs. For each subplot, the starting stage is the stage whose observation of REE on each gene was used to split the genes into “Has REE in the starting stage” and “Does not have REE in the starting stage”, and the “subsequent stage” is the stage that is after the starting stage along the development timeline. In addition, for each subplot only genes with FPKM > 0.1 at the starting stage was considered. The number shown below each boxplot is the median of all FPKM medians (without the transformation of $\log_{10}(\text{FPKM median} + 1 \times 10^{-4})$) for that boxplot. The starting stage for each subplot is: oocyte (GV) for (a), oocyte (MII) for (b), zygote for (c), 2-cell for (d), and 4-cell for (e).

We then examined this more closely by taking oocyte (GV) as an example stage: we picked all those genes whose FPKM median is larger than 200 in oocyte (GV), plotted their FPKM median along the developmental progress as a heatmap (**Additional Fig. 20**, left), and examined whether there was a (statistically significant) trend of FPKM dropping for each gene by running a simple linear regression of FPKM median against stage index (1 to 6 from oocyte (GV) to 8-cel, respectively) (**Additional Fig. 20**, right).

As one can see, when targeted by REE(s) in the oocyte (GV) stage, the regression coefficient for a given gene (if statistically significantly not zero) can only be negative, indicating that the overall pattern of this gene’s FPKM median is dropping along the development progress. In comparison,

genes not targeted by REE(s) in the oocyte (GV) stage might have positive coefficients.

These results suggested that genes targeted by REEs are likely to have a drop in their expression level as the development progresses.

Additional Fig. 20. An example heatmap for the gene FPKM median along the development progress, considering only those genes whose FPKM median is larger than 200 in the oocyte (GV) stage. Left, the heatmap. Right, the corresponding linear regression result for each gene, including the sign of regression coefficient and the binned Benjamini-Hochberg-adjusted p -value. The regression of per-stage FPKM medians against stages was carried out by treating the stages (from oocyte (GV) to 8-cell) as the integers 1 to 6, respectively.

We added these plots as **Supplementary Figures 29-30** to our manuscript (Page 6, Lines 130-132 in main text; Pages 50-54, Lines 545-574 in Supplementary Information).

References

- [1] S. Qiu, W. Li, H. Xiong, D. Liu, Y. Bai, K. Wu, X. Zhang, H. Yang, K. Ma, Y. Hou, and B. Li. Single-cell RNA sequencing reveals dynamic changes in A-to-I RNA editome during early human embryogenesis. *BMC Genomics*, 17:766, 2016. doi: 10.1186/s12864-016-3115-2.
- [2] Q. Q. Sha, Y. Z. Zhu, S. Li, Y. Jiang, L. Chen, X. H. Sun, L. Shen, X. H. Ou, and H. Y. Fan. Characterization of zygotic genome activation-dependent maternal mRNA clearance in mouse. *Nucleic Acids Res*, 48: 879–894, 2020. doi: 10.1093/nar/gkz1111.
- [3] R. Vassena, S. Boué, E. González-Roca, B. Aran, H. Auer, A. Veiga, and J. C. Izpisua Belmonte. Waves of early transcriptional activation and pluripotency program initiation during human preimplantation development. *Development*, 138:3699–709, 2011. doi: 10.1242/dev.064741.
- [4] I. Buchumenski, K. Holler, L. Appelbaum, E. Eisenberg, J. P. Junker, and E. Y. Levanon. Systematic identification of A-to-I RNA editing in zebrafish development and adult organs. *Nucleic Acids Res*, 49:4325–4337, 2021. doi: 10.1093/nar/gkab247.
- [5] G. Ramaswami, R. Zhang, R. Piskol, L. P. Keegan, P. Deng, M. A. O’Connell, and J. B. Li. Identifying RNA editing sites using RNA sequencing data alone. *Nat Methods*, 10:128–32, 2013. doi: 10.1038/nmeth.2330.

- [6] J. K. Pickrell, Y. Gilad, and J. K. Pritchard. Comment on "Widespread RNA and DNA sequence differences in the human transcriptome". *Science*, 335:1302; author reply 1302, 2012. doi: 10.1126/science.1210484.
- [7] W. Lin, R. Piskol, M. H. Tan, and J. B. Li. Comment on "Widespread RNA and DNA sequence differences in the human transcriptome". *Science*, 335:1302; author reply 1302, 2012. doi: 10.1126/science.1210624.
- [8] C. L. Kleinman and J. Majewski. Comment on "Widespread RNA and DNA sequence differences in the human transcriptome". *Science*, 335: 1302; author reply 1302, 2012. doi: 10.1126/science.1209658.
- [9] S. H. Roth, E. Y. Levanon, and E. Eisenberg. Genome-wide quantification of ADAR adenosine-to-inosine RNA editing activity. *Nat Methods*, 16:1131–1138, 2019. doi: 10.1038/s41592-019-0610-9.
- [10] S. Osenberg, N. Paz Yaacov, M. Safran, S. Moshkovitz, R. Shtrichman, O. Sherf, J. Jacob-Hirsch, G. Keshet, N. Amariglio, J. Itskovitz-Eldor, and G. Rechavi. Alu sequences in undifferentiated human embryonic stem cells display high levels of A-to-I RNA editing. *PLoS One*, 5: e11173, 2010. doi: 10.1371/journal.pone.0011173.
- [11] R. Shtrichman, I. Germanguz, R. Mandel, A. Ziskind, I. Nahor, M. Safran, S. Osenberg, O. Sherf, G. Rechavi, and J. Itskovitz-Eldor. Altered A-to-I RNA editing in human embryogenesis. *PLoS One*, 7: e41576, 2012. doi: 10.1371/journal.pone.0041576.
- [12] I. Germanguz, R. Shtrichman, S. Osenberg, A. Ziskind, A. Novak, H. Domev, I. Laevsky, J. Jacob-Hirsch, Y. Feiler, G. Rechavi, and J. Itskovitz-Eldor. ADAR1 is involved in the regulation of reprogramming human fibroblasts to induced pluripotent stem cells. *Stem Cells Dev*, 23:443–56, 2014. doi: 10.1089/scd.2013.0206.
- [13] T. Zhang, C. Yin, A. Fedorov, L. Qiao, H. Bao, N. Beknazarov, S. Wang, A. Gautam, R. M. Williams, J. C. Crawford, S. Peri, V. Studitsky, A. A. Beg, P. G. Thomas, C. Walkley, Y. Xu, M. Poptsova, A. Herbert, and S. Balachandran. ADAR1 masks the cancer immunotherapeutic promise of ZBP1-driven necroptosis. *Nature*, 606:594–602, 2022. doi: 10.1038/s41586-022-04753-7.

- [14] W. H. Cuddleston, J. Li, X. Fan, A. Kozenkov, M. Lalli, S. Khalique, S. Dracheva, E. A. Mukamel, and M. S. Breen. Cellular and genetic drivers of RNA editing variation in the human brain. *Nat Commun*, 13: 2997, 2022. doi: 10.1038/s41467-022-30531-0.
- [15] A. S. Nikitina, A. V. Lipatova, A. O. Goncharov, A. A. Kliuchnikova, M. A. Pyatnitskiy, K. G. Kuznetsova, A. Hamad, P. O. Vorobyev, O. N. Alekseeva, M. Mahmoud, Y. Shakiba, K. S. Anufrieva, G. P. Arapidi, M. V. Ivanov, I. A. Tarasova, M. V. Gorshkov, P. M. Chumakov, and S. A. Moshkovskii. Multiomic Profiling Identified EGF Receptor Signaling as a Potential Inhibitor of Type I Interferon Response in Models of Oncolytic Therapy by Vesicular Stomatitis Virus. *Int J Mol Sci*, 23, 2022. doi: 10.3390/ijms23095244.
- [16] O. Gabay, Y. Shoshan, E. Kopel, U. Ben-Zvi, T. D. Mann, N. Bressler, R. Cohen-Fultheim, A. A. Schaffer, S. H. Roth, Z. Tzur, E. Y. Levanon, and E. Eisenberg. Landscape of adenosine-to-inosine RNA recoding across human tissues. *Nat Commun*, 13:1184, 2022. doi: 10.1038/s41467-022-28841-4.
- [17] D. Katrekar, Y. Xiang, N. Palmer, A. Saha, D. Meluzzi, and P. Mali. Comprehensive interrogation of the ADAR2 deaminase domain for engineering enhanced RNA editing activity and specificity. *Elife*, 11, 2022. doi: 10.7554/eLife.75555.
- [18] T. Hwang, S. Kim, T. Chowdhury, H. J. Yu, K. M. Kim, H. Kang, J. K. Won, S. H. Park, J. H. Shin, and C. K. Park. Genome-wide perturbations of Alu expression and Alu-associated post-transcriptional regulations distinguish oligodendroglioma from other gliomas. *Commun Biol*, 5:62, 2022. doi: 10.1038/s42003-022-03011-w.
- [19] E. Picardi, L. Mansi, and G. Pesole. Detection of A-to-I RNA Editing in SARS-COV-2. *Genes (Basel)*, 13, 2021. doi: 10.3390/genes13010041.
- [20] N. I. Vlachogiannis, S. Tual-Chalot, E. Zormpas, F. Bonini, P. A. Ntouros, M. Pappa, V. K. Bournia, M. G. Tektonidou, V. L. Souliotis, C. P. Mavragani, K. Stamatelopoulos, A. Gatsiou, P. P. Sfikakis, and K. Stellos. Adenosine-to-inosine RNA editing contributes to type I interferon responses in systemic sclerosis. *J Autoimmun*, 125:102755, 2021. doi: 10.1016/j.jaut.2021.102755.

- [21] B. R. E. Ansell, S. N. Thomas, R. Bonelli, J. E. Munro, S. Freytag, and M. Bahlo. A survey of RNA editing at single-cell resolution links interneurons to schizophrenia and autism. *RNA*, 27:1482–1496, 2021. doi: 10.1261/rna.078804.121.
- [22] X. X. Huo, S. J. Wang, H. Song, M. D. Li, H. Yu, M. Wang, H. X. Gong, X. T. Qiu, Y. F. Zhu, and J. Y. Zhang. Roles of Major RNA Adenosine Modifications in Head and Neck Squamous Cell Carcinoma. *Front Pharmacol*, 12:779779, 2021. doi: 10.3389/fphar.2021.779779.
- [23] H. Piontkivska, B. Wales-McGrath, M. Miyamoto, and M. L. Wayne. ADAR Editing in Viruses: An Evolutionary Force to Reckon with. *Genome Biol Evol*, 13, 2021. doi: 10.1093/gbe/evab240.
- [24] A. A. Adetula, X. Fan, Y. Zhang, Y. Yao, J. Yan, M. Chen, Y. Tang, Y. Liu, G. Yi, K. Li, and Z. Tang. Landscape of tissue-specific RNA Editome provides insight into co-regulated and altered gene expression in pigs (. *RNA Biol*, 18:439–450, 2021. doi: 10.1080/15476286.2021.1954380.
- [25] D. Dierks, M. A. Garcia-Campos, A. Uzonyi, M. Safra, S. Edelheit, A. Rossi, T. Sideri, R. A. Varier, A. Brandis, Y. Stelzer, F. van Werven, R. Scherz-Shouval, and S. Schwartz. Multiplexed profiling facilitates robust m6A quantification at site, gene and sample resolution. *Nat Methods*, 18:1060–1067, 2021. doi: 10.1038/s41592-021-01242-z.
- [26] S. D. Knutson and J. M. Heemstra. Protein-based molecular recognition tools for detecting and profiling RNA modifications. *Curr Opin Struct Biol*, 69:1–10, 2021. doi: 10.1016/j.sbi.2020.12.006.
- [27] M. G. Kluesner, R. N. Tasakis, T. Lerner, A. Arnold, S. Wüst, M. Binder, B. R. Webber, B. S. Moriarity, and R. Pecori. MultiEditR: The first tool for the detection and quantification of RNA editing from Sanger sequencing demonstrates comparable fidelity to RNA-seq. *Mol Ther Nucleic Acids*, 25:515–523, 2021. doi: 10.1016/j.omtn.2021.07.008.
- [28] C. Vesely and M. F. Jantsch. An I for an A: Dynamic Regulation of Adenosine Deamination-Mediated RNA Editing. *Genes (Basel)*, 12, 2021. doi: 10.3390/genes12071026.

- [29] Z. Wu, J. Zhou, X. Zhang, Z. Zhang, Y. Xie, J. B. Liu, Z. V. Ho, A. Panda, X. Qiu, P. Cejas, I. Cañadas, F. G. Akarca, J. M. McFarland, A. K. Nagaraja, L. B. Goss, N. Kesten, L. Si, K. Lim, Y. Liu, Y. Zhang, J. Y. Baek, Y. Liu, D. T. Patil, J. P. Katz, J. Hai, C. Bao, M. Stachler, J. Qi, J. J. Ishizuka, H. Nakagawa, A. K. Rustgi, K. K. Wong, M. Meyerson, D. A. Barbie, M. Brown, H. Long, and A. J. Bass. Reprogramming of the esophageal squamous carcinoma epigenome by SOX2 promotes ADAR1 dependence. *Nat Genet*, 53:881–894, 2021. doi: 10.1038/s41588-021-00859-2.
- [30] I. Buchumenski, S. H. Roth, E. Kopel, E. Katsman, A. Feiglin, E. Y. Levanon, and E. Eisenberg. Global quantification exposes abundant low-level off-target activity by base editors. *Genome Res*, 2021. doi: 10.1101/gr.275770.121.
- [31] J. E. San, S. Ngcapu, A. M. Kanzi, H. Tegally, V. Fonseca, J. Giandhari, E. Wilkinson, C. W. Nelson, W. Smidt, A. M. Kiran, B. Chimukanga, S. Pillay, L. Singh, M. Fish, I. Gazy, D. P. Martin, K. Khanyile, R. Lessells, and T. de Oliveira. Transmission dynamics of SARS-CoV-2 within-host diversity in two major hospital outbreaks in South Africa. *Virus Evol*, 7:veab041, 2021. doi: 10.1093/ve/veab041.
- [32] Y. Shiromoto, M. Sakurai, M. Minakuchi, K. Ariyoshi, and K. Nishikura. ADAR1 RNA editing enzyme regulates R-loop formation and genome stability at telomeres in cancer cells. *Nat Commun*, 12:1654, 2021. doi: 10.1038/s41467-021-21921-x.
- [33] H. Wang, S. Chen, J. Wei, G. Song, and Y. Zhao. A-to-I RNA Editing in Cancer: From Evaluating the Editing Level to Exploring the Editing Effects. *Front Oncol*, 10:632187, 2020. doi: 10.3389/fonc.2020.632187.
- [34] P. J. Teoh, M. Y. Koh, and W. J. Chng. ADARs, RNA editing and more in hematological malignancies. *Leukemia*, 35:346–359, 2021. doi: 10.1038/s41375-020-01076-2.
- [35] L. Mansi, M. A. Tangaro, C. Lo Giudice, T. Flati, E. Kopel, A. A. Schaffer, T. Castrignanò, G. Chillemi, G. Pesole, and E. Picardi. REDI-portal: millions of novel A-to-I RNA editing events from thousands of RNAseq experiments. *Nucleic Acids Res*, 49:D1012–D1019, 2021. doi: 10.1093/nar/gkaa916.

- [36] J. Behroozi, S. Shahbazi, M. R. Bakhtiarizadeh, and H. Mahmoodzadeh. Genome-Wide Characterization of RNA Editing Sites in Primary Gastric Adenocarcinoma through RNA-seq Data Analysis. *Int J Genomics*, 2020:6493963, 2020. doi: 10.1155/2020/6493963.
- [37] A. Belkadi, G. Thareja, A. Halama, Y. Mahmoud, D. Jones, S. Agnew, J. Malek, and K. Suhre. Identification of genetic variants controlling RNA editing and their effect on RNA structure stabilization. *Eur J Hum Genet*, 28:1753–1762, 2020. doi: 10.1038/s41431-020-0688-7.
- [38] S. E. James, S. Ngcapu, A. M. Kanzi, H. Tegally, V. Fonseca, J. Giandhari, E. Wilkinson, B. Chimukangara, S. Pillay, L. Singh, M. Fish, I. Gazy, K. Khanyile, R. Lessells, and T. de Oliveira. High Resolution analysis of Transmission Dynamics of Sars-Cov-2 in Two Major Hospital Outbreaks in South Africa Leveraging Intra-host Diversity. *medRxiv*, 2020. doi: 10.1101/2020.11.15.20231993.
- [39] H. Tsvion-Visbord, E. Kopel, A. Feiglin, T. Sofer, R. Barzilay, T. Ben-Zur, O. Yaron, D. Offen, and E. Y. Levanon. Increased RNA editing in maternal immune activation model of neurodevelopmental disease. *Nat Commun*, 11:5236, 2020. doi: 10.1038/s41467-020-19048-6.
- [40] M. Hanan, A. Simchovitz, N. Yayon, S. Vaknine, R. Cohen-Fultheim, M. Karmon, N. Madrer, T. M. Rohrlisch, M. Maman, E. R. Bennett, D. S. Greenberg, E. Meshorer, E. Y. Levanon, H. Soreq, and S. Kadener. A Parkinson’s disease CircRNAs Resource reveals a link between circ- SLC8A1 and oxidative stress. *EMBO Mol Med*, 12:e11942, 2020. doi: 10.15252/emmm.201911942.
- [41] A. A. Schaffer and E. Y. Levanon. ALU A-to-I RNA Editing: Millions of Sites and Many Open Questions. *Methods Mol Biol*, 2181:149–162, 2021. doi: 10.1007/978-1-0716-0787-9_9.
- [42] S. Di Giorgio, F. Martignano, M. G. Torcia, G. Mattiuz, and S. G. Conticello. Evidence for host-dependent RNA editing in the transcriptome of SARS-CoV-2. *Sci Adv*, 6:eabb5813, 2020. doi: 10.1126/sciadv.abb5813.
- [43] O. An, K. T. Tan, Y. Li, J. Li, C. S. Wu, B. Zhang, L. Chen, and H. Yang. CSI NGS Portal: An Online Platform for Automated NGS

Data Analysis and Sharing. *Int J Mol Sci*, 21, 2020. doi: 10.3390/ijms21113828.

- [44] I. C. Vallecillo-Viejo, N. Liscovitch-Brauer, J. F. Diaz Quiroz, M. F. Montiel-Gonzalez, S. E. Nemes, K. J. Rangan, S. R. Levinson, E. Eisenberg, and J. J. C. Rosenthal. Spatially regulated editing of genetic information within a neuron. *Nucleic Acids Res*, 48:3999–4012, 2020. doi: 10.1093/nar/gkaa172.
- [45] C. Lo Giudice, M. A. Tangaro, G. Pesole, and E. Picardi. Investigating RNA editing in deep transcriptome datasets with REDIttools and REDlportal. *Nat Protoc*, 15:1098–1131, 2020. doi: 10.1038/s41596-019-0279-7.
- [46] M. Barak, H. T. Porath, G. Finkelstein, B. A. Knisbacher, I. Buchumenski, S. H. Roth, E. Y. Levanon, and E. Eisenberg. Purifying selection of long dsRNA is the first line of defense against false activation of innate immunity. *Genome Biol*, 21:26, 2020. doi: 10.1186/s13059-020-1937-3.

Reviewer comments:

Reviewer #1 (Remarks to the Author: Overall significance):

The authors have addressed all the concerns raised by this reviewer. My comments on overall significance and impact are similar to my previous comments.

Reviewer #1 (Remarks to the Author: Strength of the claims):

No additional question

Reviewer #4 (Remarks to the Author: Overall significance):

In this manuscript, the authors collected 2071 RNA-seq data and compiled the largest human embryonic A-to-I RNA editome, firstly identified thousands of REE and proposed REE may enhance maternal mRNA clearance. Their findings are interesting, but before the manuscript can be published in *Communications Biology*, some questions should be addressed, at least, the authors should support enough evidences to make sure all the sites that they identified are truly A-to-I RNA editing sites, especially the sites with editing levels of 100%.

Reviewer #4 (Remarks to the Author: Strength of the claims):

The detailed problems are below:

1. spelling mistakes, figure 1b, “seprarte” should be “separate”.
2. In this study, the authors adapted the pipeline published by Ramaswami et al., but in

Ramaswami's pipeline they removed PCR duplicates and required variants with a base quality score ≥ 25 and a mapping quality score ≥ 20 , which are not found in this study, while these two are very important, especially removing PCR duplicates for single cell RNA-seq, the authors should update their pipeline.

3. While the authors acquired the pipeline using RNA-seq alone, and discard all the known SNPs from several powerful study, so, expectedly, there will be no or fewer A-to-I candidates overlapped with DNA variants, and there is no sense for Figure 1c and 1d. At the same time, the authors should present the proportions or percentages of A-to-G or A-to-G/T-to-C in all 12 or 6 mismatch types, and the triple motif of identified A-to-I candidates and so on, to supporting their pipeline can be used to embryonic RNA-seq datasets. At the same time, I agreed with reviewer #3, the authors should present their results in the same way as other RNA editing papers, for example, like Additional Fig. 13., they should present the Additional Fig. 14. in the same way, present the proportion or percentage of 12 or 6 mismatch types without type like "A>G;T>C", while we can calculate the false positive rate of the pipeline using the proportion or percentage of A-to-G in all 12 mismatch types or A-to-G/T-to-C in all 6 mismatch types.

4. In total, the authors identified 989191 editing sites, but the authors didn't present how they obtained these sites from these 2071 samples, in general, if there are many samples, we only combined the A-to-I editing sites from the samples with false positive rate $\leq 5\%$ or smaller, for example, in Ramaswami's nature method paper, the proportion of A-to-G mismatch was required to be at least 80%, and the samples with high false positive rate and low A-to-G or A-to-G/T-to-C should be discarded for this step. The authors should supply the mapping rates, sequencing depth and A-to-G or A-to-G/T-to-C proportions for each samples.

5. Agreed with reviewer #3, there is so many sites with editing levels of 100%, the authors should extract the sites with editing levels of 100% and present the triple motif of these sites, present the IGV figure with aligned reads of some samples and make sure these sites are real A-to-I RNA editing sites. In supplementary figure 11a, all editing levels of chr21_33264079 are 100%, and its triple motif is CAA which is not ADAR's preference motif (DOI: 10.1038/ncomms1324), to remove the unknown SNP, some studies removes the variants with level $\geq 97\%$ in a single sample, so the authors should carefully check their sites and supply evidences to make sure the sites with editing levels of 100% are real rather not unknown SNPs.

6. The samples' number influences the identified A-to-I editing sites' number, more samples output more sites, while there are 1707 normal samples and only 274 abnormal samples, that's why more editing sites were found in normal samples in Figure 1e, so for the comparison analyses between normal and abnormal samples should be more carefully done to avoiding the effect of big sample number difference.

7. For the correlation with ADAR1 expression, the authors said most of the correlations were still not statistically significant (Additional Fig. 17), but in the other investigating edits in human embryos using pilot embryo RNA-seq paper (DOI:10.1186/s12864-016-3115-2) cited by this study, "Correlation tests indicated that RNA-editing levels were strongly correlated with ADAR expression levels (Pearson's correlation test, $P = 5.99E-13$, $r = 0.74$; Fig. 2d)." and "It is worth noting that although the ADARB1 expression levels remained low in cells of all stages investigated, we detected a moderate correlation between the ADARB1 expression levels and the A-to-I RNA-editing levels (Pearson's correlation test, $P = 3.38E-4$, $r = 0.42$; Fig. 2e)." . Maybe for this analysis, the authors should not combine the data from different datasets, while different datasets would

influence the correlation test results.

Response to reviewers' comments

We thank the reviewers for their further valuable comments, and have revised the manuscript accordingly. Specifically,

1. We added the mapping rates, sequencing depths, and A-to-G proportions across all 12 nucleotide changes for each sample as Supplementary Tables 10-12, respectively, along with their distribution plot for reference (Supplementary Fig. 31a-c).
2. We emphasized certain technical details (removing PCR duplicates and requiring reads to have an average quality score ≥ 25 and a mapping quality score ≥ 20) to the identification pipeline to make key steps clear, and fixed the misspelling in Figure 1B.

All the changes in the manuscript and supplementary material are highlighted in blue font. Please see the replies to the specific reviewer comments below for further details. We thank again for the editors and reviewers' efforts.

Reviewer comments:

Reviewer #1 (Remarks to the Author: Overall significance):

The authors have addressed all the concerns raised by this reviewer. My comments on overall significance and impact are similar to my previous comments.

We thank again the reviewer for all the efforts and valuable suggestions.

Reviewer #1 (Remarks to the Author: Strength of the claims):

No additional question

We thank again the reviewer for all the efforts and valuable suggestions.

Reviewer #4 (Remarks to the Author: Overall significance):

In this manuscript, the authors collected 2071 RNA-seq data and compiled the largest human embryonic A-to-I RNA editome, firstly identified thousands of REE and proposed REE may enhance maternal mRNA clearance. Their findings are interesting, but before the manuscript can be published in *Communications Biology*, some questions should be addressed, at least, the authors should support enough evidences to make sure all the sites that they identified are truly A-to-I RNA editing sites, especially the sites with editing levels of 100%.

We thank the reviewer's comments.

We have refined additional technical details and provided additional sample metrics to the identification pipeline of A-to-I edits to support the reliability of our identification (see responses to Comments 1, 2, 3, 4, 6, and 7 below).

We also agree with the potential pitfalls in the reliability about edits with editing levels of 100%, as raised by the reviewer (Comment 5); we then pointed out that, because the proportion of such edits is very low in early-stage embryos where REE (i.e., the main focus of this manuscript) was mostly investigated, excluding such sites should not affect the major conclusions of REE drawn from our analysis.

Please refer to the specific response to each comment for more details.

Reviewer #4 (Remarks to the Author: Strength of the claims):

The detailed problems are below:

1. spelling mistakes, figure 1b, "seprarte" should be "separate".

We thank the reviewer for pointing out this misspelling in Figure 1B.
We have fixed this and updated it in the manuscript (Page 36, Line 861 in main text).

2. In this study, the authors adapted the pipeline published by Ramaswami et al., but in Ramaswami's pipeline they removed PCR duplicates and required variants with a base quality score ≥ 25 and a mapping quality score ≥ 20 , which are not found in this study, while these two are very important, especially removing PCR duplicates for single cell RNA-seq, the authors should update their pipeline.

Thanks for the comments.

We are sorry that we had not described these details clearly in the main text; we had only described in the Supplementary Notes 6. Specifically, we have previously removed PCR duplicates (Page 7, Lines 147-149 in Supplementary Information) and required the reads to have an average quality score ≥ 25 (Page 7, Lines 142-144 in Supplementary Information) and a mapping quality score ≥ 20 (Page 7, Lines 149-151 in Supplementary Information). We have further polished the text to make it clear. Specifically, we have emphasized that the duplicates we removed are PCR duplicates (Page 7, Lines 147-149 in Supplementary Information) and also reiterated these technical details in the Results (Page 5, Lines 94-97 in main text) and Methods section (Page 14, Lines 331-333 in main text) of the main text to make them clear.

3. While the authors acquired the pipeline using RNA-seq alone, and discard all the known SNPs from several powerful study, so, expectedly, there will be no or fewer A-to-I candidates overlapped with DNA variants, and there is no sense for Figure 1c and 1d.

We thank the reviewer for pointing out our imprecise statement for these figures.

The Figure 1c was used to demonstrate the design of the dataset for validating whether our pipeline could result in the expected observation that "there will be no or fewer A-to-I candidates overlapped with DNA variants": because this dataset is a DNA-RNA-paired single-cell sequencing dataset, we can know whether a candidate RNA variant is truly an RNA editing, or arises from DNA variants.

While the Figure 1d was used to show that we did observe this expectedly, we are sorry that we did not state the result of Figure 1d precisely enough in the main text. As we have stated in the Supplementary Notes 1, “Comparison of the edits in each cell with its genomic variants revealed that all A375 cells with at least one edit identified had a zero ratio of genomic variant-overlapping edits after, but not before, our filtering (Fig. 1d). This result supported the validity of our pipeline.” (Page 2, Lines 26-29 in Supplementary Information). Therefore, while the “raw” case in Figure 1d (i.e., before applying the filters in our pipeline) did have a generally very low ratio of identified A-to-I RNA edits that overlapped with the DNA variants in the same cell across samples, the “filter” case in Figure 1d (i.e., after applying the filters in our pipeline) had a zero ratio of such edits (which is logically correct but intuitively confusing if stated as “generally very low”).

We have fixed the related description in Results (Page 5, Lines 101-105 in main text), Figure 1 legend (Page 37, Lines 864-866 in main text), and Supplementary Notes 1 (Page 2, Lines 26-29 in Supplementary Information).

At the same time, the authors should present the proportions or percentages of A-to-G or A-to-G/T-to-C in all 12 or 6 mismatch types, and the triple motif of identified A-to-I candidates and so on, to supporting their pipeline can be used to embryonic RNA-seq datasets.

At the same time, I agreed with reviewer #3, the authors should present their results in the same way as other RNA editing papers, for example, like Additional Fig. 13., they should present the Additional Fig. 14. in the same way, present the proportion or percentage of 12 or 6 mismatch types without type like “A>G;T>C”, while we can calculate the false positive rate of the pipeline using the proportion or percentage of A-to-G in all 12 mismatch types or A-to-G/T-to-C in all 6 mismatch types.

Thanks for the comment.

For the proportions or percentages of A-to-G and A-to-G/T-to-C in all 12 or 6 mismatch types, we have provided such information for both the DNA-RNA-paired single-cell dataset (in the form of a histogram of mean (#variants) across all samples; Supplementary Figure 1) and the embryonic datasets (Figure 1f and Supplementary Figure 6). Because we have identified edits that are either strand-definite (A-to-G) or strand-ambiguous (A-to-G/T-to-C; see Step (13) in Supplementary Notes 6), we had provided the proportion in Figure 1f and Supplementary Figure 6 as the ratio of “the union of A-to-G variants and A-to-G/T-to-C variants” to all variants.

We updated the previous Additional Fig. 14 (i.e., the previous Supplementary Fig. 1) in the same way as the previous Additional Fig. 13, as shown below (**Additional Fig. 1** and also the new **Supplementary Fig. 1a**; Page 37, Lines 866-868 in main text and Pages 14-15, Lines 317-325 in Supplementary Information). From this result we observed that the mean A- to-G proportion is 63.23%, and we computed the false discovery rate (which we believe is the false positive rate raised by the reviewer) following the way of Ramaswami et al.¹, which is $((1-63.23\%)/11)/63.23\%=5.29\%$.

Additional Fig. 1. Percentage of all possible 12 types of simple nucleotide changes (for the DNA-RNA-paired single-cell dataset). Bar height and error bars display the mean and standard deviation, respectively, of percentage across the samples. Only samples with at least 10 edits identified were considered in this plot.

We have also explicitly stated the definition of strand-definite/strand-ambiguous edits in Step (13) of **Supplementary Notes 6** (Page 12, Lines 273-274 in Supplementary Information), as well as how the A-to-G proportion was computed in figure legends of the **Figure 1f**, and the **Supplementary Figure 6** (Page 37, Lines 875-877 in main text; Page 23, Lines 373-375 in Supplementary Information).

For the triple motif of identified A-to-I candidate edits, we have previously provided this information for the embryonic dataset (Figure 1h). For the DNA-RNA-paired single-cell dataset, we ran the analysis and provided the motif below (**Additional Fig. 2** and also **Supplementary Figure 1b**).

Additional Fig. 2. The triple motif of edits identified in the DNA-RNA- paired single-cell dataset.

4. In total, the authors identified 989191 editing sites, but the authors didn't present how they obtained these sites from these 2071 samples, in general, if there are many samples, we only combined the A-to-I editing sites from the samples with false positive rate $\leq 5\%$ or smaller, for example, in Ramaswami's nature method paper, the proportion of A-to-G mismatch was required to be at least 80%, and the samples with high false positive rate and low A-to-G or A-to-G/T-to-C should be discarded for this step. The authors should supply the mapping rates, sequencing depth and A-to-G or A-to-G/T-to-C proportions for each samples.

We thank the reviewer for pointing out this important problem.

We have previously presented how we obtained these sites from these samples, as detailed in Supplementary Notes 6.

While A-to-G ratio is an important metric evaluating the overall identification of RNA editing, this was not included in the Ramaswami et al. pipeline and thus we did not use it to discard certain samples. In addition, while from our observation, some samples in pre- and post-implantation of human early embryonic development do have a low A-to-G ratio (Fig. 1f, Supplementary Fig. 6, and **Additional Fig. 3c** below), we note that studies of RNA editing in such samples are scarce. In particular, for the only pilot identification of RNA edits in such samples, as done by Qiu et al.², we noted that they used much fewer pre-implantation samples than us (and they did not use samples later than morula), so we are not sure whether their reporting an average A-to-G ratio of $\geq 80\%$ in early samples is a general phenomenon; in addition, we have tried our best to reproduce their results in the first revision of this manuscript, yet their identification cannot be reproduced even after several rounds of communications (see the “Comparison with Qiu et al identification results” and “Additional Notes 1” attached). Therefore, we cannot conclude that such low ratio is mainly due to non-biological technical artifacts. We also note that, in this manuscript we mostly focused on the REE (Recurrent Embryonic Edits) that require $\geq 50\%$ chances of occurring in a given stage; such requirement should make its analysis more robust to potential false positives mentioned by the reviewer here than general edits’. For the last question, we have added the mapping rates, sequencing depth, and A-to-G proportions across all 12 nucleotide changes for each sample as **Supplementary Tables 10-12**, respectively, along with their distribution plot for reference (**Additional Fig. 3** and also **Supplementary Fig. 31a- c**). We noted that, while the overall mapping rate seems to be low, it is not directly relevant to our study, because we have discarded some reads during the pipeline (e.g., keeping only those reads with average base quality score ≥ 25 , as discussed in the reviewer’s second comment above). In addition, we have tried our best to make the identification results as reliable as possible, including sticking to datasets that are from published works (so their quality, at least with respect to the scientific questions in their corresponding works, should be reliable), and adapting the Ramaswami et al. pipeline where mapping rate was not used as a threshold.

a

b

Additional Fig. 3. The distribution plot of mapping rates (defined as the ratio of “the number of reads kept after GATK recalibration” to “the total number of trimmed reads prior to BWA mapping”) (a), sequencing depth (defined as the samtools coverage-reported mean depth across the whole genome) (b), and A-to-G proportions across all 12 nucleotide changes (c) across samples analyzed.

5. Agreed with reviewer #3, there is so many sites with editing levels of 100%, the authors should extract the sites with editing levels of 100% and present the triple motif of these sites, present the IGV figure with aligned reads of some samples and make sure these sites are real A-to-I RNA editing sites. In supplementary figure 11a, all editing levels of chr21_33264079 are 100%, and its triple motif is CAA which is not ADAR's preference motif (DOI: 10.1038/ncomms1324), to remove the unknown SNP, some studies removes the variants with level 97% in a single sample, so the authors should carefully check their sites and supply evidences to make sure the sites with editing levels of 100% are real rather not unknown SNPs.

Thanks for the comments.

We have previously rightly acknowledged that edits (not editing sites, because the editing level might change in different samples) with a editing

level of 100% should be examined with caution (Supplementary Fig. 4). We have updated the legend of Supplementary Figure 4 and Supplementary Figure 11 to emphasize that such edits might still not be real A-to-I RNA edits (Page 21, Lines 358-359 in Supplementary Information).

Nevertheless, we cannot conclude immediately from our computational results that these edits are definitely noise and not truly A-to-I edits; additional experimental validations are needed to confirm which of these RNA edits are truly A-to-I edits and which are not. On the other hand, the proportion of such RNA edits with an editing level of 100% is very low in early-stage embryos where REE – the main focus of this manuscript – was mostly investigated (Supplementary Fig. 4b; note that Supplementary Fig. 4a and 4b present the same distribution in log10-scale and untransformed scale, respectively). Therefore, excluding such sites should not affect the major conclusions of REE drawn from our analysis, and thus we did not explicitly exclude them from our identification results.

6. The samples' number influences the identified A-to-I editing sites' number, more samples output more sites, while there are 1707 normal samples and only 274 abnormal samples, that's why more editing sites were found in normal samples in Figure 1e, so for the comparison analyses between normal and abnormal samples should be more carefully done to avoiding the effect of big sample number difference.

Thanks for the comments.

We agree with the reviewer's comment that the large difference in sample size between normal and abnormal samples could contribute to the identification of more edits in normal samples, and have reported again the number of samples again (which are 1,797 normal and 274 abnormal samples; Page 13, Lines 319-321 in main text) in the legend of Figure 1e (Page 37, Lines 869-873 in main text).

We assume that "the comparison analyses between normal and abnormal samples" the reviewer refers to are Supplementary Figure 2 (where all edits were examined) and all analyses in the Result section "Certain REE-matching edits could undergo organized loss in embryos with uniparental disomy and those from elder mothers" (where REE were examined in normal and abnormal embryos (or embryos from younger and elder mothers) paired in the corresponding case study). For Supplementary Figure 2, we have reduced the effect of different sample size by focusing on samples from

the same stage, as shown in Supplementary Fig. 5 – the sample sizes of normal and abnormal samples for each of the stages analyzed in Supplementary Figure 2 are similar. For the analysis in “Certain REE-matching edits could undergo organized loss in embryos with uniparental disomy and those from elder mothers”, we only used those normal and abnormal embryos (or embryos from younger and elder mothers) in the corresponding published case study, where they are designed to have comparable sample sizes.

Therefore, we have tried our best to reduce the effect of sample size difference between normal and abnormal samples in all comparisons between normal and abnormal samples.

7. For the correlation with ADAR1 expression, the authors said most of the correlations were still not statistically significant (Additional Fig. 17), but in the other investigating edits in human embryos using pilot embryo RNA-seq paper (DOI: 10.1186/s12864-016-3115-2) cited by this study, “Correlation tests indicated that RNA-editing levels were strongly correlated with ADAR expression levels (Pearson’s correlation test, $P = 5.99E-13$, $r = 0.74$; Fig. 2d).” and “It is worth noting that although the ADARB1 expression levels remained low in cells of all stages investigated, we detected a moderate correlation between the ADARB1 expression levels and the A-to-I RNA-editing levels (Pearson’s correlation test, $P = 3.38E-4$, $r = 0.42$; Fig. 2e).” . Maybe for this analysis, the authors should not combine the data from different datasets, while different datasets would influence the correlation test results.

Thanks for the comments.

We first note that these two types of correlations are based on different definitions of per-sample “editing level”. In particular, while in the Qiu et al. paper (DOI: 10.1186/s12864-016-3115-2)² it has been defined as “... edited bases per million mapped bases in each cell.”, here in the Additional Fig. 17 in the last revision (also Supplementary Fig. 21) it’s AEI (Alu-Editing Index), an editing activity metric that computes weighted average level of A-to-G across genome-wide Alu elements, where the weights are the coverage of each site³. This AEI definition of editing level has been found to have a weak correlation with ADAR1 and ADAR2 (which is the alias of ADARB1) expression levels, as stated in Roth et al.: “Expression levels of the ADAR enzymes correlate, in many tissues, with editing measures. Thus, they are widely used as a proxy for estimating global ADAR activity. However, more often than not, the correlation between ADAR expression and editing activity

is rather weak (Fig. 3b and Supplementary Figs. 6 and 7), presumably due to additional layers of ADARs regulation at the protein level.”³ (we also provided Roth et al.’s Fig. 3b below as a reference (**Additional Fig. 4**)). This is consistent with our observation in Supplementary Fig. 21.

Additional Fig. 4. Figure 3b from Roth et al.³.

In addition, the A-to-I identification result by Qiu et al. (DOI: 10.1186/s12864-016-3115-2)² could not be reproduced after several rounds of communications (see response to Comment 4 above and also the attached Additional Notes 1). Moreover, early embryos need to undergo drastic transcriptome changes, such as maternal mRNA clearance⁴ and zygotic genomic activation⁵, which might further complicate the relationship between the expression level of *ADAR* and editing level. Therefore, we chose not to dissect this relationship further.

We also note that we have previously decided to combine datasets from different resources to identify from many normal embryos of the same stage

from different sources those biological signals (in particular, REE in this manuscript) that are more robust to non-biological artifacts such as sampling bias. Therefore, in future study, if one'd like to investigate the correlation further, we recommend to carry out this investigation first on this combined normal embryo set, rather than on individual datasets with smaller sample sizes.

References

1. G. Ramaswami, R. Zhang, R. Piskol, L.P. Keegan, P. Deng, M.A. O'Connell, and J. B. Li. Identifying RNA editing sites using RNA sequencing data alone. *Nat Methods*, 10:128–32, 2013. doi: 10.1038/nmeth.2330.
2. S. Qiu, W. Li, H. Xiong, D. Liu, Y. Bai, K. Wu, X. Zhang, H. Yang, K. Ma, Y. Hou, and B. Li. Single-cell RNA sequencing reveals dynamic changes in A-to-I RNA editome during early human embryogenesis. *BMC Genomics*, 17:766, 2016. doi: 10.1186/s12864-016-3115-2.
3. S.H. Roth, E.Y. Levanon, and E. Eisenberg. Genome-wide quantification of ADAR adenosine-to-inosine RNA editing activity. *Nat Methods*, 16: 1131–1138, 2019. doi: 10.1038/s41592-019-0610-9.
4. Q. Q. Sha, Y. Z. Zhu, S. Li, Y. Jiang, L. Chen, X. H. Sun, L. Shen, X. H. Ou, and H. Y. Fan. Characterization of zygotic genome activation-dependent maternal mRNA clearance in mouse. *Nucleic Acids Res*, 48: 879–894, 2020. doi: 10.1093/nar/gkz1111.
5. R. Vassena, S. Boué, E. González-Roca, B. Aran, H. Auer, A. Veiga, and J. C. Izpisua Belmonte. Waves of early transcriptional activation and pluripotency program initiation during human preimplantation development. *Development*, 138:3699–709, 2011. doi: 10.1242/dev.064741.